# Breaking the centralized barrier for cross-device federated learning*

**Sai Praneeth Karimireddy**
EPFL
sai.karimireddy@epfl.ch

**Martin Jaggi**
EPFL
martin.jaggi@epfl.ch

**Satyen Kale**
Google Research
satyenkale@google.com

**Mehryar Mohri**
Google Research
mohri@google.com

**Sashank J. Reddi**
Google Research
sashank@google.com

**Sebastian U. Stich**
EPFL
sebastian.stich@epfl.ch

**Ananda Theertha Suresh**
Google Research
theertha@google.com

## Abstract

Federated learning (FL) is a challenging setting for optimization due to the heterogeneity of the data across different clients which can cause a *client drift* phenomenon. In fact, designing an algorithm for FL that is uniformly better than simple centralized training has been a major open problem thus far. In this work, we propose a general algorithmic framework, MIME, which i) mitigates client drift and ii) adapts an arbitrary centralized optimization algorithm such as momentum and Adam to the cross-device federated learning setting. MIME uses a combination of *control-variates* and *server-level optimizer state* (e.g. momentum) at every client-update step to ensure that each local update mimics that of the centralized method run on i.i.d. data. We prove a reduction result showing that MIME can translate the convergence of a generic algorithm in the centralized setting into convergence in the federated setting. Moreover, we show that, when combined with momentum-based variance reduction, MIME is provably *faster than any centralized method*–the first such result. We also perform a thorough experimental exploration of MIME's performance on real world datasets (implemented here).

## 1 Introduction

Federated learning (FL) is an increasingly important large-scale learning framework where the training data remains distributed over a large number of clients, which may be mobile phones or network sensors [38, 37, 43, 44, 28]. A server then orchestrates the clients to train a single model, here referred to as a *server model*, without ever transmitting client data over the network, thereby providing some basic levels of data privacy and security.

Two important settings are distinguished in FL [28, Table 1]: the *cross-device* and the *cross-silo* settings. The cross-silo setting corresponds to a relatively small number of reliable clients, typically organizations, such as medical or financial institutions. In contrast, in the *cross-device* federated learning setting, the number of clients may be extremely large and include, for example, all 3.5 billion active android phones [25]. Thus, in that setting, we may never make even a single pass over

---

*This work was also appears under the alternative title "Mime: Mimicking Centralized Stochastic Algorithms in Federated Learning" [31].

the entire clients' data during training. The cross-device setting is further characterized by resource-poor clients communicating over a highly unreliable network. Together, the essential features of this setting give rise to unique challenges not present in the cross-silo setting. In this work, we are interested in the more challenging cross-device setting, for which we will formalize and study stochastic optimization algorithms. Importantly, recent advances in FL optimization, such as SCAFFOLD [32] or FedDyn [1], are *not anymore applicable* since they are designed for the cross-silo setting.

**The problem.**    The de facto standard algorithm for the cross-device setting is FEDAVG [43], which performs multiple SGD updates on the available clients before communicating to the server. While this approach can reduce the *frequency* of communication required, performing multiple steps on the same client can lead to 'over-fitting' to its atypical local data, a phenomenon known as *client drift* [32]. This in turn leads to slower convergence and can, somewhat counter-intuitively, require *larger total communication* [69]. Despite significant attention received from the optimization community, the communication complexity of heterogeneous cross-device has not improved upon that of simple centralized methods, which take no local steps (aka SERVER-ONLY methods). Furthermore, algorithmic innovations such as momentum [59, 14], adaptivity [35, 75, 77], and clipping [71, 72, 76] are critical to the success of deep learning applications. The lack of a theoretical understanding of the impact of multiple client steps has also hindered adapting these techniques in a principled manner into the client updates, in order to replace the vanilla SGD update of FEDAVG.

To overcome such deficiencies, we propose a new framework, MIME, that mitigates client drift and can adapt an arbitrary centralized optimization algorithm, e.g. SGD with momentum or Adam, to the federated setting. In each local client update, MIME uses global optimizer state, e.g. momentum or adaptive learning rates, and an SVRG-style correction to mimic the updates of the centralized algorithm run on i.i.d. data. This optimizer state is computed only at the server level and kept fixed throughout the local steps, thereby avoiding overfitting to the atypical local data of any single client.

**Contributions.**    We summarize our main results below.

- MIME **framework.** We formalize the cross-device federated learning problem, and propose a new framework MIME that can adapt arbitrary centralized algorithms to this setting.
- **Convergence result.** We prove a result showing that MIME successfully reduces client drift. We also prove that the convergence of any generic algorithm in the centralized setting translates convergence of its MIME version in the federated setting.
- **Speed-up over centralized methods.** By carefully tracking the bias introduced due to multiple local steps, we prove that MIME with momentum-based variance reduction (MVR) can beat a lower bound for centralized methods, thus breaking a fundamental barrier. This is the first such result in FL, and also the first general result showing asymptotic speed-up due to local steps.
- **Empirical validation.** We propose a simpler variant, MIMELITE, with an empirical performance similar to MIME. We report the results of thorough experimental analysis demonstrating that both MIME and MIMELITE indeed converge faster than FEDAVG.

**Related work.**    *Analysis of* FEDAVG: Much of the recent work in federated learning has focused on analyzing FEDAVG. For identical clients, FEDAVG coincides with parallel SGD, for which [78] derived an analysis with asymptotic convergence. Sharper and more refined analyses of the same method, sometimes called local SGD, were provided by [56], and more recently by [57], [47], [34], and [70], for identical functions. Their analysis was extended to heterogeneous clients in [68, 74, 32, 34, 36]. [11] derived a tight characterization of FedAvg with quadratic functions and demonstrated the sensitivity of the algorithm to both client and server step sizes. Matching upper and lower bounds were recently given by [32] and [69] for general functions, proving that FEDAVG can be slower than even SGD for heterogeneous data, due to the *client-drift*.

*Comparison to* SCAFFOLD: For the cross-silo setting where the number of clients is relatively low, [32] proposed the SCAFFOLD algorithm, which uses control-variates (similar to SVRG) to correct for client drift. However, their algorithm crucially relies on *stateful clients* which repeatedly participate in the training process. FedDyn [1] reduces the communication requirements, but also requires persistent stateful clients. In contrast, we focus on the cross-device setting where clients may be visited only once during training and where they are *stateless* (and thus SCAFFOLD and FedDyn are inapplicable). This is akin to the difference between the finite-sum (corresponding to cross-silo) and stochastic (cross-device) settings in traditional centralized optimization [39].

*Comparison to FedAvg and variants:* [26] and [67] observed that using *server momentum* significantly improves over vanilla FEDAVG. This idea was generalized by [49], who replaced the server update with an arbitrary optimizer, e.g. Adam. However, these methods only modify the server update while using SGD for the client updates. We henceforth refer to this *meta algorithm* as FedAvg. FedAvgSGD, FedAvgMom, FedAvgAdam denote specific instantiations of the server optimizer in FedAvg with SGD, Momentum or Adam. MIME, on the other hand, ensures that every *local client update* resembles the optimizer e.g. MIME would apply momentum in every client update and not just at the server level. Beyond this, [40] proposed to add a regularizer to ensure client updates remain close. However, this may slow down convergence (cf. Fig. 5 and [32, 66]). Other orthogonal directions which can be combined with MIME include tackling computation heterogeneity, where some clients perform many more updates than others [66], improving fairness by modifying the objective [44, 41], incorporating differential privacy [20, 2, 61], Byzantine adversaries [48, 65, 30], secure aggregation [8, 24], etc. We defer additional discussion to the extensive survey by [28].

*Momentum based variance reduction.* Initial optimal methods for stochastic non-convex optimization like SPIDER [17] and SARAH [46] required intermittently computing very large batch gradients. Subsequently, it was shown that momentum based variance reduction (MVR) methods obtained a similar optimal rate without needing such large batch gradient computations [62, 14]. Momentum is an exponential moving average of many stochastic gradients and so it has much smaller variance than the stochastic gradients themselves. However, because these gradients are computed at different parameters it also has a bias. MVR adds a small additional correction term which significantly reduces this bias and provides improved rates.

## 2   Problem setup

This section formalizes the problem of cross-device federated learning [28]. Cross-device FL is characterized by a large number of client devices like mobile phones which may potentially connect to the server at most once. Due to their transient nature, it is not possible to store any state on the clients, precluding an algorithm like SCAFFOLD. Furthermore, each client has only a few samples, and there is wide heterogeneity in the samples across clients. Finally, communication is a major bottleneck and a key metric for optimization in this setting is the number of communication rounds.

Thus, our objective will be to minimize the following quantity within the fewest number of client-server communication rounds:

$$f(\boldsymbol{x}) = \mathbb{E}_{i \sim \mathcal{C}} \left[ f_i(\boldsymbol{x}) := \frac{1}{n_i} \sum_{\nu=1}^{n_i} f_i(\boldsymbol{x}; \zeta_{i,\nu}) \right]. \tag{1}$$

Here, $f_i$ denotes the loss function of client $i$ and $\{\zeta_{i,1}, \ldots, \zeta_{i,n_i}\}$ its local data. Since the number of clients is extremely large, while the size of each local data is rather modest, we represent the former as an expectation and the latter as a finite sum. In each round, the algorithm samples a subset of clients (of size $S$) and performs some updates to the server model. Due to the transient and heterogeneous nature of the clients, it is easy to see that the problem becomes intractable with arbitrarily dissimilar clients. Thus, it is **necessary** to assume bounded dissimilarity across clients.

**(A1)** $G^2$-**BGV** or bounded inter-client gradient variance: there exists $G \geq 0$ such that

$$\mathbb{E}_{i \sim \mathcal{C}}[\|\nabla f_i(\boldsymbol{x}) - \nabla f(\boldsymbol{x})\|^2] \leq G^2 \,, \ \forall \boldsymbol{x} \,.$$

Next, we also characterize the variance in the Hessians.

**(A2)** $\delta$-**BHV** or bounded Hessian variance: Almost surely, the loss function of any client $i$ satisfies

$$\|\nabla^2 f_i(\boldsymbol{x}; \zeta) - \nabla^2 f(\boldsymbol{x})\| \leq \delta \,, \ \forall \boldsymbol{x} \,.$$

This is in contrast to the usual smoothness assumption that can be stated as:

**(A2\*)** $L$-**smooth**: $\|\nabla^2 f_i(\boldsymbol{x}; \zeta)\| \leq L \,, \ \forall \boldsymbol{x} \,,$ a.s. for any $i$.

Note that if $f_i(\boldsymbol{x}; \zeta)$ is $L$-smooth then (A2) is satisfied with $\delta \leq 2L$, and hence (A2) is *weaker* than (A2\*). In realistic examples we expect the clients to be similar and hence that $\boldsymbol{\delta} \ll \boldsymbol{L}$. In addition, we assume that $f(\boldsymbol{x})$ is bounded from below by $f^\star$ and is $L$-smooth, as is standard.

# 3 Mime framework

In this section we describe how to adapt an arbitrary centralized optimizer (referred to as the "base" optimizer) which may have internal state (e.g. momentum) to the federated learning problem (1) while ensuring there is no client-drift. Algorithm 4 describes our framework. We develop two variants, MIME and MIMELITE, which consist of three components i) a base optimizer we are seeking to mimic, ii) the global (server) optimizer state computation, and iii) the local client updates.

---

**Algorithm 1** Mime and MimeLite

**input:** initial $x$ and $s$, learning rate $\eta$ and base optimizer $\mathcal{B} = (\mathcal{U}, \mathcal{V})$
**for** each round $t = 1, \cdots, T$ **do**
  sample subset $\mathcal{S}$ of clients
  **communicate** $(x, s)$ to all clients $i \in \mathcal{S}$
  **communicate** $c \leftarrow \frac{1}{|\mathcal{S}|} \sum_{j \in \mathcal{S}} \nabla f_j(x)$ (only Mime)
  **on client** $i \in \mathcal{S}$ **in parallel do**
    initialize local model $y_i \leftarrow x$
    **for** $k = 1, \cdots, K$ **do**
      sample mini-batch $\zeta$ from local data
      $g_i \leftarrow \nabla f_i(y_i; \zeta) - \nabla f_i(x; \zeta) + c$ **(Mime)**
      $g_i \leftarrow \nabla f_i(y_i; \zeta)$ **(MimeLite)**
      update $y_i \leftarrow y_i - \eta \mathcal{U}(g_i, s)$
    **end for**
    compute full local-batch gradient $\nabla f_i(x)$
    **communicate** $(y_i, \nabla f_i(x))$
  **end on client**
  $s \leftarrow \mathcal{V}\left(\frac{1}{|\mathcal{S}|} \sum_{i \in \mathcal{S}} \nabla f_i(x), s\right)$ (update optimizer state)
  $x \leftarrow \frac{1}{|\mathcal{S}|} \sum_{i \in \mathcal{S}} y_i$ (update server parameters)
**end for**

---

**Base optimizer.** We assume the centralized base optimizer we are imitating can be decomposed into two steps: an *update step* $\mathcal{U}$ which updates the parameters $x$, and a *optimizer state update step* $\mathcal{V}(\cdot)$ which keeps track of global optimizer state $s$. Each step of the base optimizer $\mathcal{B} = (\mathcal{U}, \mathcal{V})$ uses a gradient $g$ to update the parameter $x$ and the optimizer state $s$ as follows:

$$x \leftarrow x - \eta \mathcal{U}(g, s),$$
$$s \leftarrow \mathcal{V}(g, s). \quad \text{(BASEOPT)}$$

As an example, consider SGD with momentum. The state here is the momentum $m_t$ and uses the following update steps:

$$x_t = x_{t-1} - \eta\left((1-\beta)\nabla f_i(x_{t-1}) + \beta m_{t-1}\right),$$
$$m_t = (1-\beta)\nabla f_i(x_{t-1}) + \beta m_{t-1}.$$

Thus, SGD with momentum can be represented in the above generic form with $\mathcal{U}(g, s) = (1-\beta)g + \beta s$ and $\mathcal{V}(g, s) = (1-\beta)g + \beta s$. Table 5 in Appendix shows how other algorithms like Adam, Adagrad, etc. can be represented in this manner. We keep the update $\mathcal{U}$ to be linear in the gradient $g$, whereas $\mathcal{V}$ can be more complicated. This implies that while the parameter update step $\mathcal{U}$ is relatively resilient to receiving a biased gradient $g$ while $\mathcal{V}$ can be much more sensitive.

**Compute optimizer state globally, apply locally.** When updating the optimizer state of the base algorithm, we use only the gradient computed at the server parameters. Further, they remain fixed throughout the local updates of the clients. This ensures that these optimizer state remain unbiased and representative of the global function $f(\cdot)$. At the end of the round, the server performs

$$s \leftarrow \mathcal{V}\left(\frac{1}{|\mathcal{S}|} \sum_{i \in \mathcal{S}} \nabla f_i(x), s\right),$$
$$\nabla f_i(x) = \frac{1}{n_i} \sum_{\nu=1}^{n_i} \nabla f_i(x; \zeta_{i,\nu}). \quad \text{(OPTSTATE)}$$

Note that we use full-batch gradients computed at the server parameters $x$, not client parameters $y_i$.

**Local client updates.** Each client $i \in \mathcal{S}$ performs $K$ updates using $\mathcal{U}$ of the base algorithm and a minibatch gradient. There are two variants possible corresponding to MIME and MIMELITE differentiated using colored boxes. Starting from $y_i \leftarrow x$, repeat the following $K$ times

$$y_i \leftarrow y_i - \eta \mathcal{U}(g_i, s) \quad \text{(CLIENTSTEP)}$$

where $g_i \leftarrow \nabla f_i(y_i; \zeta)$ for MIMELITE, and $g_i \leftarrow \nabla f_i(y_i; \zeta) - \nabla f_i(x; \zeta) + \frac{1}{|\mathcal{S}|} \sum_{j \in \mathcal{S}} \nabla f_j(x)$ for MIME. MIMELITE simply uses the local minibatch gradient whereas MIME uses an SVRG

style correction [27]. This is done to reduce the noise from sampling a local mini-batch. While this correction yields faster rates in theory (and in practice for convex problems), in deep learning applications we found that MIMELITE closely matches the performance of MIME.

Finally, there are two modifications made in practical FL: we weight all averages across the clients by the number of datapoints $n_i$ [43], and we perform $K$ epochs instead of $K$ steps [66].

## 4 Theoretical analysis of Mime

Table 1 summarizes the rates of MIME (highlighted in blue) and MIMELITE (highlighted in green) and compares them to SERVER-ONLY methods when using SGD, Adam and momentum methods as the base algorithms. We will first examine the convergence of MIME and MIMELITE with a generic base optimizer and show that its properties are preserved in the federated setting. We then examine a specific momentum based base optimizer, and prove that Mime and MimeLite can be asymptotically faster than the best server-only method. This is the first result to prove the usefulness of local steps and demonstrate asymptotic speed-ups.

### 4.1 Convergence with a generic base optimizer

We will prove a generic reduction result demonstrating that if the underlying base algorithm converges, and is robust to slight perturbations, then MIME and MIMELITE also preserve the convergence of the algorithm when applied to the federated setting with additinoal local steps.

**Theorem I.** *Suppose that we have $G^2$ inter-client gradient variance (A1), $L$-smooth $\{f_i\}$ (A2*), and $\sigma^2$ intra-client gradient variance (A3). Further, suppose that the updater $\mathcal{U}$ of our base-optimizer $\mathcal{B} = (\mathcal{U}, \mathcal{V})$ satisfies i) linearity for a fixed state $\mathbf{s}$: $\mathcal{U}(\mathbf{g}_1 + \mathbf{g}_2; \mathbf{s}) = \mathcal{U}(\mathbf{g}_1; \mathbf{s}) + \mathcal{U}(\mathbf{g}_2; \mathbf{s})$, and ii) Lipschitzness: $\|\mathcal{U}(\mathbf{g}; \mathbf{s})\| \le B\|\mathbf{g}\|$ for some $B \ge 0$. Then, running* MIME *or* MIMELITE *with $K$ local updates and step-size $\eta$ is equivalent to running a **centralized** algorithm with step-size $\tilde{\eta} := K\eta \le \frac{1}{2LB}$, and updates*

$$\mathbf{x}_t \leftarrow \mathbf{x}_{t-1} - \tilde{\eta}\mathcal{U}(\mathbf{g}_t + \boxed{\mathbf{e}_t}, \mathbf{s}_{t-1}), \text{ and}$$

$$\mathbf{s}_t \leftarrow \mathcal{V}(\mathbf{g}_t, \mathbf{s}_{t-1}), \text{ where we have}$$

*an unbiased gradient $\mathbb{E}_t[\mathbf{g}_t] = \nabla f(\mathbf{x}_{t-1})$, with variance bounded as*

$$\mathbb{E}_t\|\mathbf{g}_t - \nabla f(\mathbf{x}_{t-1})\|^2 \le \begin{cases} \frac{G^2}{S} & \text{MIME}, \\ \frac{G^2}{S} + \frac{\sigma^2}{KS} & \text{MIMELITE}. \end{cases}$$

*and finally a small error bounded as*

$$\frac{1}{B^2 L^2 \tilde{\eta}^2}\mathbb{E}_t\|\boxed{\mathbf{e}_t}\|^2 \le \begin{cases} \mathbb{E}_t\|\mathbf{g}_t\|^2 & \text{MIME}, \\ \mathbb{E}_t\|\mathbf{g}_t\|^2 + G^2 + \frac{\sigma^2}{K} & \text{MIMELITE}. \end{cases}$$

Here, we have proven that MIME and MIMELITE truly mimic the centralized base algorithm with very small perturbations—the magnitude of $\mathbf{e}_t$ is $\mathcal{O}(\tilde{\eta}^2)$. The key to the result is the linearity of the parameter update step $\mathcal{U}(\,\cdot\,; \mathbf{s})$. By separating the base optimizer into a very simple parameter step $\mathcal{U}$ and a more complicated optimizer state update step $\mathcal{V}$, we can ensure that commonly used algorithms such as momentum, Adam, Adagrad, and others all satisfy this property. Armed with this general reduction, we can easily obtain specific convergence results.

**Corollary II** ((Mime/MimeLite) with SGD)**.** *Given that the conditions in Theorem I are satisfied, let us run $T$ rounds with $K$ local steps using SGD as the base optimizer and output $\mathbf{x}^{out}$. This output satisfies $\mathbb{E}\|\nabla f(\mathbf{x}^{out})\|^2 \le \epsilon$ for $F := f(\mathbf{x}_0) - f^\star$, $\tilde{G}^2 := G^2 + \sigma^2/K$ and*

- *$\mu$-PL inequality: $\eta = \tilde{\mathcal{O}}\left(\frac{1}{\mu KT}\right)$, and*

$$T = \begin{cases} \tilde{\mathcal{O}}\left(\frac{LG^2}{\mu S \epsilon} + \frac{LF}{\mu}\log\left(\frac{1}{\epsilon}\right)\right) & \text{MIME}, \\ \tilde{\mathcal{O}}\left(\frac{L\tilde{G}^2}{\mu S \epsilon} + \frac{L\tilde{G}}{\mu \sqrt{\epsilon}} + \frac{LF}{\mu}\log\left(\frac{1}{\epsilon}\right)\right) & \text{MIMELITE}. \end{cases}$$

- ***Non-convex:** for $\eta = \mathcal{O}\left(\sqrt{\frac{FS}{L\tilde{G}^2 TK^2}}\right)$, and*

$$T = \begin{cases} \mathcal{O}\left(\frac{LG^2 F}{S\epsilon^2} + \frac{LF}{\epsilon}\right) & \text{MIME}, \\ \mathcal{O}\left(\frac{L\tilde{G}^2 F}{S\epsilon^2} + \frac{L^2 \tilde{G}F}{\epsilon^{3/2}} + \frac{LF}{\epsilon}\right) & \text{MIMELITE}. \end{cases}$$

Table 1: Number of communication rounds required to reach $\|\nabla f(\boldsymbol{x})\|^2 \leq \epsilon$ (log factors are ignored) with $S$ clients sampled each round. All analyses except SCAFFOLD assume $G^2$ bounded gradient dissimilarity (A1). All analyses assume $L$-smooth losses, except MimeLiteMVR and MimeMVR, which only assume $\delta$ bounded Hessian dissimilarity (A2). Convergence of SCAFFOLD depends on the total number of clients $N$ which is potentially infinite. FEDAVG and MIMELITE are slightly slower than the server-only methods due to additional drift terms in most cases. MIME is the fastest and either matches or improves upon the optimal statistical rates (first term in the rates). In fact, MimeMVR and MimeLiteMVR beat lower bounds for any server-only method when $\delta \ll L$.

| Algorithm | Non-convex | $\mu$-PL inequality |
|---|---|---|
| SCAFFOLD[a] [32] | $\left(\frac{N}{S}\right)^{\frac{2}{3}} \frac{L}{\epsilon}$ | $\frac{N}{S} + \frac{L}{\mu}$ |
| SGD | | |
|   SERVER-ONLY [21] | $\frac{LG^2}{S\epsilon^2} + \frac{L}{\epsilon}$ | $\frac{G^2}{\mu S\epsilon} + \frac{L}{\mu}$ |
|   MimeLiteSGD$\equiv$ FedAvgSGD [c] | $\frac{LG^2}{S\epsilon^2} + \frac{L^2 G}{\epsilon^{3/2}} + \frac{L}{\epsilon}$ | $\frac{G^2}{\mu S\epsilon} + \frac{LG}{\mu\sqrt{\epsilon}} + \frac{L}{\mu}$ |
|   MimeSGD | $\frac{LG^2}{S\epsilon^2} + \frac{L}{\epsilon}$ | $\frac{G^2}{\mu S\epsilon} + \frac{L}{\mu}$ |
| ADAM | | |
|   SERVER-ONLY [75][b] | $\frac{L}{\epsilon - G^2/S}$ | − |
|   MimeLiteAdam[bc] | $\frac{L\sqrt{S}}{\epsilon - G^2/S}$ | − |
|   MimeAdam[b] | $\frac{L}{\epsilon - G^2/S}$ | − |
| Momentum Variance Reduction (MVR) | | |
|   SERVER-ONLY [14] | $\frac{LG}{\sqrt{S}\epsilon^{3/2}} + \frac{G^2}{S\epsilon} + \frac{L}{\epsilon}$ | − |
|   MimeLiteMVR[d] | $\frac{\delta(G+\sigma)}{\epsilon^{3/2}} + \frac{G^2+\sigma^2}{\epsilon} + \frac{\delta}{\epsilon}$ | − |
|   **MimeMVR**[d] | $\frac{\delta G}{\sqrt{S}\epsilon^{3/2}} + \frac{G^2}{S\epsilon} + \frac{\delta}{\epsilon}$ | − |
| SERVER-ONLY lower bound [5] | $\Omega\left(\frac{LG}{\sqrt{S}\epsilon^{3/2}} + \frac{G^2}{S\epsilon} + \frac{L}{\epsilon}\right)$ | $\Omega\left(\frac{G^2}{S\epsilon}\right)$ |

[a] Num. clients ($N$) can be same order as num. total rounds or even $\infty$, making the bounds vacuous.
[b] Adam requires large batch-size $S \geq G^2/\epsilon$ to converge [50, 75]. Convergence of FedAdam with client sampling is unknown ([49] only analyze with full client participation).
[c] Requires $K \geq \sigma^2/G^2$ number of local updates. Typically, intra-client variance is small ($\sigma^2 \lesssim G^2$).
[d] Requires $K \geq L/\delta$ number of local updates. Faster than the lower bound (and hence any SERVER-ONLY algorithm) when $\delta \ll L$ i.e. our methods can take advantage of Hessian similarity, whereas SERVER-ONLY methods cannot. In worst case, $\delta \approx L$ and all methods are comparable.

If we take a sufficient number of local steps $K \geq G^2/\sigma^2$, then we have $\tilde{G} = \mathcal{O}(G)$ in the above rates. On comparing with the rates in Table 1 for SERVER-ONLY SGD, we see that MIME exactly matches its rates. MIMELITE matches the asymptotic term but has a few higher order terms. Note that when using SGD as the base optimizer, MIMELITE becomes exactly the same as FEDAVG and hence has the same rate of convergence.

**Corollary III** ((Mime/MimeLite) with Adam). *Suppose that the conditions in Theorem I are satisfied, and further $|\nabla_j f_i(\boldsymbol{x})| \leq H$ for any coordinate $j \in [d]$. Then let us run $T$ rounds using Adam as the base optimizer with $K$ local steps, $\beta_1 = 0$, $\varepsilon_0 > 0$, $\eta \leq \varepsilon_0^2/KL(H+\varepsilon_0)$, and any $\beta_2 \in [0, 1)$. Output $\boldsymbol{x}^{out}$ chosen randomly from $\{\boldsymbol{x}_1, \ldots \boldsymbol{x}_T\}$ satisfies $\mathbb{E}\|\nabla f(\boldsymbol{x}^{out})\|^2 \leq \epsilon$ for*

$$T = \begin{cases} \mathcal{O}\left(\frac{LF(H+\varepsilon_0)^2}{\varepsilon_0^2(\epsilon - \tilde{G}^2/S)}\right) & \text{MIME Adam ,} \\ \mathcal{O}\left(\frac{LF(H+\varepsilon_0)^2\sqrt{S}}{\varepsilon_0^2(\epsilon - \tilde{G}^2/S)}\right) & \text{MIMELITE Adam .} \end{cases}$$

*where $F := f(\boldsymbol{x}_0) - f^\star$, $\tilde{G}^2 := G^2 + \sigma^2/K$.*

Note that here $\varepsilon_0$ represents a small positive parameter used in Adam for regularization, and is different from the error $\epsilon$. Similar to the SERVER-ONLY analysis of Adam [75], we assume $\beta_1 = 0$

and that batch size is large enough such that $S \geq G^2/\epsilon$. A similar analysis can also be carried out for AdaGrad, and other novel variants of Adam [42].

## 4.2 Circumventing server-only lower bounds

The rates obtained above, while providing a safety-check, do not beat those of the SERVER-ONLY approach. The previous best rates for cross-device FL correspond to MimeLiteSGD which is $\mathcal{O}(\frac{LG^2}{S\epsilon^2} + \frac{L^2G}{\epsilon^{3/2}})$ [34, 36, 69]. While, using a separate server-learning rate can remove the effect of the second term [33], this at best matches the rate of SERVER-ONLY SGD $\mathcal{O}(\frac{LG^2}{S\epsilon^2})$. This is significantly slower than simply using momentum based variance reduction (MVR) as in in the FL setting (SERVER-ONLY MVR) which has a communication complexity of $\mathcal{O}(\frac{LG}{\sqrt{S}\epsilon^{3/2}})$ [14]. Thus, even though the main reason for studying local-step methods was to improve the communication complexity, none thus far show such improvement. The above difficulty of beating SERVER-ONLY may not be surprising given the two sets of strong lower bounds known.

**Necessity of local steps.** Firstly, [5] show a gradient oracle lower bound of $\Omega(\frac{LG}{\sqrt{S}\epsilon^{3/2}})$. This matches the complexity of MVR, and hence at first glance it seems that SERVER-ONLY MVR is optimal. However, the lower bound is really only on the number of gradients computed and not on the number of clients sampled (sample complexity) [18], or number of rounds of communication required. In particular, multiple local updates increases number of gradients computed *without needing additional communication* offers us a potential way to side-step such lower bounds. A careful analysis of the bias introduced as a result of such local steps is a key part of our analysis.

**Necessity of $\delta$-BHD.** A second set of lower bounds directly study the number of communication rounds required in heterogeneous optimization [6, 69]. These results prove that there exist settings where local steps provide no advantage and SERVER-ONLY methods are optimal. This however contradicts real world experimental evidence [43]. As before, the disparity arises due to the contrived settings considered by the lower bounds. For distributed optimization (with full client participation) and convex quadratic objectives, $\delta$-BHD (A2) was shown to be a sufficient [54, 51] and *necessary* [6] condition to circumvent these lower bounds and yield highly performant methods. We similarly leverage $\delta$-BHD (A2) to design novel methods which significantly extend prior results to i) all smooth non-convex functions (not just quadratics), and ii) cross-device FL with client sampling.

We now state our convergence results with momentum based variance reduction (MVR) as the base-algorithm since it is known to be optimal in the SERVER-ONLY setting.

**Theorem IV.** *For $L$-smooth $f$ with $G^2$ gradient dissimilarity* (A1)*, $\delta$ Hessian dissimilarity* (A2) *and $F := (f(\boldsymbol{x}^0) - f^\star)$, let us run MVR as the base algorithm for $T$ rounds with $K \geq L/\delta$ local steps and generate an output $\boldsymbol{x}^{out}$. This output satisfies $\mathbb{E}\|\nabla f(\boldsymbol{x}^{out})\|^2 \leq \epsilon$ for*

- **MimeMVR** : $\eta = \mathcal{O}\left(\min\left(\frac{1}{\delta K}, \left(\frac{SF}{G^2TK^3}\right)^{1/3}\right)\right)$, *momentum* $\beta = 1 - \mathcal{O}\left(\frac{\delta^2 S^{2/3}}{(TG^2)^{2/3}}\right)$, *and*

$$T = \mathcal{O}\left(\frac{\delta G F}{\sqrt{S}\epsilon^{3/2}} + \frac{G^2}{S\epsilon} + \frac{\delta F}{\epsilon}\right).$$

- **MimeLiteMVR** : $\eta = \mathcal{O}\left(\min\left(\frac{1}{\delta K}, \left(\frac{F}{\hat{G}^2TK^3}\right)^{1/3}\right)\right)$, *momentum* $\beta = 1 - \mathcal{O}\left(\frac{\delta^2}{(T\hat{G}^2)^{2/3}}\right)$, *and*

$$T = \mathcal{O}\left(\frac{\delta \hat{G} F}{\epsilon^{3/2}} + \frac{\hat{G}^2}{\epsilon} + \frac{\delta F}{\epsilon}\right).$$

Here, we define $\hat{G}^2 := G^2 + \sigma^2$ and the expectation in $\mathbb{E}\|\nabla f(\boldsymbol{x}^{out})\|^2 \leq \epsilon$ is taken both over the sampling of the clients during the running of the algorithm, the sampling of the mini-batches in local updates, and the choice of $\boldsymbol{x}^{out}$ (which is chosen randomly from the client iterates $\boldsymbol{y}_i$).

Remarkably, the rates of our methods are independent of $L$ and only depend on $\delta$. Thus, when $\delta \leq L$ and $\delta \leq L/S$ for MimeMVR and MimeLiteMVR, the rates beat the server only lower bound of $\Omega(\frac{LG}{\sqrt{S}\epsilon^{3/2}})$. In fact, if the Hessian variance is small and $\delta \approx 0$, our methods only need $\mathcal{O}(1/\epsilon)$ rounds to communicate. Intuitively, our results show that local steps are very useful when heterogeneity (represented by $\delta$) is smaller than optimization difficulty (captured by smoothness constant $L$).

MimeMVR uses a momentum parameter $\beta$ of the order of $(1 - \mathcal{O}(TG^2)^{-2/3})$ i.e. as $T$ increases, $\beta$ asymptotically approaches 1. In contrast, previous analyses of distributed momentum (e.g. [73]) prove rates of the form $\frac{G^2}{S(1-\beta)\epsilon^3}$, which are worse than that of standard SGD by a factor of $\frac{1}{1-\beta}$.

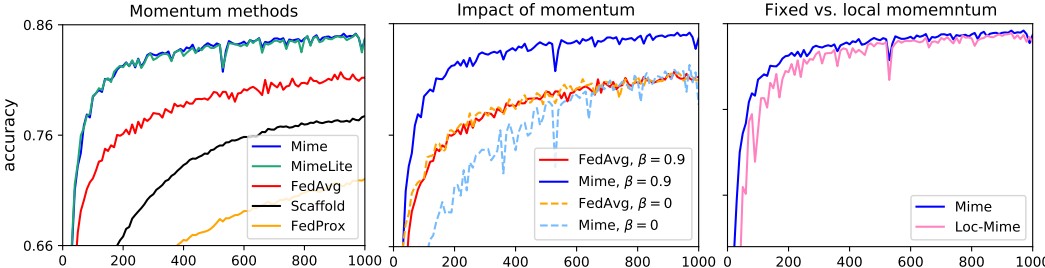

Figure 1: **Mime**, **MimeLite**, **FedAvg**, **Scaffold**, **FedProx**, and **Loc-Mime** with SGD+momentum using 10 local epochs, run on EMNIST62 and a 2 hidden layer (300u-100) MLP. (Left) Mime and MimeLite are nearly identical and outperform the rest (7× faster). (Center) Mime makes better use of momentum than FedAvg, with a large increase in performance. (Right) Locally adapting momentum slows down convergence and makes it more unstable.

Thus, ours is also the first result which theoretically showcases the usefulness of using large momentum in distributed and federated learning. While we only prove the utility of local steps for MimeMVR, we believe our theory can be extended to other local update methods as well.

Our analysis is highly non-trivial and involves two crucial ingredients: i) computing the momentum at the server level to ensure that it remains unbiased and then applying it locally during every client update to reduce variance, and ii) carefully keeping track of the bias introduced via additional local steps. Our experiments (Sec. 5) verify our theoretical insights are indeed applicable in deep learning settings as well. See App. B for a proof sketch and App. G–H detailed proofs.

## 5 Experimental analysis on real world datasets

We run experiments on *natively* federated datasets to confirm our theory and accurately measure real world performance. Our main findings are i) MIME and MIMELITE consistently outperform FEDAVG, and ii) momentum and adaptivity significantly improves performance.

### 5.1 Setup

**Algorithms.** We consider three (meta) algorithms: FEDAVG, MIME, and MIMELITE. Each of these adapt four base optimizers: SGD, momentum, Adam, and Adagrad.
FEDAVG follows [49] who run multiple epochs of SGD on each client sampled, and then aggregate the net client updates. This aggregated update is used as a pseudo-gradient in the base optimizer (called server optimizer). The learning rate for the server optimizer is fixed to 1 as in [67]. This is done to ensure all algorithms have the same number of hyper-parameters.
MIME and MIMELITE follow Algorithm 4 and also run a fixed number of epochs on the client. However, note that this requires communicating both the full local-batch gradient as well as the parameter updates doubling the communication required to be sent by the client. For a fairer comparison, we split the sampled clients in MIME and MIMELITE into two groups–the first communicates only full local-batch gradient and the latter communicates only parameter updates. Thus, all methods have **equal client communication** to the server. This variant retains the convergence guarantees up to constants (details in the Appendix). We also run Loc-MIME where instead of keeping the global optimizer state fixed, we update it locally within the client. The optimizer state is reset after the round finishes. In all methods, aggregation is weighted by the number of samples on the clients.

**Datasets and models.** We run five simulations on three real-world federated datasets: EMNIST62 with i) a linear classifier, ii) an MLP, and iii) a CNN, iv) a charRNN on Shakespeare, and v) an LSTM for next word prediction on StackOverflow, all accessed through Tensorflow Federated [60]. The learning rates were individually tuned and other optimizer hyper-parameters such as $\beta$ for momentum, $\beta_1$, $\beta_2$, $\varepsilon_0$ for Adam and AdaGrad were left to their default values, unless explicitly stated otherwise. We refer to Appendix C for additional setup details and discussion.

### 5.2 Ablation and comparative study
In order to study the different algorithms, we train a 2 hidden layer (300$\mu$-100) MLP on EMNIST62 with 10 local epochs for 1k rounds and use SGD+momentum (with tuned $\beta$) as the base optimizer.

**Mime ≈ MimeLite > FedAvg > SCAFFOLD > FedProx.** Fig. 1 (left) shows MIME and MIMELITE have nearly identical performance, and are about 7× faster than FedAvg. This implies

Table 2: Validation % accuracies after training for 1000 rounds. Best results for each dataset is underlined and the best within each base optimizer is bolded. The number of clients sampled per round has been reduced for MIME and MIMELITE to ensure all methods have **equal client and server communication**. Final accuracies obtained by MIME and MIMELITE are competitive with FEDAVG, especially with adaptive base optimizers. FEDAVG seems unstable with Adam.

| | | EMNIST logistic | EMNIST CNN | Shakespeare | StackOverflow |
|---|---|---|---|---|---|
| SGD | FedAvgSGD | 66.8 | **85.8** | **56.7** | **23.8** |
| | MimeLiteSGD | 66.8 | **85.8** | **56.7** | **23.8** |
| | MimeSGD | **67.4** | 85.3 | 56.1 | 12.5 |
| MOMENTUM | FedAvgMom | 67.4 | 85.7 | **55.4** | **22.2** |
| | MimeLiteMom | 67.4 | **86.0** | 49.8 | 19.9 |
| | MimeMom | **67.5** | 85.9 | 53.6 | 19.3 |
| ADAM | FedAvgAdam | 67.3 | 85.9 | 18.5 | 3.2 |
| | MimeLiteAdam | **68.0** | 86.4 | 54.0 | 21.5 |
| | MimeAdam | **68.0** | **86.6** | **54.1** | **22.8** |
| ADAGRAD | FedAvgAdagrad | **67.6** | 86.3 | 55.5 | **24.2** |
| | MimeLiteAdagrad | 66.6 | 85.5 | 56.8 | 23.8 |
| | MimeAdagrad | 67.4 | **86.3** | **57.1** | 14.7 |

our strategy of applying momentum to client updates is faster than simply using server momentum. FedProx [40] uses an additional regularizer $\mu$ tuned over $[0.1, 0.5, 1]$ ($\mu = 0$ is the same as FedAvg). Regularization does not seem to reduce client drift but still slows down convergence [66]. SCAF-FOLD [32] is also slower than Mime and FedAvg in this setup. This is because in cross-device setting with a large number of clients ($N = 3.4k$) means that each client is visited less than 6 times during the entire training (20 clients per round for 1k rounds). This means that the correction term utilized by SCAFFOLD uses control-variates which are quite stale (computed about 200 rounds ago) which slows down the convergence. In contrast, the SVRG correction term in Mime is computed using clients sampled in the current or previous rounds, and so is much more accurate.

**With momentum $>$ without momentum.** Fig. 1 (center) examines the impact of momentum on FedAvg and Mime. Momentum slightly improves the performance of FedAvg, whereas it has a significant impact on the performance of Mime. This is also in line with our theory and confirms that Mime's strategy of applying it locally at every client update makes better use of momentum.

**Fixed $>$ locally updated optimizer state.** Finally, we check how the performance of Mime changes if instead of keeping the momentum fixed throughout a round, we let it change. The latter is a way to combine global and local momentum. The momentum is reset at the end of the round ignoring the changes the clients make to it. Fig. 1 (right) shows that this *worsens* the performance, confirming that it is better to keep the global optimizer state fixed as predicted by our theory.

Together, the above observations validate all aspects of Mime (and MimeLite) design: compute statistics at the server level, and apply them unchanged at every client update.

### 5.3 Large scale comparison with equal server and client communication

We perform a larger scale study closely matching the setup of [49]. For both MIME and MIMELITE, only half the clients compute and transmit the updated parameters, and other half transmit the full local-batch gradients. Hence, client to server communication cost is the same for all methods for all clients. However, MIME and MIMELITE require sending additional optimization state to the clients. Hence, we also reduce the number of clients sampled in each round to ensure *sum total* of communication at each round is $40\times$ model size for EMNIST and Shakespeare experiments, and $100\times$ model size for the StackOverflow next word prediction experiment.

Since we only perform 1 local epoch, the hyper-parameters (e.g. epsilon for adaptive methods) are more carefully chosen following [49], and MIME and MIMELITE use significantly fewer clients per round, the difference between FEDAVG and MIME is smaller here. Table 2 summarizes the results.

For the image classification tasks of EMNIST62 logistic and EMNIST62 CNN, Mime and MimeLite with Adam achieve the best performance. Using momentum (both with SGD and in Adam) significantly improves their performance. In contrast, FedAvgAdam is more unstable with worse performance. This is because FedAvg is excessively sensitive to hyperparameters (cf. App. E).

We next consider the character prediction task on Shakespeare dataset, and next word prediction on StackOverflow. Here, the momentum based methods (SGD+momentum and Adam) are slower than their non-momentum counterparts (vanilla SGD and AdaGrad). This is because the mini-batch gradients in these tasks are *sparse*, with the gradients corresponding to tokens not in the mini-batch being zero. This sparsity structure is however destroyed when using momentum or Adam. For the same reason, Mime which uses an SVRG correction also significantly increases the gradient density.

**Discussion.** For traditional tasks such as image classification, we observe that Mime (especially with Adam) usually outperforms MimeLite which in turn outperforms FedAvg. These methods are able to successfully leverage momentum and adaptivity to improve performance. For tasks where the client gradients are sparse, the SVRG correction used by Mime hinders performance. Adapting our techniques to work with sparse gradients (à la Yogi [75]) could lead to further improvements. Also, note that we reduce communication by naïvely reducing the number of participating clients per round. More sophisticated approaches to save on client communication including quantization or sparsification [58, 3], or even novel algorithmic innovations [1] could be explored. Further, server communication could be reduced using memory efficient optimizers e.g. AdaFactor [55] or SM3 [4].

# 6  Conclusion

Our work initiated a formal study of the cross-device federated learning problem and provided theoretically justified algorithms. We introduced a new framework MIME which overcomes the natural client-heterogeneity in such a setting, and can adapt arbitrary centralized algorithms such as Adam without additional hyper-parameters. We demonstrated the superiority of MIME via strong convergence guarantees and empirical evaluations. Further, we proved that a particular instance of our method, MimeMVR, beat centralized lower-bounds, demonstrating that additional local steps can yield asymptotic improvements for the first time. We believe our analysis will be of independent interest beyond the federated setting for understanding the sample complexity of non-convex optimization, and for yielding improved analysis of decentralized optimization algorithms.

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
