# Supplementary material for MIME

## Contents of Appendix

# A How momentum can help reduce client drift

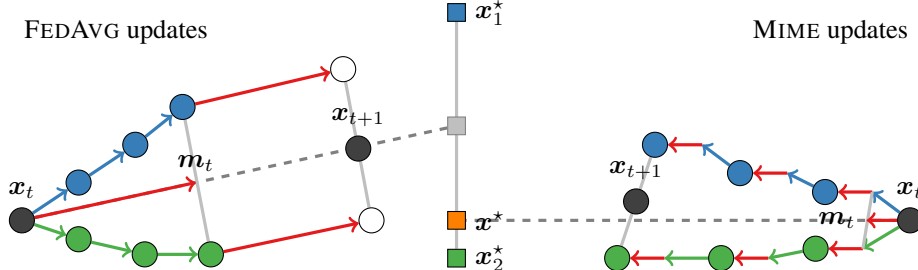

Figure 2: Client-drift in FEDAVG (left) and MIME (right) is illustrated for 2 clients with 3 local steps and momentum parameter $\beta = 0.5$. The local SGD updates of FEDAVG (shown using arrows for client 1 and client2) move towards the average of client optima $\frac{x_1^\star + x_2^\star}{2}$ which can be quite different from the true global optimum $x^\star$. Server momentum $m_t$ only speeds up the convergence to the wrong point in this case. In contrast, MIME uses unbiased momentum computed at the server parameter $x_t$ and applies it locally at every update. This keeps the updates of MIME closer to the true optimum $x^\star$.

In this section we examine the tension between reducing communication by running multiple client updates each round, and degradation in performance due to client drift [32]. To simplify the discussion, we assume a single client is sampled each round and that clients use full-batch gradients.

**Server-only approach.** A simple way to avoid the issue of client drift is to take no local steps. We sample a client $i \sim \mathcal{C}$ and run SGD with momentum (Mom) with momentum parameter $\beta$ and step size $\eta$:

$$
\begin{aligned}
\boldsymbol{x}_t &= \boldsymbol{x}_{t-1} - \eta\left((1-\beta)\nabla f_i(\boldsymbol{x}_{t-1}) + \beta\boldsymbol{m}_{t-1}\right), \\
\boldsymbol{m}_t &= (1-\beta)\nabla f_i(\boldsymbol{x}_{t-1}) + \beta\boldsymbol{m}_{t-1}.
\end{aligned}
\tag{2}
$$

Here, the gradient $\nabla f_i(\boldsymbol{x}_t)$ is *unbiased* i.e. $\mathbb{E}[\nabla f_i(\boldsymbol{x}_t)] = \nabla f(\boldsymbol{x}_t)$ and hence we are guaranteed convergence. However, this strategy can be communication-intensive and we are likely to spend all our time waiting for communication with very little time spent on computing the gradients.

FEDAVG **approach.** To reduce the overall communication rounds required, we need to make more progress in each round of communication. The FedAvg meta algorithm utilizes a base optimizer, a client learning rate and a server learning rate. Each client performs K local update steps of SGD using the client learning rate and communicates the net update (difference between final and initial parameters) to the server. This difference is then treated as a 'pseudo-gradient' and is input into the optimizer (say momentum or Adam) to update the server parameters using the server learning rate. When the base optimizer uses momentum, this momentum is computed at the server level using the pseudo-gradients and is referred to as server momentum.

Starting from $\boldsymbol{y}_0 = \boldsymbol{x}_{t-1}$, FEDAVG [43] runs multiple SGD steps on the sampled client $i \sim \mathcal{C}$

$$
\boldsymbol{y}_k = \boldsymbol{y}_{k-1} - \eta\nabla f_i(\boldsymbol{y}_{k-1}) \text{ for } k \in [K],
\tag{3}
$$

and then a pseudo-gradient $\tilde{\boldsymbol{g}}_t = -(\boldsymbol{y}_K - \boldsymbol{x}_t)$ replaces $\nabla f_i(\boldsymbol{x}_{t-1})$ in the SGDm algorithm (2). This is referred to as server-momentum since it is computed and applied only at the server level [26]. However, such updates give rise to *client-drift* resulting in performance worse than the naïve server-only strategy (2). This is because by using multiple local updates, (3) starts over-fitting to the local client data, optimizing $f_i(\boldsymbol{x})$ instead of the actual global objective $f(\boldsymbol{x})$. The net effect is that FEDAVG moves towards an incorrect point (see Fig 2, left). If $K$ is sufficiently large, approximately

$$
\boldsymbol{y}_K \rightsquigarrow \boldsymbol{x}_i^\star, \quad \text{where } \boldsymbol{x}_i^\star := \arg\min_{\boldsymbol{x}} f_i(\boldsymbol{x})
$$

$$
\Rightarrow \mathbb{E}_{i\sim\mathcal{C}}[\tilde{\boldsymbol{g}}_t] \rightsquigarrow (\boldsymbol{x}_t - \mathbb{E}_{i\sim\mathcal{C}}[\boldsymbol{x}_i^\star]).
$$

Further, the server momentum is based on $\tilde{\boldsymbol{g}}_t$ and hence is also biased. Thus, it cannot correct for the client drift. We next see how a different way of using momentum can mitigate client drift.

**Mime approach.** FEDAVG experiences client drift because both the momentum and the client updates are biased. To fix the former, we compute momentum using only global optimizer state as in (2) using the sampled client $i \sim \mathcal{C}$:

$$\boldsymbol{m}_t = (1 - \beta)\nabla f_i(\boldsymbol{x}_{t-1}) + \beta\boldsymbol{m}_{t-1} \,. \tag{4}$$

To reduce the bias in the local updates, we will apply this unbiased momentum every step $k \in [K]$:

$$\boldsymbol{y}_k = \boldsymbol{y}_{k-1} - \eta((1 - \beta)\nabla f_i(\boldsymbol{y}_{k-1}) + \beta\boldsymbol{m}_{t-1}) \,. \tag{5}$$

Note that the momentum term is kept fixed during the local updates i.e. there is no local momentum used, only global momentum is applied locally. Since $\boldsymbol{m}_{t-1}$ is a moving average of unbiased gradients computed over multiple clients, it intuitively is a good approximation of the general direction of the updates. By taking a convex combination of the local gradient with $\boldsymbol{m}_{t-1}$, the update (5) is potentially also less biased. In this way MIME combines the communication benefits of taking multiple local steps and prevents client-drift (see Fig 2, right). Appendix B makes this intuition precise.

# B Proof sketch

In this section, we provide an intuition behind our proof of convergence of MimeMVR. There are three main components: i) how momentum reduces the effect of client drift, ii) how local steps can take advantage of Hessian similarity, and iii) why the SVRG correction improves constants.

**Improving the statistical term via momentum.** Intuitively, using momentum locally at every client update reduces client drift by incorporating information about other clients from past rounds. Assume that we sample a single client $i_t$ in round $t$ and that we use full-batch gradients. Also let the local client update at step $k$ round $t$ be of the form

$$\boldsymbol{y} \leftarrow \boldsymbol{y} - \eta\boldsymbol{d}_k \,. \tag{6}$$

The ideal choice of update is of course $\boldsymbol{d}_k^\star = \nabla f(\boldsymbol{y})$ but however this is unattainable. Instead, MIME with momentum $\beta = 1 - a$ uses $\boldsymbol{d}_k^{\text{SGDm}} = \tilde{\boldsymbol{m}}_k \leftarrow a\nabla f_i(\boldsymbol{y}) + (1 - a)\boldsymbol{m}_{t-1}$ where $\boldsymbol{m}_{t-1}$ is the momentum computed at the server. The variance of this update can then be bounded as

$$\mathbb{E}\|\tilde{\boldsymbol{m}}_k - \nabla f(\boldsymbol{y})\|^2 \lesssim a^2 \, \mathbb{E}\|\nabla f_{i_t}(\boldsymbol{y}) - \nabla f(\boldsymbol{y})\|^2 + (1 - a) \, \mathbb{E}\|\boldsymbol{m}_{t-1} - \nabla f(\boldsymbol{y})\|^2$$
$$\approx a^2 G^2 + (1 - a) \, \mathbb{E}\|\boldsymbol{m}_{t-1} - \nabla f(\boldsymbol{x}_{t-2})\|^2 \approx aG^2 \,.$$

The last step follows by unrolling the recursion on the variance of $\boldsymbol{m}$. We also assumed that $\eta$ is small enough that $\boldsymbol{y} \approx \boldsymbol{x}_{t-2}$. This way, momentum can reduce the variance of the update from $G^2$ to $(aG^2)$ by using past gradients computed on different clients. Of course, this also introduces additional bias into the update. To reduce this bias requires slightly modifying the momentum algorithm similar to [14]. The full analysis is carried out in Appendix H.

**Improving the optimization term via local steps.** The optimization (second) term in Theorem IV is $\frac{\delta K + L}{\epsilon K}$. In contrast, the optimization term of the server-only methods is $L/\epsilon$. Since in most cases $\delta \ll L$, the former can be significantly smaller than the latter. This rate also suggests that the best choice of number of local updates is $L/\delta$ i.e. we should perform more client updates when they have more similar Hessians. This generalizes results of [32] from quadratics to all functions.

This improvement is due to a careful analysis of the *bias* in the gradients computed during the local update steps. Note that for client parameters $\boldsymbol{y}_{k-1}$, the gradient $\mathbb{E}[\nabla f_{i_t}(\boldsymbol{y}_{k-1})] \neq \mathbb{E}[\nabla f(\boldsymbol{y}_{k-1})]$ since $\boldsymbol{y}_{k-1}$ was also computed using the same loss function $f_{i_t}$. In fact, only the first gradient computed at $\boldsymbol{x}_{t-1}$ is unbiased. Dropping the subscripts $k$ and $t$, we can bound this bias as:

$$\mathbb{E}[\nabla f_i(\boldsymbol{y}) - \nabla f(\boldsymbol{y})] = \mathbb{E}[\underbrace{\nabla f_i(\boldsymbol{y}) - \nabla f_i(\boldsymbol{x})}_{\approx \nabla^2 f_i(\boldsymbol{x})(\boldsymbol{y} - \boldsymbol{x})} + \underbrace{\nabla f(\boldsymbol{x}) - \nabla f(\boldsymbol{y}_i)}_{\approx \nabla^2 f(\boldsymbol{x})(\boldsymbol{x} - \boldsymbol{y}_i)}] + \underbrace{\mathbb{E}_i[\nabla f_i(\boldsymbol{x})] - \nabla f(\boldsymbol{x})}_{=0 \text{ since unbiased}}$$
$$\approx \mathbb{E}[(\nabla^2 f_i(\boldsymbol{x}) - \nabla^2 f(\boldsymbol{x}))(\boldsymbol{y}_i - \boldsymbol{x})] \approx \delta \, \mathbb{E}[(\boldsymbol{y}_i - \boldsymbol{x})] \,.$$

Thus, the Hessian dissimilarity (A2) control the bias, and hence the usefulness of local updates. This intuition can be made formal using Lemma 3. Note that this improved analysis is potentially applicable to any local update methods and is not specific to Mime.

**Mini-batches via SVRG correction.** In our previous discussion about momentum and local steps, we assumed that the clients compute full batch gradients and that only one client is sampled per round. However, in practice a large number ($S$) of clients are sampled and further the clients use mini-batch gradients. The SVRG correction reduces this within-client variance since

$$\text{Var}\Big(\nabla f_i(\boldsymbol{y}_i; \zeta) - \nabla f_i(\boldsymbol{x}; \zeta) + \tfrac{1}{|\mathcal{S}|} \textstyle\sum_{i \in \mathcal{S}} \nabla f_i(\boldsymbol{x})\Big) \lesssim L^2 \|\boldsymbol{y}_i - \boldsymbol{x}\|^2 + \frac{G^2}{S} \approx \frac{G^2}{S}\,.$$

Here, we used the smoothness of $f_i(\cdot; \zeta)$ and assumed that $\boldsymbol{y}_i \approx \boldsymbol{x}$ since we don't move too far within a single round. Thus, the SVRG correction allows us to use minibatch gradients in the local updates while still ensuring that the variance is of the order $G^2/S$. In practical deep learning, this SVRG correction may not very effective [15] and so can be dropped, though it is useful to derive the optimal theoretical rates.

## C Experimental setup

### C.1 Description of ablation study

We train a 2 hidden layer MLP with 300u-100 neurons on the EMNIST62 (extended MNIST) dataset [12]. The clients' data is separated according to the original authors of the characters [10]. All methods are augmented with momentum–Mime and MimeLite use momentum in the client updates, and the others use server momentum. The momentum parameter is searched over $\beta \in [0, 0.9, 0.99]$. For Adam, we fix $\beta_1 = 0.9$, $\beta_2 = 0.99$, and $\epsilon = 10^{-3}$. For both FedProx and SCAFFOLD, $\beta = 0$ (no server momentum) yielded the best performance. For FedAvg, Mime, and MimeLite $\beta = 0.9$ was the fastest. For FedProx, the regularization parameter $\mu$ was searched over $[0.1, 0.5, 1]$ and $\mu = 0.1$ had highest test accuracy.

### C.2 Description of large scale experiments

We perform 4 tasks over 3 datasets: i) On the EMNIST62 dataset [12] we run a convex multi-class (62 classes) logistic regression model, and ii) a convolution model with two CNN layers and two dense layers and dropout. iii) On the SHAKESPEARE dataset, we train a single layer LSTM model with state size of 256 and embedding size of 8 to predict the next character [43]. iv) Finally, on the STACKOVERFLOW dataset [16], we train a next word prediction language model with embedding size of 96, a LSTM layer of size 670, and a vocabulary size of 1000. In all cases we report the top-1 test accuracy in our experiments.

All datasets use the metadata indicating the original authors to separate them into multiple clients yielding naturally partitioned datasets. Table 3 summarizes the statistics about the different datasets. Note that the average number of rounds a client participates in (computed as sampled clients×number of rounds/number of clients) provides an indication of how much of the training data is seen with SHAKESPEARE being closest to the cross-silo setting and STACKOVERFLOW representing the most cross-device in nature.

Table 3: Details about the datasets used and experiment setting.

|  | EMNIST62 | SHAKESPEARE | STACKOVERFLOW |
|---|---|---|---|
| Clients | 3,400 | 715 | 342,477 |
| Examples | 671,585 | 16,068 | 135,818,730 |
| Batch size | 10 | 10 | 10 |
| Number of local epochs | 1 | 1 | 1 |
| Total number of rounds | 1000 | 1000 | 1000 |
| Avg. rounds each client participates | 5.9 | 28 | 0.15 |

We use Tensorflow federated datasets [60] to generate the datasets. Our federated learning simulation code is written in FedJAX [52, 53] and is open-sourced at github.cm/google/fedjax (see documentation). Black and white was reversed in EMNIST62 (i.e. subtracted from 1) to make them similar to MNIST. The preprocessing for SHAKESPEARE and STACKOVERFLOW datasets exactly matches that of [49].

Table 4: Effective number of sampled clients.

|  | Total Comm. | EMNIST62 | SHAKESPEARE | STACKOVERFLOW |
|---|---|---|---|---|
| FedAvg | 2× | 20 | 20 | 50 |
| MimeLiteMom | 5× | 8 | 8 | 20 |
| MimeLiteAdagrad | 5× | 8 | 8 | 20 |
| MimeLiteAdam | 6× | 6 | 6 | 16 |
| MimeMom | 6× | 6 | 6 | 16 |
| MimeAdagrad | 6× | 6 | 6 | 16 |
| MimeAdam | 7× | 5 | 5 | 14 |

## C.3 Practicality of experiments

In the experiments we only cared about the number of communication rounds, ignoring that MIME actually needs twice the number of bits per round and that the SERVER-ONLY methods have a much smaller computational requirement. This is standard in the federated learning setting as introduced by [43] and is justified because most of the time in cross-device FL is spent in establishing connections with devices rather than performing useful work such as communication or computation. In other words, *latency* and not bandwidth or computation are critical in cross device FL. However, one can certainly envision cases where this is not true. Incorporating communication compression strategies [58, 3, 33, 64] or client-model compression strategies [9, 19, 22] into our MIME framework can potentially address such issues and are important future research directions.

Regarding the algorithms evaluated, we chose not to include MVR as a base optimizer. This is because it is not a popular choice is practice even in the centralized setting, and serves more as a theoretical stand in to explain the benefit of the simpler SGD with momentum algorithm. Hence, we wouldn't expect MimeMVR to perform better than MimeMom. In general, our goal was to "mimic" centralized methods – methods which have better empirical performance (momentum and Adam) we showed also perform well in the federated setting when combined with Mime, and similarly methods which have better theoretical rates (MVR) have good rates with Mime as well.

Further, as we noted previously, we believe both the datasets and the tasks being studied here are close to real world settings since they contain natural heterogeneity. We now discuss our choice of other parameters in the experiment setup (number of training rounds, sampled clients, batch-size, etc.) Each round of federated learning takes 3 mins in the real world and is relatively independent of the size of communication [7] implying that training 1000 rounds takes **2 days** even for small models. In contrast, running a centralized simulation takes about 15 mins. This underscores the importance of ensuring that the algorithms for federated learning converge in as few rounds as possible, as well as have very easy to set default hyper-parameters. Thus, in our experimental setup we keep all parameters other than the learning rate to their default values. In practice, this learning rate can be set by set using a small centralized dataset on the server (as in [23]). Thus, it is crucial for federated frameworks to be able to translate algorithms which work well in centralized settings directly to the federated setting without additial hyper-parameter tuning. The choice of batch size being 10 was made both keeping in mind the limited memory available to each client as well as to match prior work. Finally, while we limit ourselves to sampling 20–50 workers per round due to computational constraints, in real world FL thousands of devices are often available for training simultaneously each round [7]. They also note that the probability of each of these devices being available has clear patterns and is far from uniform sampling. Conducting a large scale experimental study which mimics these alternate forms of heterogeneity is an important direction for future work.

## C.4 Hyperparameter search

We run two hyper-parameter sweeps in our experiments: first a *light* setup which is reported in the main paper, and one we believe reflects the real world performance, and second a *heavy* tuning setting to showcase the performance of the methods as we vary the hyper-parameters.

**Light-sweep setting (9×).** For all Momentum methods, we pick momentum $\beta = 0.9$. For Adam methods, we fix $\beta_1 = 0.9$ and $\beta_2 = 0.99$, and $\varepsilon_0 = 1 \times 10^{-7}$. For Adagrad we use the default initialization value of 0.1 and use $\varepsilon_0 = 1 \times 10^{-7}$. None of the algorithms use weight decay, clipping

etc. The learning rate is then tuned to obtain the best test accuracy. For all experiments, unless explicitly mentioned otherwise, the learning rate is searched over a grid ($9\times$):

$$\eta \in \left[1, 1 \times 10^{-0.5}, 1 \times 10^{-1}, 1 \times 10^{-1.5}, 1 \times 10^{-2}, 1 \times 10^{-2.5}, 1 \times 10^{-3}, 1 \times 10^{-3.5}, 1 \times 10^{-4}\right].$$

The server learning rate for all methods is kept at its default value of $1$.

**Heavy-sweep setting (567$\times$).** For all Momentum methods, we pick momentum $\beta = 0.9$. For Adam methods, we fix $\beta_1 = 0.9$ and $\beta_2 = 0.99$. For Adagrad we use the default initialization value of $0.1$. None of the algorithms use weight decay, clipping etc. The learning rate is then tuned to obtain the best test accuracy.

For all experiments, unless explicitly mentioned otherwise, the **client** learning rate is searched over a grid ($9\times$):

$$\eta_{\text{client}} \in \left[1, 1 \times 10^{-0.5}, 1 \times 10^{-1}, 1 \times 10^{-1.5}, 1 \times 10^{-2}, 1 \times 10^{-2.5}, 1 \times 10^{-3}, 1 \times 10^{-3.5}, 1 \times 10^{-4}\right].$$

Further, we also search for the **server** learning rate is searched over a grid ($9\times$):

$$\eta_{\text{server}} \in \left[1 \times 10^{1}, 1 \times 10^{0.5}, 1, 1 \times 10^{-0.5}, 1 \times 10^{-1}, 1 \times 10^{-1.5}, 1 \times 10^{-2}, 1 \times 10^{-2.5}, 1 \times 10^{-3}\right].$$

Finally, for the **adaptive** methods such as Adam and Adagrad, we also tune the $\varepsilon_0$ parameter over a grid ($7\times$):

$$\varepsilon_0 \in \left[1, 1 \times 10^{-1}, 1 \times 10^{-2}, 1 \times 10^{-3}, 1 \times 10^{-4}, 1 \times 10^{-5}, 1 \times 10^{-6}, 1 \times 10^{-7}\right].$$

### C.5 Comparison with previous results

As far as we are aware, [49] is the only prior work which conducts a systematic experimental study of federated learning algorithms over multiple realistic datasets. The algorithms comparable across the two works (e.g. FedAvgSGD, FedAvgMom, and FedAvgAdam) have qualitatively similar performance except with one exception: FedAvgAdam consistently underperforms FedAvgMom. This difference, as we show later, is because FedAvgAdam does not work with the default choices of hyper-parameters such as $\epsilon$ and requires additional tuning. As we explain in Section C.3, we chose to keep these parameters to the default values of their centralized counterparts to compare methods in a 'low-tuning' setting. We also point that while FedAvgAdam struggles to perform in this setup, MimeAdam and MimeLiteAdam are very stable and even often outperform their SGD counterparts.

### C.6 Additional algorithmic details

Table 5: Decomposing base algorithms into a parameter update ($\mathcal{U}$) and statistics tracking ($\mathcal{V}$).

| Algorithm | Tracked statistics $s$ | Update step $\mathcal{U}$ | Tracking step $\mathcal{V}$ |
|---|---|---|---|
| SGD | – | $\boldsymbol{x} - \eta \boldsymbol{g}$ | – |
| SGDm/Mom | $\boldsymbol{m}$ | $\boldsymbol{x} - \eta((1-\beta)\boldsymbol{g} + \beta\boldsymbol{m})$ | $\boldsymbol{m} = (1-\beta)\boldsymbol{g} + \beta\boldsymbol{m}$ |
| AdaGrad | $\boldsymbol{v}$ | $\boldsymbol{x} - \frac{\eta}{\epsilon + \sqrt{\boldsymbol{v}}} \boldsymbol{g}$ | $\boldsymbol{v} = \boldsymbol{g}^2 + \boldsymbol{v}$ |
| Adam | $\boldsymbol{m}, \boldsymbol{v}$ | $\boldsymbol{x} - \frac{\eta}{\epsilon + \sqrt{\boldsymbol{v}}}((1-\beta_1)\boldsymbol{g} + \beta_1\boldsymbol{m})$ | $\boldsymbol{m} = (1-\beta_1)\boldsymbol{g} + \beta_1\boldsymbol{m}$ 
 $\boldsymbol{v} = (1-\beta_2)\boldsymbol{g}^2 + \beta_2\boldsymbol{v}$ |

## D  Additional Adam experiments

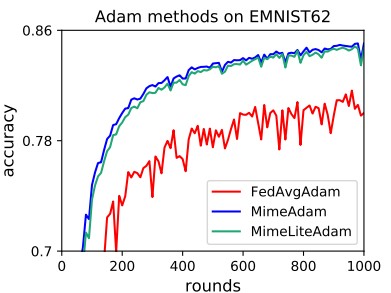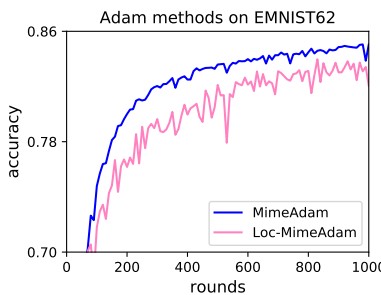

Figure 3: **Mime**, **MimeLite**, **FedAvg**, and **Loc-Mime** with Adam using 10 local epochs, run on EMNIST62 and a 2 hidden layer (300u-100) MLP. (Left) Mime and MimeLite are nearly identical and outperform FedAvg. (Right) Locally adapting Adam state slows down convergence and makes it more unstable. Both these results are consistent with the earlier momentum results.

## E  Stability of methods to hyper-parameters

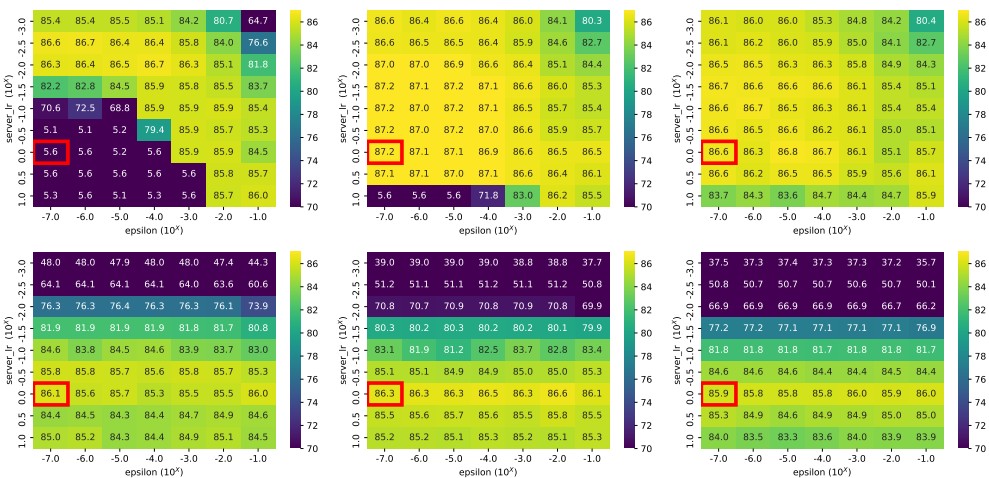

Figure 4: Stability of adaptive methods with varying server learning: FedAvg (left), Mime (middle) and MimeLite (right) with Adam (top) and Adagrad (bottom) as base algorithms are run on EMNIST62 with CNN. For each value of server learning rate ($y$-axis) and $\varepsilon_0$ ($x$-axis), the client learning rate was tuned over the $9\times$ grid and the accuracy reported. The red box highlights the default configuration in a centralized setting. We see that FedAvgAdam is very sensitive to the server learning rate and $\varepsilon_0$, performing poorly in the default centralized parameter regimes. Mime and MimeLite acheive their best performance with the centralized parameters. This justifies our claim that Mime and MimeLite can **adapt** any centralized method with the same hyper-parameters and only require tuning of a single learning rate. This, we believe, is crucial for real world deployment.

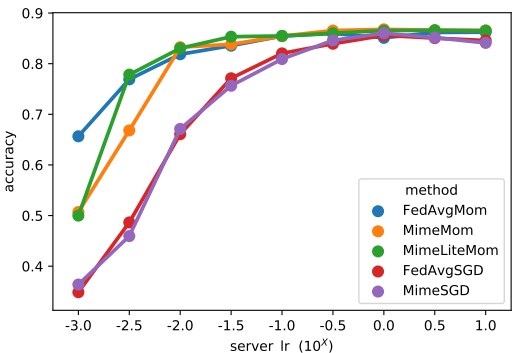

Figure 5: Stability of non-adaptive methods with varying server learning: FedAvg, Mime and MimeLite with SGD and momentum ($\beta = 0.9$) as base algorithms are run on EMNIST62 with CNN. For each value of server learning rate, the client learning rate was tuned over the $9\times$ grid. The momentum methods are more insensitive to the server learning rate than the SGD methods. Server learning rate of 1 (default value) seems to work well for all methods.

## F    Technicalities

We examine some additional definitions and introduce some technical lemmas.

### F.1    Assumptions and definitions

We make precise a few definitions and explain some of their implications. We first discuss the two assumptions on the dissimilarity between the gradients (A1) and the Hessians (A2). Loosely, these two quantities are an extension of the concepts of **variance** and **smoothness** which occur in centralized SGD analysis to the federated learning setting. Just as the variance and smoothness are completely orthogonal concepts, we can have settings where $G^2$ (gradient dissimilarity) is large while $\delta$ (Hessian dissimilarity) is small, or vice-versa.

Our assumption about the bound on the $G$ gradient dissimilarity can easily be extended to $(G, B)$ gradient dissimilarity used by [33]:

$$\mathbb{E}_i \|\nabla f_i(\boldsymbol{x})\|^2 \leq G^2 + B^2 \|\nabla f(\boldsymbol{x})\|^2 . \tag{7}$$

All the proofs in the paper extend in a straightforward manner to the above weaker notion. Since this notion does not present any novel technical challenge, we omit it in the rest of the proofs. Note however that the above weaker notion can potentially capture the fact that by increasing the model capacity, we can reduce $G$. In the extreme case, by taking a sufficiently over-parameterized model, it is possible to make $G = 0$ in certain settings [63]. However, this comes both at a cost of increased resource requirements (i.e. higher memory and compute requirements per step) but can also result in other constants increasing (e.g. $B$ and $L$).

The second crucial definition we use in this work is that of $\delta$ bounded *Hessian* dissimilarity (A2). This has been used previously in the analyses of distributed [54, 6, 51] and federated learning [32], but has been restricted to quadratics. Here, we show how to extend both the notion as well as the analysis to general smooth functions. The main manner we will use this assumption is in Lemma 3 to claim that for any $\boldsymbol{x}$ and $\boldsymbol{y}$ the following holds:

$$\mathbb{E}\|\nabla f_i(\boldsymbol{y}; \zeta) - \nabla f_i(\boldsymbol{x}; \zeta) + \nabla f(\boldsymbol{x}) - \nabla f(\boldsymbol{y})\|^2 \leq \delta^2 \|\boldsymbol{y} - \boldsymbol{x}\|^2 . \tag{8}$$

Here the expectation is over the choice of client $i$. To understand what the above condition means, it is illuminating to define $\Psi_i(\boldsymbol{z}) = f_i(\boldsymbol{z}; \zeta) - f(\boldsymbol{z})$. Then, we can rewrite (A2) and (8) respectively as

$$\|\nabla^2 \Psi_i(\boldsymbol{z})\| \leq \delta \quad \text{and} \quad \mathbb{E}\|\nabla \Psi_i(\boldsymbol{y}) - \nabla \Psi_i(\boldsymbol{x})\|^2 \leq \delta^2 \|\boldsymbol{y} - \boldsymbol{x}\|^2 .$$

Thus (8) and (A2) are both different notions of smoothness of $\Psi_i(\boldsymbol{x})$ (formal definition of smoothness will follow soon). The latter definition closely matches the notion of *squared-smoothness* used by [5] and is a promising relaxation of (A2). However, we run into some technical issues since in our case the variable $\boldsymbol{y}$ can also be a random variable and depend on the choice of the client $i$. Extending

our results to this weaker notion of Hessian-similarity and proving tight non-convex lower bounds is an exciting theoretical challenge.

Finally note that if the functions $f_i(\boldsymbol{x}; \zeta)$ are assumed to be smooth as in [54, 6, 32], then $\Psi_i((\boldsymbol{x})$ is $2L$-smooth. Thus, we *always* have that $\delta \le 2L$. But, as shown in [54], it is possible to have $\delta \ll L$ if the data distribution amongst the clients is similar. Further, the lower bound from [6] proves that Hessian-similarity is the crucial quantity capturing the number of rounds of communication required for distributed/federated optimization.

We next define the terms smoothness and strong-convexity which we repeatedly use in the paper.

(**A2\***) $f_i$ is almost surely **L-smooth** and satisfies:

$$\|\nabla f_i(\boldsymbol{x}; \zeta) - \nabla f_i(\boldsymbol{y}; \zeta)\| \le L\|\boldsymbol{x} - \boldsymbol{y}\|, \text{ for any } \boldsymbol{x}, \boldsymbol{y}. \tag{9}$$

The assumption (A2\*) also implies the following quadratic upper bound on $f_i$

$$f_i(\boldsymbol{y}) \le f_i(\boldsymbol{x}) + \langle \nabla f_i(\boldsymbol{x}), \boldsymbol{y} - \boldsymbol{x} \rangle + \frac{L}{2}\|\boldsymbol{y} - \boldsymbol{x}\|^2. \tag{10}$$

Further, if $f_i$ is twice-differentiable, (A2\*) implies that $\|\nabla^2 f_i(\boldsymbol{x}; \zeta)\| \le \beta$ for any $\boldsymbol{x}$.

(**A3**) We assume that the **intra-client gradient variance** is bounded by $\sigma^2$. For any client $i$, the following holds almost surely at any fixed $\boldsymbol{x}$:

$$\mathbb{E}_{\zeta_i}[\nabla f_i(\boldsymbol{x}; \zeta)] = \nabla f_i(\boldsymbol{x}), \quad \text{and} \quad \mathbb{E}_{\zeta_i}\|\nabla f_i(\boldsymbol{x}; \zeta) - \nabla f_i(\boldsymbol{x})\|^2 \le \sigma^2.$$

Note that we expect the intra-client variance to be smaller than inter-client variance and so typically $\sigma^2 \le G^2$.

(**A4**) $f$ satisfies the $\mu$-**PL inequality** [29] for $\mu > 0$ if:

$$\|\nabla f(\boldsymbol{x})\|^2 \ge 2\mu(f(\boldsymbol{x}) - f^\star).$$

Note that PL-inequality is much weaker than the standard notion of strong-convexity, and in fact is even satisfied by some non-convex functions [29].

## F.2 Some technical lemmas

Now we cover some technical lemmas which are useful for computations later on. First, we state a relaxed triangle inequality true for the squared $\ell_2$ norm.

**Lemma 1** (relaxed triangle inequality). *Let $\{\boldsymbol{v}_1, \dots, \boldsymbol{v}_\tau\}$ be $\tau$ vectors in $\mathbb{R}^d$. Then the following are true:*

1. *$\|\boldsymbol{v}_i + \boldsymbol{v}_j\|^2 \le (1 + c)\|\boldsymbol{v}_i\|^2 + (1 + \frac{1}{c})\|\boldsymbol{v}_j\|^2$ for any $c > 0$, and*

2. *$\|\sum_{i=1}^\tau \boldsymbol{v}_i\|^2 \le \tau \sum_{i=1}^\tau \|\boldsymbol{v}_i\|^2$.*

*Proof.* The proof of the first statement for any $c > 0$ follows from the identity:

$$\|\boldsymbol{v}_i + \boldsymbol{v}_j\|^2 = (1 + c)\|\boldsymbol{v}_i\|^2 + (1 + \tfrac{1}{c})\|\boldsymbol{v}_j\|^2 - \|\sqrt{c}\boldsymbol{v}_i + \tfrac{1}{\sqrt{c}}\boldsymbol{v}_j\|^2.$$

For the second inequality, we use the convexity of $\boldsymbol{x} \to \|\boldsymbol{x}\|^2$ and Jensen's inequality

$$\left\|\frac{1}{\tau}\sum_{i=1}^\tau \boldsymbol{v}_i\right\|^2 \le \frac{1}{\tau}\sum_{i=1}^\tau \|\boldsymbol{v}_i\|^2. \qquad \square$$

Next we state an elementary lemma about expectations of norms of random vectors.

**Lemma 2** (separating mean and variance). *Let $\{\Xi_1, \dots, \Xi_\tau\}$ be $\tau$ random variables in $\mathbb{R}^d$ which are not necessarily independent. First suppose that their mean is $\mathbb{E}[\Xi_i] = \xi_i$ and variance is bounded as $\mathbb{E}[\|\Xi_i - \xi_i\|^2] \le \sigma^2$. Then, the following holds*

$$\mathbb{E}[\|\sum_{i=1}^\tau \Xi_i\|^2] \le \|\sum_{i=1}^\tau \xi_i\|^2 + \tau^2\sigma^2.$$

*Now instead suppose that their* conditional mean *is* $\mathbb{E}[\Xi_i|\Xi_{i-1}, \ldots \Xi_1] = \xi_i$ *i.e. the variables* $\{\Xi_i - \xi_i\}$ *form a martingale difference sequence, and the variance is bounded by* $\mathbb{E}[\|\Xi_i - \xi_i\|^2] \leq \sigma^2$ *as before. Then we can show the tighter bound*

$$\mathbb{E}[\|\sum_{i=1}^{\tau} \Xi_i\|^2] \leq 2\|\sum_{i=1}^{\tau} \xi_i\|^2 + 2\tau\sigma^2 \,.$$

*Proof.* For any random variable $X$, $\mathbb{E}[X^2] = \mathbb{E}[(X - \mathbb{E}[X])^2] + (\mathbb{E}[X])^2$ implying

$$\mathbb{E}[\|\sum_{i=1}^{\tau} \Xi_i\|^2] = \|\sum_{i=1}^{\tau} \xi_i\|^2 + \mathbb{E}[\|\sum_{i=1}^{\tau} \Xi_i - \xi_i\|^2] \,.$$

Expanding the above expression using relaxed triangle inequality (Lemma 1) proves the first claim:

$$\mathbb{E}[\|\sum_{i=1}^{\tau} \Xi_i - \xi_i\|^2] \leq \tau \sum_{i=1}^{\tau} \mathbb{E}[\|\Xi_i - \xi_i\|^2] \leq \tau^2\sigma^2 \,.$$

For the second statement, $\xi_i$ is not deterministic and depends on $\Xi_{i-1}, \ldots, \Xi_1$. Hence we have to resort to the cruder relaxed triangle inequality to claim

$$\mathbb{E}[\|\sum_{i=1}^{\tau} \Xi_i\|^2] \leq 2\|\sum_{i=1}^{\tau} \xi_i\|^2 + 2\,\mathbb{E}[\|\sum_{i=1}^{\tau} \Xi_i - \xi_i\|^2]$$

and then use the tighter expansion of the second term:

$$\mathbb{E}[\|\sum_{i=1}^{\tau} \Xi_i - \xi_i\|^2] = \sum_{i,j} \mathbb{E}[(\Xi_i - \xi_i)^\top (\Xi_j - \xi_j)] = \sum_i \mathbb{E}[\|\Xi_i - \xi_i\|^2] \leq \tau\sigma^2 \,.$$

The cross terms in the above expression have zero mean since $\{\Xi_i - \xi_i\}$ form a martingale difference sequence. $\qquad\square$

### F.3 Properties of functions with bounded Hessian dissimilarity

We now study two lemmas which hold for any functions which satisfy (A2) and (A3). The first is closely related to the notion of smoothness (A2*).

**Lemma 3** (similarity)**.** *The following holds for any two functions* $f_i(\cdot)$ *and* $f(\cdot)$ *satisfying* (A2) *and* (A3)*, and any* $\boldsymbol{x}, \boldsymbol{y}$*:*

$$\|\nabla f_i(\boldsymbol{y}; \zeta) - \nabla f_i(\boldsymbol{x}; \zeta) + \nabla f(\boldsymbol{x}) - \nabla f(\boldsymbol{y})\|^2 \leq \delta^2 \|\boldsymbol{y} - \boldsymbol{x}\|^2 \,.$$

*Proof.* Consider the function $\Psi(\boldsymbol{z}) := f_i(\boldsymbol{z}; \zeta) - f(\boldsymbol{z})$. By the assumption (A2), we know that $\|\nabla^2 \Psi(\boldsymbol{z})\| \leq \delta$ for all $\boldsymbol{z}$ i.e. $\Psi$ is $\delta$-smooth. By standard arguments based on taking limits [45], this implies that

$$\|\nabla \Psi(\boldsymbol{y}) - \nabla \Psi(\boldsymbol{x})\| \leq \delta \|\boldsymbol{y} - \boldsymbol{x}\| \,.$$

Plugging back the definition of $\Psi$ into the above inequality proves the lemma. $\qquad\square$

Next, we see how weakly-convex functions satisfy a weaker notion of "averaging does not hurt". This is used to get a handle on the effect of averaging of parameters in FedAvg.

**Lemma 4** (averaging)**.** *Suppose* $f$ *is* $\delta$-*weakly convex. Then, for any* $\gamma \geq \delta$*, and a sequence of parameters* $\{\boldsymbol{y}_i\}_{i \in \mathcal{S}}$ *and* $\boldsymbol{x}$*:*

$$\frac{1}{|\mathcal{S}|} \sum_{i \in \mathcal{S}} f(\boldsymbol{y}_i) + \frac{\gamma}{2} \|\boldsymbol{x} - \boldsymbol{y}_i\|^2 \geq f(\bar{\boldsymbol{y}}) + \frac{\gamma}{2} \|\boldsymbol{x} - \bar{\boldsymbol{y}}\|^2 \,, \text{ where } \bar{\boldsymbol{y}} := \frac{1}{|\mathcal{S}|} \sum_{i \in \mathcal{S}} \boldsymbol{y}_i \,.$$

*Proof.* Since $f$ is $\delta$-weakly convex, $\Phi(\boldsymbol{z}) := f(\boldsymbol{z}) + \frac{\gamma}{2} \|\boldsymbol{z} - \boldsymbol{x}\|^2$ is convex. This proves the claim since $\frac{1}{|\mathcal{S}|} \sum_{i \in \mathcal{S}} \Phi(\boldsymbol{y}_i) \geq \Phi(\bar{\boldsymbol{y}})$ by convexity. $\qquad\square$

# G  Convergence with a generic base optimizer

Let us rewrite the Mime and MimeLite updates using notation convenient for analysis. In each round $t$, we sample clients $\mathcal{S}^t$ such that $|\mathcal{S}^t| = S$. The server communicates the server parameters $\boldsymbol{x}^{t-1}$ as well as the average gradient across the sampled clients $\boldsymbol{c}^t$ defined as

$$\boldsymbol{c}^t = \frac{1}{S} \sum_{i \in \mathcal{S}^t} \nabla f_i(\boldsymbol{x}^{t-1}) \,. \tag{11}$$

Note that computing $\boldsymbol{c}^t$ (required only by Mime but not by MimeLite) itself requires additional communication. In this proof, we do not make any assumption on how $\boldsymbol{c}^t$ is computed as long as it is unbiased and is computed over $S$ clients. In particular, it can either be computed on the sampled $\mathcal{S}^t$ or a different set of an independent sampled clients $\tilde{\mathcal{S}}^t$.

Then each client $i \in \mathcal{S}^t$ makes a copy $\boldsymbol{y}_{i,0}^t = \boldsymbol{x}^{t-1}$ and perform $K$ local client updates. In each local client update $k \in [K]$, the client samples a dataset $\zeta_{i,k}^t$ and

$$\boldsymbol{y}_{i,k}^t = \boldsymbol{y}_{i,k-1}^t - \eta \mathcal{U}(\nabla f_i(\boldsymbol{y}_{i,k-1}^t; \zeta_{i,k}^t) - \nabla f_i(\boldsymbol{x}^{t-1}; \zeta_{i,k}^t) + \boldsymbol{c}^t; \boldsymbol{s}^{t-1}) \qquad \text{(Mime client update)}$$

$$= \boldsymbol{y}_{i,k-1}^t - \eta \mathcal{U}(\nabla f_i(\boldsymbol{y}_{i,k-1}^t; \zeta_{i,k}^t); \boldsymbol{s}^{t-1}) \,. \qquad \text{(MimeLite client update)}$$

After $K$ such local updates, the server then aggregates the new client parameters as

$$\boldsymbol{x}^t = \frac{1}{S} \sum_{i \in \mathcal{S}^t} \boldsymbol{y}_{i,K}^t \qquad \text{(Update server parameters)}$$

$$\boldsymbol{s}^t = \mathcal{V}(\boldsymbol{c}^t, \boldsymbol{s}^{t-1}) \,. \qquad \text{(Update server statistics)}$$

## G.1  Proof of Theorem I (generic reduction)

**Computing server update.**

**Lemma 5** (Deviation from central update.). *For a linear updater $\mathcal{U}$ the server update for Mime can be written as*

$$\boldsymbol{x}^t = \boldsymbol{x}^{t-1} - \tilde{\eta} \mathcal{U}\left( \frac{1}{S} \sum_i \nabla f_i(\boldsymbol{x}) + \boxed{\boldsymbol{e}^t} ; \boldsymbol{s}^{t-1} \right),$$

*and for MimeLite is becomes*

$$\boldsymbol{x}^t = \boldsymbol{x}^{t-1} - \tilde{\eta} \mathcal{U}\left( \frac{1}{KS} \sum_{i,k} \nabla f_i(\boldsymbol{x}; \zeta_{i,k}) + \boxed{\boldsymbol{e}^t} ; \boldsymbol{s}^{t-1} \right),$$

*for $\tilde{\eta} := K\eta$. The error is defined as $\boxed{\boldsymbol{e}^t} = \frac{1}{KS} \sum_{i,k} (\nabla f_i(\boldsymbol{y}_{i,k-1}; \zeta_{i,k}) - \nabla f_i(\boldsymbol{x}; \zeta_{i,k}))$*

*Proof.* Because the updater $\mathcal{U}$ is linear in its first parameter, we can rewrite the update to the server for MimeLite as

$$\boldsymbol{x}^t - \boldsymbol{x}^{t-1} = \frac{1}{S} \sum_{i \in \mathcal{S}^t} \sum_{k=1}^K -\eta \mathcal{U}(\nabla f_i(\boldsymbol{y}_{i,k-1}^t; \zeta_{i,k}^t); \boldsymbol{s}^{t-1})$$

$$= \eta K \mathcal{U}\left( \frac{1}{KS} \sum_{i,k} \nabla f_i(\boldsymbol{y}_{i,k-1}^t; \zeta_{i,k}^t); \boldsymbol{s}^{t-1} \right)$$

We drop the dependence on $t$ when obvious from context and $i$ by default sums over $\mathcal{S}^t$ and $k$ over $[K]$ by default. Using our definition of $\boldsymbol{e}^t$ we have

$$\boldsymbol{x}^t - \boldsymbol{x}^{t-1} = \eta K \mathcal{U}\left( \frac{1}{KS} \sum_{i,k} \nabla f_i(\boldsymbol{y}_{i,k-1}^t; \zeta_{i,k}^t); \boldsymbol{s}^{t-1} \right)$$

$$= \tilde{\eta} \mathcal{U}\left( \frac{1}{KS} \sum_{i,k} \nabla f_i(\boldsymbol{x}; \zeta_{i,k}) + \boldsymbol{e}^t; \boldsymbol{s}^{t-1} \right).$$

Now let us examine the update of Mime. Again assuming $K$ is a multiple of epoch, we have $\sum_{i,k} \nabla f_i(\boldsymbol{x}; \zeta_{i,k}^t) = K \sum_i \nabla f_i(\boldsymbol{x}) = KS\boldsymbol{x}$. Hence,

$$
\begin{aligned}
\boldsymbol{x}^t - \boldsymbol{x}^{t-1} &= \frac{1}{S} \sum_{i \in \mathcal{S}^t} \sum_{k=1}^{K} -\eta \mathcal{U}(\nabla f_i(\boldsymbol{y}_{i,k-1}; \zeta_{i,k}^t) - \nabla f_i(\boldsymbol{x}; \zeta_{i,k}^t) + \boldsymbol{c}; \boldsymbol{s}^{t-1}) \\
&= \eta K \mathcal{U}(\boldsymbol{c} + \boldsymbol{e}^t; \boldsymbol{s}^{t-1}) \\
&= \eta K \mathcal{U}\left(\frac{1}{S} \sum_i \nabla f_i(\boldsymbol{x}) + \boldsymbol{e}^t; \boldsymbol{s}^{t-1}\right).
\end{aligned}
$$

Thus we showed the lemma for both Mime and MimeLite. $\qquad\square$

**Lemma 6** (Defining error). *For $\boldsymbol{e}^t$ defined in Lemma 5, assuming all functions $f_i(\,\cdot\,, \zeta)$ are L-smooth, we have*

$$
\mathbb{E}\|\boldsymbol{e}^t\|^2 \le L^2 \mathcal{E}_K^t, \quad \text{where } \mathcal{E}_K^t := \frac{1}{KS} \sum_{i,k} \mathbb{E}\|\boldsymbol{y}_{i,k-1} - \boldsymbol{x}\|^2.
$$

*Proof.* Using the smoothness of the individual functions and the definition of $\boldsymbol{e}^t$,

$$
\begin{aligned}
\mathbb{E}\|\boldsymbol{e}^t\|^2 &= \mathbb{E}\|\frac{1}{KS} \sum_{i,k} (\nabla f_i(\boldsymbol{y}_{i,k-1}; \zeta_{i,k}) - \nabla f_i(\boldsymbol{x}; \zeta_{i,k}))\|^2 \\
&\le \frac{1}{KS} \sum_{i,k} \mathbb{E}\|\nabla f_i(\boldsymbol{y}_{i,k-1}) - \nabla f_i(\boldsymbol{x}; \zeta_{i,k})\|^2 \le L^2 \mathcal{E}_K^t.
\end{aligned}
$$

$\qquad\square$

Henceforth, we will call $\mathcal{E}_K^t$ as the error, or as the client-drift following [32].

**Bounding error in MimeLite.** Now we will try bound the client drift $\mathcal{E}^t$ for MimeLite.

**Lemma 7** (MimeLite error). *Suppose that all functions $f_i(\,\cdot\,, \zeta)$ are L-smooth (A2\*), $\sigma^2$ variance (A3), and (A1) is satisfied, and the updater $\mathcal{U}$ has B-Lipschitz updates. Then using step-size $\tilde{\eta} \le \frac{1}{2BL}$,*

$$
\frac{1}{18B^2\tilde{\eta}^2} \mathcal{E}^K \le \mathbb{E}\|\nabla f(\boldsymbol{x})\|^2 + G^2 + \frac{\sigma^2}{2K}.
$$

*Proof.* For $K = 1$, we have $\mathbb{E}\|\boldsymbol{y}_{i,1} - \boldsymbol{x}\|^2 \le B^2\eta^2(G^2 + \sigma^2) + B^2\eta^2 \mathbb{E}\|\nabla f(\boldsymbol{x})\|^2$. The lemma is easily shown to be true. Assuming $K \ge 2$ henceforth, and starting from the client update of MimeLite we have

$$
\begin{aligned}
\mathbb{E}\|\boldsymbol{y}_{i,k} - \boldsymbol{x}\|^2 &= \mathbb{E}\|\boldsymbol{y}_{i,k-1} - \eta \mathcal{U}(\nabla f_i(\boldsymbol{y}_{i,k-1}^t; \zeta_{i,k}^t); \boldsymbol{s}^{t-1}) - \boldsymbol{x}\|^2 \\
&\le \mathbb{E}\|\boldsymbol{y}_{i,k-1} - \eta \mathcal{U}(\nabla f_i(\boldsymbol{y}_{i,k-1}^t; \boldsymbol{s}^{t-1}) - \boldsymbol{x}\|^2 + B^2\eta^2\sigma^2 \\
&\le \left(1 + \frac{1}{K-1}\right) \mathbb{E}\|\boldsymbol{y}_{i,k-1} - \boldsymbol{x}\|^2 + K\eta^2 \mathbb{E}\|\mathcal{U}(\nabla f_i(\boldsymbol{y}_{i,k-1}^t; \boldsymbol{s}^{t-1})\|^2 + B^2\eta^2\sigma^2 \\
&\le \left(1 + \frac{1}{K-1}\right) \mathbb{E}\|\boldsymbol{y}_{i,k-1} - \boldsymbol{x}\|^2 + KB^2\eta^2 \mathbb{E}\|\nabla f_i(\boldsymbol{y}_{i,k-1}) \pm \nabla f_i(\boldsymbol{x})\|^2 + B^2\eta^2\sigma^2 \\
&\le \left(1 + \frac{1}{K-1}\right) \mathbb{E}\|\boldsymbol{y}_{i,k-1} - \boldsymbol{x}\|^2 \\
&\qquad + 2KB^2\eta^2 \mathbb{E}\|\nabla f_i(\boldsymbol{x})\|^2 + 2KB^2L^2\eta^2 \mathbb{E}\|\boldsymbol{y}_{i,k-1} - \boldsymbol{x}\|^2 + B^2\eta^2\sigma^2 \\
&\le \left(1 + \frac{2}{K-1}\right) \mathbb{E}\|\boldsymbol{y}_{i,k-1} - \boldsymbol{x}\|^2 + 2KB^2\eta^2 \mathbb{E}\|\nabla f(\boldsymbol{x})\|^2 + 2KB^2\eta^2 G^2 + B^2\eta^2\sigma^2.
\end{aligned}
$$

Here, we used the condition on our step size that $\tilde{\eta} = K\eta \leq \frac{1}{2LB}$, which implies that $2KB^2L^2\eta^2 \leq \frac{1}{K-1}$. Unrolling this recursion, we have

$$\mathbb{E}\|\boldsymbol{y}_{i,k} - \boldsymbol{x}\|^2 \leq \left(2KB^2\eta^2\,\mathbb{E}\|\nabla f(\boldsymbol{x})\|^2 + 2KB^2\eta^2G^2 + B^2\eta^2\sigma^2\right)\sum_{k=1}^{K}\left(1 + \frac{2}{K-1}\right)^k.$$

Note that $\left(1 + \frac{2}{K-1}\right)^k \leq 9$. Averaging then over $k$ and $i$, we get

$$\mathcal{E}_K^t \leq 18K^2B^2\eta^2\,\mathbb{E}\|\nabla f(\boldsymbol{x})\|^2 + 18K^2B^2\eta^2G^2 + 9KB^2\eta^2\sigma^2\,.$$

Finally, recalling that $\tilde{\eta} = K\eta$ finishes the lemma. $\qquad\square$

**Bounding error in Mime.** Next we will try bound the client drift $\mathcal{E}^t$ for Mime. The additional SVRG correction term used in Mime improves the bound on the error.

**Lemma 8** (Mime Error). *Suppose that all functions $f_i(\,\cdot\,, \zeta)$ are $L$-smooth* (A2\*), *$\sigma^2$ variance* (A3), *and* (A1) *is satisfied, and the updater $\mathcal{U}$ has $B$-Lipschitz updates. Then using step-size $\tilde{\eta} \leq \frac{1}{2BL}$,*

$$\mathcal{E}^K \leq 18B^2\tilde{\eta}^2\,\mathbb{E}\left\|\frac{1}{S}\sum_i \nabla_i f(\boldsymbol{x})\right\|^2.$$

*Proof.* For $K = 1$, the Mime update loos like

$$\mathbb{E}\|\boldsymbol{y}_{i,1} - \boldsymbol{x}\|^2 = \eta^2\,\mathbb{E}\|\mathcal{U}(\boldsymbol{c}; \boldsymbol{s}^{t-1})\|^2$$
$$\leq \eta^2B^2\,\mathbb{E}\|\boldsymbol{c}\|^2\,.$$

Assuming $K \geq 2$ henceforth, and starting from the client update of Mime we have

$$\mathbb{E}\|\boldsymbol{y}_{i,k} - \boldsymbol{x}\|^2 = \mathbb{E}\|\boldsymbol{y}_{i,k-1} - \eta\mathcal{U}(\nabla f_i(\boldsymbol{y}_{i,k-1}; \zeta_{i,k}^t) - \nabla f_i(\boldsymbol{x}; \zeta_{i,k}^t) + \boldsymbol{c}^t; \boldsymbol{s}^{t-1}) - \boldsymbol{x}\|^2$$

$$\leq \left(1 + \frac{1}{K-1}\right)\mathbb{E}\|\boldsymbol{y}_{i,k-1} - \boldsymbol{x}\|^2$$
$$\quad + K\eta^2\,\mathbb{E}\|\mathcal{U}(\nabla f_i(\boldsymbol{y}_{i,k-1}; \zeta_{i,k}^t) - \nabla f_i(\boldsymbol{x}; \zeta_{i,k}^t) + \boldsymbol{c}^t; \boldsymbol{s}^{t-1})\|^2$$

$$\leq \left(1 + \frac{1}{K-1}\right)\mathbb{E}\|\boldsymbol{y}_{i,k-1} - \boldsymbol{x}\|^2 + K\eta^2B^2\,\mathbb{E}\|\nabla f_i(\boldsymbol{y}_{i,k-1}; \zeta_{i,k}^t) - \nabla f_i(\boldsymbol{x}; \zeta_{i,k}^t) + \boldsymbol{c}^t\|^2$$

$$\leq \left(1 + \frac{1}{K-1}\right)\mathbb{E}\|\boldsymbol{y}_{i,k-1} - \boldsymbol{x}\|^2$$
$$\quad + 2K\eta^2B^2\,\mathbb{E}\|\nabla f_i(\boldsymbol{y}_{i,k-1}; \zeta_{i,k}^t) - \nabla f_i(\boldsymbol{x}; \zeta_{i,k}^t)\|^2 + 2K\eta^2B^2\,\mathbb{E}\|\boldsymbol{c}^t\|^2$$

$$\leq \left(1 + \frac{1}{K-1} + 2K\eta^2B^2L^2\right)\mathbb{E}\|\boldsymbol{y}_{i,k-1} - \boldsymbol{x}\|^2 + 2K\eta^2B^2\,\mathbb{E}\|\boldsymbol{c}^t\|^2$$

$$\leq \left(1 + \frac{2}{K-1}\right)\mathbb{E}\|\boldsymbol{y}_{i,k-1} - \boldsymbol{x}\|^2 + 2K\eta^2B^2\,\mathbb{E}\|\boldsymbol{c}^t\|^2\,.$$

Here, we used the condition on our step size that $\tilde{\eta} = K\eta \leq \frac{1}{2LB}$, which implies that $2KB^2L^2\eta^2 \leq \frac{1}{K-1}$. Unrolling this recursion, we have

$$\mathbb{E}\|\boldsymbol{y}_{i,k} - \boldsymbol{x}\|^2 \leq 2KB^2\eta^2\,\mathbb{E}\|\boldsymbol{c}^t\|^2\sum_{k=1}^{K}\left(1 + \frac{2}{K-1}\right)^k \leq 18K^2B^2\eta^2\,\mathbb{E}\|\boldsymbol{c}^t\|^2\,.$$

Note that $\left(1 + \frac{2}{K-1}\right)^k \leq 9$. Averaging then over $k$ and $i$, recalling that $\tilde{\eta} = K\eta$ get

$$\mathcal{E}_K^t \leq 18B^2\tilde{\eta}^2\,\mathbb{E}\|\boldsymbol{c}^t\|^2\,.$$

$\qquad\square$

**Putting it together (Theorem I).**

**Lemma 9.** *The updates of Mime and MimeLite for $g^t$ satisfying $\mathbb{E}[g^t] = \nabla f(x^{t-1})$, and we have for $\tilde{\eta} \leq \frac{1}{2BL}$*

$$x^t = x^{t-1} - \tilde{\eta}\mathcal{U}(c^t + e^t; s^{t-1})$$
$$s^t = \mathcal{V}(c^t; s^{t-1}).$$

*Where, we have*

$$\frac{1}{18B^2L^2\tilde{\eta}^2}\,\mathbb{E}_t\|e_t\|^2 \leq \begin{cases} \mathbb{E}\|c_t\|^2 & \text{MIME}, \\ \mathbb{E}\|\nabla f(x^t)\|^2 + G^2 + \frac{\sigma^2}{2K} & \text{MIMELITE}. \end{cases}$$

*Proof.* Now, combining Lemmas 5, 6, shows that running Mime or MimeLite is equivalent to

$$x^t = x^{t-1} - \tilde{\eta}\mathcal{U}(g^t + e^t; s^{t-1})$$
$$s^t = \mathcal{V}(g^t; s^{t-1}),$$

where for Mime we use

$$g^t_{\text{Mime}} = \frac{1}{S}\sum_i \nabla f_i(x) \text{ with } \mathbb{E}[g^t_{\text{Mime}}] = \nabla f(x^{t-1}) \text{ and } \mathbb{E}\|g^t_{\text{Mime}} - \nabla f(x^{t-1})\|^2 \leq \frac{G^2}{S}.$$

and for MimeLite we use

$$g^t_{\text{MimeLite}} = \frac{1}{KS}\sum_{i,k} \nabla f_i(x; \zeta_{i,k}) \text{ with } \mathbb{E}[g^t_{\text{Mime}}] = \nabla f(x^{t-1}) \text{ and } \mathbb{E}\|g^t_{\text{Mime}} - \nabla f(x^{t-1})\|^2 \leq \frac{G^2}{S} + \frac{\sigma^2}{KS}.$$

This shows the first part of the theorem. For the second part of the theorem, using the bound from Lemma 8 for Mime,

$$\mathbb{E}\|e^t\| \leq L^2\mathcal{E}^t_K \leq 18L^2B^2\tilde{\eta}^2\,\mathbb{E}\|c^t\|^2.$$

For MimeLite, we will instead use the bound from Lemma 7,

$$\mathbb{E}\|e^t_{\text{MimeLite}}\| \leq L^2\mathcal{E}^t_K + \frac{\sigma^2}{KS} \leq 18L^2B^2\tilde{\eta}^2\,\mathbb{E}\|\nabla f(x^t)\|^2 + 18L^2B^2\tilde{\eta}^2G^2 + \frac{9L^2B^2\tilde{\eta}^2\sigma^2}{K} + \frac{\sigma^2}{KS}.$$

$\square$

Note that the Lemma we proved here is slightly stronger than the theorem in the main section (up to constants which were suppressed).

## G.2 Convergence of MimeSGD and MimeLiteSGD (Corollary II)

Theorem I shows that Mime and MimeLite mimic a centralized algorithm quite closely up to error $\mathcal{O}(\tilde{\eta}^2)$. Then, analyzing the sensitivity of the base algorithm to such perturbation yields specific rates of convergence. We perform such an analysis using SGD as our base optimizer.

Properties of SGD as the base optimizer:

- $s^t$ is empty i.e. there are no global statistics used.
- $\mathcal{U}(g; s^{t-1}) = g$ for any $g$ and $B = 1$.

With this in mind, we proceed.

**Lemma 10** (Progress in one round). *Given that $f$ is $L$-smooth, and for any step-size $\tilde{\eta} \leq \frac{1}{2(B+2)L}$ for $B \geq 1$ we have*

$$f(x^t) \leq f(x^{t-1}) - \frac{\tilde{\eta}}{4}\,\mathbb{E}\|\nabla f(x^{t-1})\|^2 + \tilde{\eta}\,\mathbb{E}\|e^t\|^2 + \frac{L\tilde{\eta}^2G^2}{S}.$$

*Proof.* Starting from the update equation and the smoothness of $f$, we have

$$\mathbb{E}\,f(\boldsymbol{x}^t) \leq \mathbb{E}\,f(\boldsymbol{x}^{t-1}) + \mathbb{E}\langle \nabla f(\boldsymbol{x}^{t-1}), \boldsymbol{x}^t - \boldsymbol{x}^{t-1}\rangle + \frac{L}{2}\,\mathbb{E}\|\boldsymbol{x}^t - \boldsymbol{x}^{t-1}\|^2$$

$$= \mathbb{E}\,f(\boldsymbol{x}^{t-1}) - \tilde{\eta}\,\mathbb{E}\|\nabla f(\boldsymbol{x}^{t-1})\|^2 + \tilde{\eta}\langle \nabla f(\boldsymbol{x}^{t-1}), \boldsymbol{e}^t\rangle + \frac{L\tilde{\eta}^2}{2}\,\mathbb{E}\|\boldsymbol{c}^t + \boldsymbol{e}^t\|^2$$

$$\leq \mathbb{E}\,f(\boldsymbol{x}^{t-1}) - \frac{\tilde{\eta}}{2}\,\mathbb{E}\|\nabla f(\boldsymbol{x}^{t-1})\|^2 + \frac{\tilde{\eta}}{2}\|\boldsymbol{e}^t\|^2 + \frac{2L\tilde{\eta}^2}{2}\,\mathbb{E}\|\boldsymbol{c}^t\|^2 + \frac{2L\tilde{\eta}^2}{2}\,\mathbb{E}\|\boldsymbol{e}^t\|^2$$

$$\leq \mathbb{E}\,f(\boldsymbol{x}^{t-1}) - \left(\frac{\tilde{\eta}}{2} - \frac{2L\tilde{\eta}^2}{2}\right)\mathbb{E}\|\nabla f(\boldsymbol{x}^{t-1})\|^2 + \left(L\tilde{\eta}^2 + \frac{\tilde{\eta}}{2}\right)\mathbb{E}\|\boldsymbol{e}^t\|^2 + \frac{2L\tilde{\eta}^2 G^2}{2S}\,.$$

Using the bound on the step size that $\tilde{\eta} \leq \frac{1}{4L}$ yields the lemma. $\qquad\square$

**One round progress for MimeSGD.** Next, we specialize the convergence rate for Mime.

**Lemma 11.** *Suppose $f$ is a L-smooth function satisfying PL-inequality for $\mu \geq 0$ ($\mu = 0$ corresponds to the general case). Running MimeSGD for $\tilde{\eta} \leq \frac{1}{12BL}$ satisfies*

$$\frac{\tilde{\eta}}{16}\,\mathbb{E}\|\nabla f(\boldsymbol{x}^{t-1})\|^2 \leq (1 - \tfrac{\mu\tilde{\eta}}{8})(f(\boldsymbol{x}^{t-1}) - f^\star) - (f(\boldsymbol{x}^t) - f^\star) + \frac{3L\tilde{\eta}^2 G^2}{S}\,.$$

*Proof.* Recall from Lemma 9 that for Mime,

$$\mathbb{E}\|\boldsymbol{e}^t\|^2 \leq 18L^2 B^2 \tilde{\eta}^2\,\mathbb{E}\|\boldsymbol{c}^t\|^2 \leq 18L^2 B^2 \tilde{\eta}^2\,\mathbb{E}\|\nabla f(\boldsymbol{x}^{t-1})\|^2 + \frac{18L^2 B^2 \tilde{\eta}^2 G^2}{S}\,.$$

Combining this with Lemma 10 yields the following progress for Mime

$$f(\boldsymbol{x}^t) \leq f(\boldsymbol{x}^{t-1}) - \left(\frac{\tilde{\eta}}{4} - 18L^2 B^2 \tilde{\eta}^3\right)\mathbb{E}\|\nabla f(\boldsymbol{x}^{t-1})\|^2 + \frac{(L\tilde{\eta}^2 + 18L^2 B^2 \tilde{\eta}^3)G^2}{S}$$

$$\leq f(\boldsymbol{x}^{t-1}) - \frac{\tilde{\eta}}{8}\,\mathbb{E}\|\nabla f(\boldsymbol{x}^{t-1})\|^2 + \frac{3L\tilde{\eta}^2 G^2}{S}\,.$$

Here, we used the bound on the step size that $\tilde{\eta} \leq \frac{1}{12LB}$ implies $18L^2 B^2 \tilde{\eta}^2 \leq \frac{1}{8}$. Now using PL-inequality, we can write

$$f(\boldsymbol{x}^t) - f^\star \leq f(\boldsymbol{x}^{t-1}) - f^\star - \frac{\mu\tilde{\eta}}{8}(f(\boldsymbol{x}^{t-1}) - f^\star) - \frac{\tilde{\eta}}{16}\,\mathbb{E}\|\nabla f(\boldsymbol{x}^{t-1})\|^2 + \frac{3L\tilde{\eta}^2 G^2}{S}\,.$$

This yields the lemma. $\qquad\square$

We are now ready to derive the convergence rate.

**Convergence rate of MimeSGD on general non-convex functions.** Set $\mu = 0$ in Lemma 11 and sum over $t$

$$\frac{1}{T}\sum_{t=1}^{T}\mathbb{E}\|\nabla f(\boldsymbol{x}^{t-1})\|^2 \leq \frac{16(f(\boldsymbol{x}^0) - f^\star)}{\tilde{\eta}T} + \frac{48L\tilde{\eta}G^2}{S}$$

$$\leq 16\sqrt{\frac{3LG^2(f(\boldsymbol{x}^0) - f^\star)}{ST}} + \frac{192BL(f(\boldsymbol{x}^0) - f^\star)}{T}\,.$$

The final step used a step-size of $\tilde{\eta} = \min\left(\frac{1}{12BL}, \frac{1}{4L}, \sqrt{\frac{S(f(\boldsymbol{x}^0) - f^\star)}{3LTG^2}}\right)$. Here, we used $\boldsymbol{x}^{\text{out}} = \boldsymbol{x}^\tau$ where $\tau$ is uniformly at random chosen in $[T]$.

**Convergence rate of MimeSGD on PL-inequality.** Multiply Lemma 11 by $(1 - \frac{\mu\tilde{\eta}}{8})^{T-t}$ and sum over $t$

$$\sum_{t=1}^{T}(1 - \tfrac{\mu\tilde{\eta}}{8})^{T-t}\,\mathbb{E}\|\nabla f(\boldsymbol{x}^{t-1})\|^2 \leq \sum_{t=1}^{T}(1 - \tfrac{\mu\tilde{\eta}}{8})^{T-(t-1)}\frac{16(f(\boldsymbol{x}^{t-1}) - f^\star)}{\tilde{\eta}}$$
$$- (1 - \tfrac{\mu\tilde{\eta}}{8})^{T-t}\frac{16(f(\boldsymbol{x}^t) - f^\star)}{\tilde{\eta}} + (1 - \tfrac{\mu\tilde{\eta}}{8})^{T-t}\frac{48L\tilde{\eta}G^2}{S}$$
$$\leq (1 - \tfrac{\mu\tilde{\eta}}{8})^T\frac{16(f(\boldsymbol{x}^0) - f^\star)}{\tilde{\eta}} + \sum_{t=1}^{T}(1 - \tfrac{\mu\tilde{\eta}}{8})^{T-t}\frac{48L\tilde{\eta}G^2}{S}\,.$$

Output $\boldsymbol{x}^{\mathrm{out}} = \boldsymbol{x}^\tau$ where $\tau$ is chosen with probability proportional to $(1 - \frac{\mu\tilde{\eta}}{8})^{T-t}$. Then, this yields

$$\mathbb{E}\|\nabla f(\boldsymbol{x}^{\mathrm{out}})\|^2 \leq (1-\tfrac{\mu\tilde{\eta}}{8})^T\frac{16(f(\boldsymbol{x}^0) - f^\star)}{\tilde{\eta}} + \frac{48L\tilde{\eta}G^2}{S} \leq \tilde{\mathcal{O}}\left(\frac{\sigma^2}{\mu T} + L(f(\boldsymbol{x}^0) - f^\star)\exp\left(-\frac{\mu T}{12BL}\right)\right)\,.$$

Using an appropriate step-size $\tilde{\eta}$ yields the final rate (see Lemma 1 of [32]).

**One round progress for MimeLiteSGD.** Next, we specialize the convergence rate for MimeLite.

**Lemma 12.** *Suppose $f$ is a $L$-smooth function satisfying PL-inequality for $\mu \geq 0$ ($\mu = 0$ corresponds to the general case). Running MimeLiteSGD for $\tilde{\eta} \leq \frac{1}{12BL}$ satisfies*

$$\frac{\tilde{\eta}}{16}\,\mathbb{E}\|\nabla f(\boldsymbol{x}^{t-1})\|^2 \leq (1-\tfrac{\mu\tilde{\eta}}{8})(f(\boldsymbol{x}^{t-1})-f^\star)-(f(\boldsymbol{x}^t)-f^\star)+\frac{L\tilde{\eta}^2G^2}{S}+18L^2B^2\tilde{\eta}^3\big(G^2 + \sigma^2/K\big)\,.$$

*Proof.* Recall from Lemma 9 that,

$$\mathbb{E}\|\boldsymbol{e}^t\|^2 \leq 18L^2B^2\tilde{\eta}^2\,\mathbb{E}\|\boldsymbol{c}^t\|^2 \leq 18L^2B^2\tilde{\eta}^2\,\mathbb{E}\|\nabla f(\boldsymbol{x}^{t-1})\|^2 + 18L^2B^2\tilde{\eta}^2G^2 + \frac{9L^2B^2\tilde{\eta}^2\sigma^2}{K}\,.$$

Combining this with Lemma 10 yields the following progress for Mime

$$f(\boldsymbol{x}^t) \leq f(\boldsymbol{x}^{t-1}) - \left(\frac{\tilde{\eta}}{4} - 18L^2B^2\tilde{\eta}^3\right)\mathbb{E}\|\nabla f(\boldsymbol{x}^{t-1})\|^2 + \frac{L\tilde{\eta}^2G^2}{S} + 18L^2B^2\tilde{\eta}^3\big(G^2 + \sigma^2/K\big)$$
$$\leq f(\boldsymbol{x}^{t-1}) - \frac{\tilde{\eta}}{8}\,\mathbb{E}\|\nabla f(\boldsymbol{x}^{t-1})\|^2 + \frac{L\tilde{\eta}^2G^2}{S} + 18L^2B^2\tilde{\eta}^3\big(G^2 + \sigma^2/K\big)\,.$$

Here, we used the bound on the step size that $\tilde{\eta} \leq \frac{1}{12LB}$ implies $18L^2B^2\tilde{\eta}^2 \leq \frac{1}{8}$. Now using PL-inequality, we can write

$$f(\boldsymbol{x}^t) - f^\star - (f(\boldsymbol{x}^{t-1}) - f^\star) \leq$$
$$-\frac{\mu\tilde{\eta}}{8}(f(\boldsymbol{x}^{t-1}) - f^\star) - \frac{\tilde{\eta}}{16}\,\mathbb{E}\|\nabla f(\boldsymbol{x}^{t-1})\|^2 + \frac{L\tilde{\eta}^2G^2}{S} + 18L^2B^2\tilde{\eta}^3\big(G^2 + \sigma^2/K\big)\,.$$

This yields the lemma. $\qquad\square$

We are now ready to derive the convergence rate.

**Convergence rate of MimeLiteSGD on general non-convex functions.** Define $\tilde{G}^2 = G^2 + \sigma^2/K$. Set $\mu = 0$ in Lemma 12 and sum over $t$

$$\frac{1}{T}\sum_{t=1}^{T}\mathbb{E}\|\nabla f(\boldsymbol{x}^{t-1})\|^2 \leq \frac{16(f(\boldsymbol{x}^0) - f^\star)}{\tilde{\eta}T} + \frac{16L\tilde{\eta}G^2}{S} + 288L^2B^2\tilde{\eta}^2\tilde{G}^2$$
$$\leq 16\sqrt{\frac{LG^2(f(\boldsymbol{x}^0) - f^\star)}{ST}} + 84\left(\frac{L\tilde{G}(f(\boldsymbol{x}^0) - f^\star)}{T}\right)^{2/3} + \frac{192BL(f(\boldsymbol{x}^0) - f^\star)}{T}\,.$$

The final step used an appropriate step-size of $\tilde{\eta}$, see Lemma 2 of [32]. Here, we used $\boldsymbol{x}^{\mathrm{out}} = \boldsymbol{x}^\tau$ where $\tau$ is uniformly at random chosen in $[T]$. Finally note that if $K \geq \frac{\sigma^2}{G^2}$, then $\tilde{G}^2 \leq 2G^2$.

**Convergence rate of MimeLiteSGD on PL-inequality.** Multiply Lemma 12 by $(1 - \frac{\mu\tilde{\eta}}{8})^{T-t}$ and sum over $t$

$$\sum_{t=1}^{T}(1 - \tfrac{\mu\tilde{\eta}}{8})^{T-t}\,\mathbb{E}\|\nabla f(\boldsymbol{x}^{t-1})\|^2 \le \sum_{t=1}^{T}(1 - \tfrac{\mu\tilde{\eta}}{8})^{T-(t-1)}\frac{16(f(\boldsymbol{x}^{t-1}) - f^\star)}{\tilde{\eta}}$$

$$- (1 - \tfrac{\mu\tilde{\eta}}{8})^{T-t}\frac{16(f(\boldsymbol{x}^{t}) - f^\star)}{\tilde{\eta}}$$

$$+ \sum_{t=1}^{T}(1 - \tfrac{\mu\tilde{\eta}}{8})^{T-t}\left(\frac{16L\tilde{\eta}G^2}{S} + 288L^2B^2\tilde{\eta}^2\tilde{G}^2\right)$$

$$\le (1 - \tfrac{\mu\tilde{\eta}}{8})^{T}\frac{16(f(\boldsymbol{x}^{0}) - f^\star)}{\tilde{\eta}}$$

$$+ \sum_{t=1}^{T}(1 - \tfrac{\mu\tilde{\eta}}{8})^{T-t}\left(\frac{16L\tilde{\eta}G^2}{S} + 288L^2B^2\tilde{\eta}^2\tilde{G}^2\right).$$

Output $\boldsymbol{x}^{\text{out}} = \boldsymbol{x}^\tau$ where $\tau$ is chosen with probability proportional to $(1 - \frac{\mu\tilde{\eta}}{8})^{T-t}$. Then, this yields with appropriate step-size $\tilde{\eta}$ yields the final rate (see Lemma 1 of [32]).

$$\mathbb{E}\|\nabla f(\boldsymbol{x}^{\text{out}})\|^2 \le \tilde{\mathcal{O}}\left(\frac{\sigma^2}{\mu T} + \frac{L^2\tilde{G}^2}{\mu^2 T^2} + L(f(\boldsymbol{x}^0) - f^\star)\exp\left(-\frac{\mu T}{12BL}\right)\right).$$

### G.3 Convergence of MimeAdam and MimeLiteAdam (Corollary III)

We will largely follow the convergence analysis of [75] for the analysis of Adam. A crucial difference between their setting and ours is that in our algorithm we use the global statistics (second order moment) corresponding to $t-1$ i.e. $\sqrt{\boldsymbol{v}^{t-1}}$ instead of $\sqrt{\boldsymbol{v}^{t}}$ where the $\sqrt{\cdot}$ operator is applied element wise. Practically, this does not make a significant difference since the discount (momentum) factor for the second momentum is very large. Theoretically however, this difference simplifies our proof significantly removing otherwise hard to handle stochastic dependencies.

In this section, we will use Adam as our base optimizer with $\varepsilon_0 > 0$ parameter for stability and $\beta_1 = 0$ (i.e. RMSProp). This is identical to the setting in the centralized algorithm analyzed by [75]. The properties of our base optimizer are then:

- $\boldsymbol{s}^t = \boldsymbol{v}^t$ which is a running average estimate of the second moment and satisfies $\boldsymbol{v}^t > 0$.
- $\mathcal{U}(\boldsymbol{g}; \boldsymbol{v}^{t-1}) = \frac{\boldsymbol{g}}{\sqrt{\boldsymbol{v}^{t-1}} + \varepsilon_0}$ for any $\boldsymbol{g}$. This update for any $\boldsymbol{v}^{t-1}$ is $B$-Lipschitz for $B = \frac{1}{\varepsilon_0}$.

In this sub-section, all operations on vectors (multiplication, division, addition, comparison) are applied element-wise with appropriate broad-casting.

**One round progress of Adam.**

**Lemma 13** (Effective step-sizes). *Suppose that $|\nabla_j f_i(\boldsymbol{x})| \le H$. Then Adam has effective step-sizes*

$$\frac{1}{H + \varepsilon_0}\boldsymbol{g} \le \mathcal{U}(\boldsymbol{g}; \boldsymbol{v}^{t-1}) \le \frac{1}{\varepsilon_0}\boldsymbol{g}.$$

*Proof.* Recall that $\boldsymbol{v}^t = \beta_2\boldsymbol{v}^{t-1} + (1 - \beta_2)(\boldsymbol{c}^t)^2$ starting from $\boldsymbol{v}^0 = 0$. Thus for any $t \ge 0$, we have $\boldsymbol{v}^t \ge 0$ and hence $\sqrt{\boldsymbol{v}^{t-1}} + \varepsilon_0 \ge \varepsilon_0$. For the other side, recall that $\boldsymbol{v}^t$ is updated with centralized stochastic gradients $\boldsymbol{c}^t = \frac{1}{S}\sum_i \nabla f_i(\boldsymbol{x})$.

$$[\boldsymbol{c}^t]_j = \frac{1}{S}\sum_i [\nabla f_i(\boldsymbol{x})]_j \le H.$$

Further,
$$[\boldsymbol{v}^t]_j = \beta_2[\boldsymbol{v}^{t-1}]_j + (1 - \beta_2)[\boldsymbol{c}^t]_j^2 \le \beta_2[\boldsymbol{v}^{t-1}]_j + (1 - \beta_2)H^2 \le H^2.$$

Hence $\sqrt{\boldsymbol{v}^{t-1}} + \varepsilon_0 \le H + \varepsilon_0$. $\square$

**Lemma 14** (One round progress). *For one round of Adam with error $e^t$ in the update $\mathcal{U}$ and using $c^t$ for update $\mathcal{V}$, we have*

$$\mathbb{E}\,f(\boldsymbol{x}^t) \le \mathbb{E}\,f(\boldsymbol{x}^{t-1}) - \frac{\tilde{\eta}}{4(H+\varepsilon_0)}\|\nabla f(\boldsymbol{x}^{t-1})\|^2 + \frac{\tilde{\eta}((H+\varepsilon_0)+\varepsilon_0/(H+\varepsilon_0))}{2\varepsilon_0^2}\,\mathbb{E}\|\boldsymbol{e}^t\|^2 + \frac{L\tilde{\eta}^2 G^2}{S\varepsilon_0^2}\ .$$

*Proof.* Starting from Lemma 13 and the smoothness of $f$, we have

$$\mathbb{E}\,f(\boldsymbol{x}^t) \le \mathbb{E}\,f(\boldsymbol{x}^{t-1}) - \tilde{\eta}\,\mathbb{E}\langle\nabla f(\boldsymbol{x}^{t-1}), \mathbb{E}_t[\mathcal{U}(\boldsymbol{c}^t+\boldsymbol{e}^t)]\rangle + \frac{L\tilde{\eta}^2}{2}\,\mathbb{E}\|\mathcal{U}(\boldsymbol{c}^t+\boldsymbol{e}^t; \boldsymbol{v}^{t-1})\|^2$$

$$\le \mathbb{E}\,f(\boldsymbol{x}^{t-1}) - \tilde{\eta}\,\mathbb{E}\langle\nabla f(\boldsymbol{x}^{t-1}), \mathbb{E}_t\left[\frac{\boldsymbol{c}^t+\boldsymbol{e}^t}{\sqrt{\boldsymbol{v}^{t-1}}+\varepsilon_0}\right]\rangle + \frac{L\tilde{\eta}^2}{2}\,\mathbb{E}\|\mathcal{U}(\boldsymbol{c}^t+\boldsymbol{e}^t; \boldsymbol{v}^{t-1})\|^2$$

$$\le \mathbb{E}\,f(\boldsymbol{x}^{t-1}) - \tilde{\eta}\,\mathbb{E}\langle\nabla f(\boldsymbol{x}^{t-1}), \left[\frac{\nabla f(\boldsymbol{x}^{t-1})+\boldsymbol{e}^t}{\sqrt{\boldsymbol{v}^{t-1}}+\varepsilon_0}\right]\rangle + \frac{L\tilde{\eta}^2}{2\varepsilon_0^2}\,\mathbb{E}\|\boldsymbol{c}^t+\boldsymbol{e}^t\|^2$$

$$\le \mathbb{E}\,f(\boldsymbol{x}^{t-1}) - \frac{\tilde{\eta}}{H+\varepsilon_0}\|\nabla f(\boldsymbol{x}^{t-1})\|^2 - \tilde{\eta}\,\mathbb{E}\langle\nabla f(\boldsymbol{x}^{t-1}), \frac{\boldsymbol{e}^t}{\sqrt{\boldsymbol{v}^{t-1}}+\varepsilon_0}\rangle + \frac{L\tilde{\eta}^2}{2\varepsilon_0^2}\,\mathbb{E}\|\boldsymbol{c}^t+\boldsymbol{e}^t\|^2$$

$$\le \mathbb{E}\,f(\boldsymbol{x}^{t-1}) - \frac{\tilde{\eta}}{2(H+\varepsilon_0)}\|\nabla f(\boldsymbol{x}^{t-1})\|^2 + \frac{\tilde{\eta}(H+\varepsilon_0)}{2}\,\mathbb{E}\|\frac{\boldsymbol{e}^t}{\sqrt{\boldsymbol{v}^{t-1}}+\varepsilon_0}\|^2 + \frac{L\tilde{\eta}^2}{2\varepsilon_0^2}\,\mathbb{E}\|\boldsymbol{c}^t+\boldsymbol{e}^t\|^2$$

$$\le \mathbb{E}\,f(\boldsymbol{x}^{t-1}) - \left(\frac{\tilde{\eta}}{2(H+\varepsilon_0)} - \frac{L\tilde{\eta}^2}{\varepsilon_0^2}\right)\|\nabla f(\boldsymbol{x}^{t-1})\|^2 + \frac{\tilde{\eta}(H+\varepsilon_0)+2L\tilde{\eta}^2}{2\varepsilon_0^2}\,\mathbb{E}\|\boldsymbol{e}^t\|^2 + \frac{L\tilde{\eta}^2 G^2}{S\varepsilon_0^2}$$

$$\le \mathbb{E}\,f(\boldsymbol{x}^{t-1}) - \frac{\tilde{\eta}}{4(H+\varepsilon_0)}\|\nabla f(\boldsymbol{x}^{t-1})\|^2 + \frac{\tilde{\eta}((H+\varepsilon_0)+\varepsilon_0/(H+\varepsilon_0))}{2\varepsilon_0^2}\,\mathbb{E}\|\boldsymbol{e}^t\|^2 + \frac{L\tilde{\eta}^2 G^2}{S\varepsilon_0^2}$$

Here we used our bound on the step-size that $\tilde{\eta} \le \frac{\varepsilon_0}{4L(H+\varepsilon_0)}$. $\qquad\square$

### Convergence of MimeAdam.

**Lemma 15.** *Suppose that assumptions A1–(A3) hold and further $|\nabla_j f_i(\boldsymbol{x})| \le H$. Then, running MimeAdam with step-size $\tilde{\eta} \le \frac{\varepsilon_0^2}{12L(H+\varepsilon_0)}$, we have*

$$\frac{1}{T}\sum_{t=1}^{T}\mathbb{E}\|\nabla f(\boldsymbol{x}^{t-1})\|^2 \le \frac{96L(H+\varepsilon_0)^2(f(\boldsymbol{x}_0)-f^\star)}{\varepsilon_0^2 T} + \frac{2G^2}{S}\ .$$

Combining Lemma 14 with the bound on $\boldsymbol{e}^t$ from Lemma 9 we get,

$$\mathbb{E}\,f(\boldsymbol{x}^t) \le \mathbb{E}\,f(\boldsymbol{x}^{t-1}) - \frac{\tilde{\eta}}{4(H+\varepsilon_0)}\|\nabla f(\boldsymbol{x}^{t-1})\|^2 + \frac{\tilde{\eta}((H+\varepsilon_0)+\varepsilon_0/(H+\varepsilon_0))}{2\varepsilon_0^2}\,\mathbb{E}\|\boldsymbol{e}^t\|^2 + \frac{L\tilde{\eta}^2 G^2}{S\varepsilon_0^2}$$

$$\le \mathbb{E}\,f(\boldsymbol{x}^{t-1}) - \frac{\tilde{\eta}}{4(H+\varepsilon_0)}\|\nabla f(\boldsymbol{x}^{t-1})\|^2 + \frac{9L^2\tilde{\eta}^3((H+\varepsilon_0)+\varepsilon_0/(H+\varepsilon_0))}{\varepsilon_0^4}\,\mathbb{E}\|\boldsymbol{c}^t\|^2$$

$$\qquad + \frac{L\tilde{\eta}^2 G^2}{S\varepsilon_0^2}$$

$$\le \mathbb{E}\,f(\boldsymbol{x}^{t-1}) - \left(\frac{\tilde{\eta}}{4(H+\varepsilon_0)} - \frac{9L^2\tilde{\eta}^3((H+\varepsilon_0)+\varepsilon_0/(H+\varepsilon_0))}{\varepsilon_0^4}\right)\|\nabla f(\boldsymbol{x}^{t-1})\|^2$$

$$\qquad + \frac{L\tilde{\eta}^2 G^2}{S\varepsilon_0^2} + \frac{9L^2\tilde{\eta}^3((H+\varepsilon_0)+\varepsilon_0/(H+\varepsilon_0))G^2}{S\varepsilon_0^4}$$

$$\le \mathbb{E}\,f(\boldsymbol{x}^{t-1}) - \left(\frac{\tilde{\eta}}{4(H+\varepsilon_0)} - \frac{18L^2\tilde{\eta}^3(H+\varepsilon_0)}{\varepsilon_0^4}\right)\|\nabla f(\boldsymbol{x}^{t-1})\|^2$$

$$\qquad + \frac{L\tilde{\eta}^2 G^2}{S\varepsilon_0^2} + \frac{18L^2\tilde{\eta}^3(H+\varepsilon_0)G^2}{S\varepsilon_0^4}$$

$$\le \mathbb{E}\,f(\boldsymbol{x}^{t-1}) - \frac{\tilde{\eta}}{8(H+\varepsilon_0)}\|\nabla f(\boldsymbol{x}^{t-1})\|^2 + \frac{\tilde{\eta}G^2}{4S(H+\varepsilon_0)}\ .$$

To simplify computations, here we assumed we assumed $(H + \varepsilon_0)^2 \geq \varepsilon_0$ without loss of generality. If this is not true, we can replace $H$ by $\max(H, \sqrt{\varepsilon_0} - \varepsilon_0)$. Assuming $\tilde{\eta} \leq \frac{\varepsilon_0^2}{12L(H+\varepsilon_0)}$, we have $\frac{18L^2\tilde{\eta}^2(H+\varepsilon_0)}{\varepsilon_0^4} \leq \frac{1}{8(H+\varepsilon_0)}$. Rearranging the terms and substituting the bounds on the step-size yields the lemma.

**Convergence of MimeLiteAdam.**

**Lemma 16.** *Suppose that assumptions A1–(A3) hold and further $|\nabla_j f_i(\boldsymbol{x})| \leq H$. Then, running MimeLiteAdam with step-size $\tilde{\eta} \leq \frac{\varepsilon_0^2}{12L\sqrt{S}(H+\varepsilon_0)}$, we have for $\tilde{G}^2 := G^2 + \sigma^2/K$,*

$$\frac{1}{T} \sum_{t=1}^{T} \mathbb{E} \|\nabla f(\boldsymbol{x}^{t-1})\|^2 \leq \frac{96L\sqrt{S}(H+\varepsilon_0)^2(f(\boldsymbol{x}_0) - f^\star)}{\varepsilon_0^2 T} + \frac{2\tilde{G}^2}{S}.$$

Combining Lemma 14 with the bound on $\boldsymbol{e}^t$ from Lemma 9 we get for $\tilde{G}^2 := G^2 + \sigma^2/K$,

$$\mathbb{E} f(\boldsymbol{x}^t) \leq \mathbb{E} f(\boldsymbol{x}^{t-1}) - \frac{\tilde{\eta}}{4(H+\varepsilon_0)} \|\nabla f(\boldsymbol{x}^{t-1})\|^2 + \frac{\tilde{\eta}(H+\varepsilon_0)}{\varepsilon_0^2} \mathbb{E}\|\boldsymbol{e}^t\|^2 + \frac{L\tilde{\eta}^2 G^2}{S\varepsilon_0^2}$$

$$\leq \mathbb{E} f(\boldsymbol{x}^{t-1}) - \frac{\tilde{\eta}}{4(H+\varepsilon_0)} \|\nabla f(\boldsymbol{x}^{t-1})\|^2$$

$$+ \frac{18L^2\tilde{\eta}^3(H+\varepsilon_0)}{\varepsilon_0^4} \mathbb{E}\|\nabla f(\boldsymbol{x}^{t-1})\|^2 + \frac{18L^2\tilde{\eta}^3(H+\varepsilon_0)(\tilde{G}^2)}{\varepsilon_0^4} + \frac{L\tilde{\eta}^2 G^2}{S\varepsilon_0^2}$$

$$\leq \mathbb{E} f(\boldsymbol{x}^{t-1}) - \frac{\tilde{\eta}}{8(H+\varepsilon_0)} \|\nabla f(\boldsymbol{x}^{t-1})\|^2 + \frac{\tilde{\eta}\tilde{G}^2}{4S(H+\varepsilon_0)}$$

Again as before to simplify computations, here we assumed $(H + \varepsilon_0)^2 \geq \varepsilon_0$ without loss of generality. If this is not true, we can replace $H$ by $\max(H, \sqrt{\varepsilon_0} - \varepsilon_0)$. Assuming $\tilde{\eta} \leq \frac{\varepsilon_0^2}{12L(H+\varepsilon_0)\sqrt{S}}$, we have $\frac{18L^2\tilde{\eta}^2(H+\varepsilon_0)}{\varepsilon_0^4} \leq \frac{1}{8S(H+\varepsilon_0)}$. Rearranging the terms and substituting the bounds on the step-size yields the lemma.

## H   Circumventing server-only lower bounds

In this section we see how to use momentum based variance reduction [14, 62] to reduce the variance of the updates and improve convergence. It should be noted that MVR does not exactly fit the MIME framework (BASEOPT) since it requires computing gradients at two points on the same batch. However, it is straightforward to extend the idea of MIME to MVR as we will now do. We use MVR as a theoretical justification for why the usual momentum works well in practice. An interesting future direction would be to adapt the algorithm and analysis of [13], which does fit the framework of MIME.

For the sake of convenience, we summarize the notation used in the proof in a table.

Table 6: Summary of all notation used in the MVR proofs

| | |
|---|---|
| $\sigma^2$, $G^2$, and $\delta$ | intra-client gradient, inter-client gradient, and inter-client Hessian variance |
| $\eta, a$ | step-size, $(1 - \beta)$ momentum parameters |
| $T, t$ | total number, index of communication rounds |
| $K, k$ | total number, index of client local update steps |
| $\mathcal{S}^t$, $S$, and $i$ | sampled set, size, and index of clients in round $t$ |
| $\boldsymbol{x}^t$ | aggregated server model *after* round $t$ |
| $\boldsymbol{m}^t$ | server momentum computed *after* round $t$ |
| $\boldsymbol{c}^t$ | control variate of server *after* round $t$ (only MIME) |
| $\boldsymbol{y}_{i,k}^t$ | model parameters of $i$th client in round $t$ *after* step $k$ |
| $\zeta_{i,k}^t$ | mini-batch data used by $i$th client in round $t$ and step $k$ |
| $\boldsymbol{d}_{i,k}^t$ | parameter update by $i$th client in round $t$, step $k$ |
| $\boldsymbol{e}^t$ | error in momentum $\boldsymbol{m}^t - \nabla f(\boldsymbol{x}^{t-1})$ |
| $\Delta_{i,k}^t, \Delta^{t-1}$ | $\mathbb{E}\|\boldsymbol{y}_{i,k}^t - \boldsymbol{x}^{t-2}\|^2$, $\mathbb{E}\|\boldsymbol{x}^{t-1} - \boldsymbol{x}^{t-2}\|^2 = \Delta_{i,0}^t$ |

## H.1 Algorithm descriptions

Now, we formally describe the MIME MVR and MIMELITE MVR algorithms. In each round $t$, we sample clients $\mathcal{S}^t$ such that $|\mathcal{S}^t| = S$. The server communicates the server parameters $\boldsymbol{x}^{t-1}$, the past parameters $\boldsymbol{x}^{t-2}$, and the momentum $\boldsymbol{m}^{t-1}$ term. MIME additionally uses a control variate $\boldsymbol{c}^{t-1}$ as we describe next.

**Control variate in Mime.** MIME uses an additional control variate $\boldsymbol{c}^{t-1}$ to reduce the variance.

$$\boldsymbol{c}^{t-1} = \frac{1}{S} \sum_{i \in \mathcal{S}^t} \nabla f_i(\boldsymbol{x}^{t-2}) \,. \tag{12}$$

Note that both $\boldsymbol{c}^{t-1}$ and $\boldsymbol{m}^{t-1}$ use gradients and parameters from previous rounds (different from the previous section). A naive implementation of this method requires two steps of communication per round to implement this algorithm. Alternatively, we can reserve some clients in the previous round for computing $\boldsymbol{c}^{t-1}$ which can then be used in the current round, removing the need for two steps of communication. In particular, it can be computed on a different set of an independent sampled clients $\tilde{\mathcal{S}}^{t-1}$. In fact, all our theoretical results hold even if we use *a single client* to perform the local updates and the rest of clients are used only to compute $\boldsymbol{c}^{t-1}$ each round.

**Local client updates.** Then each client $i \in \mathcal{S}^t$ makes a copy $\boldsymbol{y}^t_{i,0} = \boldsymbol{x}^{t-1}$ and perform $K$ local client updates. In each local client update $k \in [K]$, the client samples a dataset $\zeta^t_{i,k}$. MIME performs the following update:

$$\begin{aligned}
\boldsymbol{y}^t_{i,k} &= \boldsymbol{y}^t_{i,k-1} - \eta \boldsymbol{d}^t_{i,k} \,, \text{ where} \\
\boldsymbol{d}^t_{i,k} &= a(\nabla f_i(\boldsymbol{y}^t_{i,k-1}; \zeta^t_{i,k}) - \nabla f_i(\boldsymbol{x}^{t-1}; \zeta^t_{i,k}) + \boldsymbol{c}^{t-1}) + (1-a)\boldsymbol{m}^{t-1} \\
&\quad + (1-a)(\nabla f_i(\boldsymbol{y}^t_{i,k-1}; \zeta^t_{i,k}) - \nabla f_i(\boldsymbol{x}^{t-1}; \zeta^t_{i,k})) \,.
\end{aligned} \tag{13}$$

MIMELITE on the other hand uses a very similar but simpler update scheme which does not rely on $\boldsymbol{c}^{t-1}$:

$$\begin{aligned}
\boldsymbol{y}^t_{i,k} &= \boldsymbol{y}^t_{i,k-1} - \eta \boldsymbol{d}^t_{i,k} \,, \text{ where} \\
\boldsymbol{d}^t_{i,k} &= a\nabla f_i(\boldsymbol{y}^t_{i,k-1}; \zeta^t_{i,k}) + (1-a)\boldsymbol{m}^{t-1} \\
&\quad + (1-a)(\nabla f_i(\boldsymbol{y}^t_{i,k-1}; \zeta^t_{i,k}) - \nabla f_i(\boldsymbol{x}^{t-1}; \zeta^t_{i,k})) \,.
\end{aligned} \tag{14}$$

**Server updates.** After $K$ such local updates, the server then aggregates the new client parameters as

$$\boldsymbol{x}^t = \frac{1}{S} \sum_{j \in \mathcal{S}^t} \boldsymbol{y}^t_{j,K} \,. \tag{15}$$

The momentum term is updated at the end of the round for $a \geq 0$ as

$$\boldsymbol{m}^t = \underbrace{a(\tfrac{1}{S} \sum_{j \in \mathcal{S}^t} \nabla f_j(\boldsymbol{x}^{t-1})) + (1-a)\boldsymbol{m}^{t-1}}_{\text{Mom}} + \underbrace{(1-a)(\tfrac{1}{S} \sum_{j \in \mathcal{S}^t} \nabla f_j(\boldsymbol{x}^{t-1}) - \nabla f_j(\boldsymbol{x}^{t-2}))}_{\text{correction}} \,. \tag{16}$$

As we can see, the momentum update of MVR can be broken down into the usual Mom update, and a correction. Intuitively, this correction term is very small since $f_i$ is smooth and $\boldsymbol{x}^{t-1} \approx \boldsymbol{x}^{t-2}$. Another way of looking at the update (16) is to note that if all functions are identical i.e. $f_j = f_k$ for any $j, k$, then (16) just becomes the usual gradient descent. Thus MimeMVR tries to maintain an exponential moving average of only the variance terms, reducing its bias. We refer to [14] for more detailed explanation of MVR.

## H.2 Bias in updates

The main difference in MimeMVR from the centralized versions of [62, 14] is the additional local steps which are biased. In particular, for $k \geq 1$ the expected gradient $\mathbb{E}[\nabla f_i(\boldsymbol{y}^t_{i,k})] \neq \nabla f(\boldsymbol{y}^t_{i,k})$ because $\boldsymbol{y}^t_{i,k}$ also depends on the sample $i$. This bias is in fact the underlying cause of client drift and controlling it is a crucial step for our analysis.

**Lemma 17** (Mime bias). *For any values of $\boldsymbol{x}$ and $\boldsymbol{y}_i$ where $\boldsymbol{y}_i$ may depend on $i$, the following holds for any client $i$ almost surely given that* (A1) *and* (A2) *hold:*

$$\mathbb{E}_{\mathcal{S},\zeta}\left\|\nabla f_i(\boldsymbol{y}_i;\zeta) + \frac{1}{|\mathcal{S}|}\sum_{j\in\mathcal{S}}\nabla f_j(\boldsymbol{x}) - \nabla f_i(\boldsymbol{x};\zeta) \quad - \quad \nabla f(\boldsymbol{y}_i)\right\|^2 \leq 2\delta^2\,\mathbb{E}_{\mathcal{S}}\|\boldsymbol{y}_i - \boldsymbol{x}\|^2 + \frac{2G^2}{S}\,.$$

*Proof.* We can separate the noise from the rest of the terms and expand as

$$\mathbb{E}_{\zeta,\mathcal{S}}\left\|\nabla f_i(\boldsymbol{y}_i;\zeta) + \frac{1}{|\mathcal{S}|}\sum_{j\in\mathcal{S}}\nabla f_j(\boldsymbol{x}) - \nabla f_i(\boldsymbol{x};\zeta) - \nabla f(\boldsymbol{y}_i)\right\|^2$$

$$\leq 2\,\mathbb{E}_{\mathcal{S}}\|\nabla f_i(\boldsymbol{y}_i;\zeta) + \nabla f(\boldsymbol{x}) - \nabla f_i(\boldsymbol{x};\zeta) - \nabla f(\boldsymbol{y}_i)\|^2 + 2\,\mathbb{E}_{\mathcal{S}}\left\|\frac{1}{|\mathcal{S}|}\sum_{j\in\mathcal{S}}\nabla f_j(\boldsymbol{x}) - \nabla f(\boldsymbol{x})\right\|^2$$

$$\leq 2\,\mathbb{E}_{\mathcal{S}}\|\nabla f_i(\boldsymbol{y}_i;\zeta) + \nabla f(\boldsymbol{x}) - \nabla f_i(\boldsymbol{x};\zeta) - \nabla f(\boldsymbol{y}_i)\|^2 + \frac{2G^2}{S}$$

$$\leq 2\,\mathbb{E}_{\mathcal{S}}\,\delta^2\|\boldsymbol{y}_i - \boldsymbol{x}\|^2 + \frac{2G^2}{S}\,.$$

The first inequality used Young's inequality, the second used (A1), and the last used (A2) in the form of Lemma 3. $\qquad\square$

We can perform a similar analysis of the bias of local updates encountered by MimeLite.

**Lemma 18** (MimeLite bias). *For any values of $\boldsymbol{x}$ and $\boldsymbol{y}_i$ where $\boldsymbol{y}_i$ may depend on $i$, the following holds for any client $i$ randomly chosen from $\mathcal{C}$ given that* (A1), (A2) *and* (A3) *hold:*

$$\mathbb{E}_{i,\zeta}\|\nabla f_i(\boldsymbol{y}_i;\zeta) \quad - \quad \nabla f(\boldsymbol{y}_i)\|^2 \leq 2\delta^2\,\mathbb{E}_i\|\boldsymbol{y}_i - \boldsymbol{x}\|^2 + 2G^2 + \sigma^2\,.$$

*Proof.* We can separate the noise from the rest of the terms and expand as

$$\mathbb{E}_{\zeta,i}\|\nabla f_i(\boldsymbol{y}_i;\zeta) - \nabla f(\boldsymbol{y}_i)\|^2 = \mathbb{E}_{\zeta,i}\|\nabla f_i(\boldsymbol{y}_i;\zeta) \pm \nabla f_i(\boldsymbol{x}) \pm \nabla f(\boldsymbol{x}) - \nabla f(\boldsymbol{y}_i)\|^2$$

$$\leq \mathbb{E}_i\|\nabla f_i(\boldsymbol{y}_i) \pm \nabla f_i(\boldsymbol{x}) \pm \nabla f(\boldsymbol{x}) - \nabla f(\boldsymbol{y}_i)\|^2 + \sigma^2$$

$$\leq 2\,\mathbb{E}_i\|\nabla f_i(\boldsymbol{y}_i) + \nabla f(\boldsymbol{x}) - \nabla f_i(\boldsymbol{x}) - \nabla f(\boldsymbol{y}_i)\|^2$$

$$+ 2\,\mathbb{E}_i\|\nabla f_i(\boldsymbol{x}) - \nabla f(\boldsymbol{x})\|^2 + \sigma^2$$

$$\leq 2\,\mathbb{E}_i\|\nabla f_i(\boldsymbol{y}_i) + \nabla f(\boldsymbol{x}) - \nabla f_i(\boldsymbol{x}) - \nabla f(\boldsymbol{y}_i)\|^2 + 2G^2 + \sigma^2$$

$$\leq 2\delta^2\,\mathbb{E}_i\|\boldsymbol{y}_i - \boldsymbol{x}\|^2 + 2G^2 + \sigma^2\,.$$

The first inequality used (A3), the second used Young's inequality, the third used (A1), and the last used (A2) in the form of Lemma 3. $\qquad\square$

Note that the bias for MimeLite is very similar to that of Mime, except that Mime has dependence of $\frac{G^2}{S}$, whereas MimeLite has $G^2 + \sigma^2$. Hence, the rate of convergence of MimeLite will depend on $G^2$ wheras Mime will have the optimal dependency of $G^2/S$. Hence, in the rest of the proof, we will consider **only Mime** and simply replace $G^2/S$ with $(G^2 + \sigma^2)$ to obtain the corresponding results for MimeLite.

### H.3   Change in each client update

**Client update variance.**   Now we examine the variance of our update in each local step $\boldsymbol{d}_{i,k}^t$.

**Lemma 19.** *For the client update* (13), *given* (A1) *and* (A2), *the following holds for any $a \in [0,1]$ where $\boldsymbol{e}^t := \boldsymbol{m}^t - \nabla f(\boldsymbol{x}^{t-1})$ and $\Delta_{i,k}^t := \mathbb{E}\|\boldsymbol{y}_{i,k}^t - \boldsymbol{x}^{t-2}\|^2$:*

$$\mathbb{E}\|\boldsymbol{d}_{i,k}^t - \nabla f(\boldsymbol{y}_{i,k-1}^t)\|^2 \leq 3\,\mathbb{E}\|\boldsymbol{e}^{t-1}\|^2 + 3\delta^2\Delta_{i,k-1}^t + \frac{3a^2G^2}{S}\,.$$

*Proof.* Starting from the client update (13), we can rewrite it as

$$\boldsymbol{d}_{i,k}^t - \nabla f(\boldsymbol{y}_{i,k-1}^t) = (1-a)\boldsymbol{e}^{t-1}$$
$$+ \left( \nabla f_i(\boldsymbol{y}_{i,k-1}^t; \zeta_{i,k}^t) - \nabla f_i(\boldsymbol{x}^{t-2}; \zeta_{i,k}^t)) - \nabla f(\boldsymbol{y}_{i,k-1}^t) + \nabla f(\boldsymbol{x}^{t-2}) \right)$$
$$+ a\left( \frac{1}{S} \sum_{j \in \mathcal{S}^t} \nabla f_j(\boldsymbol{x}^{t-2}) - \nabla f(\boldsymbol{x}^{t-2}) \right).$$

We can use the relaxed triangle inequality Lemma 1 to claim

$$\mathbb{E}\|\boldsymbol{d}_{i,k}^t - \nabla f(\boldsymbol{y}_{i,k-1}^t)\|^2$$
$$= 3(1-a)^2\,\mathbb{E}\|\boldsymbol{e}^{t-1}\|^2$$
$$+ 3(1-a)^2\big\|(\nabla f_i(\boldsymbol{y}_{i,k-1}^t; \zeta_{i,k}^t) - \nabla f_i(\boldsymbol{x}^{t-2}; \zeta_{i,k}^t)) - (\nabla f(\boldsymbol{y}_{i,k-1}^t) - \nabla f(\boldsymbol{x}^{t-2}))\big\|^2$$
$$+ 3a^2 \left\| \frac{1}{S} \sum_{j \in \mathcal{S}^t} \nabla f_j(\boldsymbol{x}^{t-2}) - \nabla f(\boldsymbol{x}^{t-2}) \right\|^2$$
$$\leq 3\,\mathbb{E}\|\boldsymbol{e}^{t-1}\|^2 + 3\delta^2\|\boldsymbol{y}_{i,k-1}^t - \boldsymbol{x}^{t-2}\|^2 + \frac{3a^2G^2}{S}.$$

The last inequality used the Hessian similarity Lemma 3 to bound the second term and the heterogeneity bound (A1) to bound the last term. Also, $(1-a)^2 \leq 1$ since $a \in [0,1]$. $\qquad\square$

**Distance moved in each step.** We show that the distance moved by a client in each step during the client update can be controlled.

**Lemma 20.** *For MimeMVR updates* (13) *with* $\eta \leq \frac{1}{6K\delta}$ *and given* (A1) *and* (A2)*, the following holds*

$$\Delta_{i,k}^t \leq \left(1 + \frac{1}{K}\right)\Delta_{i,k-1}^t + 18\eta^2 K a^2 \frac{G^2}{S} + 18\eta^2 K\,\mathbb{E}\|\boldsymbol{e}^{t-1}\|^2 + 6\eta^2 K\|\nabla f(\boldsymbol{y}_{i,k-1}^t)\|^2,$$

*where we define* $\Delta_{i,k}^t := \mathbb{E}\|\boldsymbol{y}_{i,k}^t - \boldsymbol{x}^{t-2}\|^2$.

*Proof.* Starting from the MimeMVR update (13) and the relaxed triangle inequality with $c = 2K$,

$$\mathbb{E}\|\boldsymbol{y}_{i,k}^t - \boldsymbol{x}^{t-2}\|^2 = \mathbb{E}\|\boldsymbol{y}_{i,k-1}^t - \eta\boldsymbol{d}_{i,k}^t - \boldsymbol{x}^{t-2}\|^2$$
$$\leq \left(1 + \frac{1}{2K}\right)\mathbb{E}\|\boldsymbol{y}_{i,k-1}^t - \boldsymbol{x}^{t-2}\|^2 + (2K+1)\eta^2\,\mathbb{E}\|\boldsymbol{d}_{i,k}^t\|^2$$
$$\leq \left(1 + \frac{1}{2K}\right)\mathbb{E}\|\boldsymbol{y}_{i,k-1}^t - \boldsymbol{x}^{t-2}\|^2 + 6K\eta^2\,\mathbb{E}\|\boldsymbol{d}_{i,k}^t - \nabla f(\boldsymbol{y}_{i,k-1}^t)\|^2$$
$$+ 6K\eta^2\,\mathbb{E}\|\nabla f(\boldsymbol{y}_{i,k-1}^t)\|^2$$
$$\leq \left(1 + \frac{1}{2K} + 18K\eta^2\delta^2\right)\mathbb{E}\|\boldsymbol{y}_{i,k-1}^t - \boldsymbol{x}^{t-2}\|^2$$
$$+ 18K\eta^2\,\mathbb{E}\|\boldsymbol{e}^{t-1}\|^2 + \frac{18K\eta^2 a^2 G^2}{S} + 6K\eta^2\,\mathbb{E}\|\nabla f(\boldsymbol{y}_{i,k-1}^t)\|^2.$$

The last inequality used the update variance bound Lemma 19. We can simplify the expression further since $\eta \leq \frac{1}{6K\delta}$ implies $18K\eta^2\delta^2 \leq \frac{1}{2K}$. $\qquad\square$

**Progress in one step.** Now we can compute the progress made in each step.

**Lemma 21.** *For any client update step with step size* $\eta \leq \min\left(\frac{1}{L}, \frac{1}{192\delta K}\right)$ *and given that* (A1)*,* (A2) *hold, we have*

$$\mathbb{E}\,f(\boldsymbol{y}_{i,k}^t) + \delta\left(1 + \frac{2}{K}\right)^{K-k}\Delta_{i,k}^t \leq \mathbb{E}\,f(\boldsymbol{y}_{i,k-1}^t) + \delta\left(1 + \frac{2}{K}\right)^{K-(k-1)}\Delta_{i,k-1}^t$$
$$- \frac{\eta}{4}\,\mathbb{E}\|\nabla f(\boldsymbol{y}_{i,k-1}^t)\|^2 + 3\eta\,\mathbb{E}\|\boldsymbol{e}^{t-1}\|^2 + \frac{3\eta a^2 G^2}{S}.$$

*Proof.* The assumption that $f$ is $L$-smooth implies a quadratic upper bound (10).

$$f(\boldsymbol{y}_{i,k}^t) - f(\boldsymbol{y}_{i,k-1}^t) \leq -\eta\langle\nabla f(\boldsymbol{y}_{i,k-1}^t), \boldsymbol{d}_{i,k}^t\rangle + \frac{L\eta^2}{2}\|\boldsymbol{d}_{i,k}^t\|^2$$

$$= -\frac{\eta}{2}\|\nabla f(\boldsymbol{y}_{i,k-1}^t)\|^2 + \frac{L\eta^2 - \eta}{2}\|\boldsymbol{d}_{i,k}^t\|^2 + \frac{\eta}{2}\|\boldsymbol{d}_{i,k}^t - \nabla f(\boldsymbol{y}_{i,k-1}^t)\|^2.$$

The second equality used the fact that for any $a, b$, $-2ab = (a-b)^2 - a^2 - b^2$. The second term can be removed since $\eta \leq \frac{1}{L}$. Taking expectation on both sides and using the update variance bound Lemma 19,

$$\mathbb{E}f(\boldsymbol{y}_{i,k}^t) - \mathbb{E}f(\boldsymbol{y}_{i,k-1}^t) \leq -\frac{\eta}{2}\mathbb{E}\|\nabla f(\boldsymbol{y}_{i,k-1}^t)\|^2 + \frac{3\eta a^2 G^2}{2S}$$

$$+ \frac{3\eta}{2}\mathbb{E}\|\boldsymbol{e}^{t-1}\|^2 + \frac{3\eta\delta^2}{2}\Delta_{i,k-1}^t$$

$$\leq -\frac{\eta}{2}\mathbb{E}\|\nabla f(\boldsymbol{y}_{i,k-1}^t)\|^2 + \frac{3\eta a^2 G^2}{2S}$$

$$+ \frac{3\eta}{2}\mathbb{E}\|\boldsymbol{e}^{t-1}\|^2 + \frac{3\eta\delta^2}{2}\Delta_{i,k-1}^t$$

Multiplying the distance bound Lemma 20 by $\delta\left(1 + \frac{2}{K}\right)^{K-k}$. Note that for any $K \geq 1$ and $k \in [K]$, we have $1 \leq \left(1 + \frac{2}{K}\right)^{K-k} \leq 8$. Then we get

$$\delta\left(1 + \frac{2}{K}\right)^{K-k}\Delta_{i,k}^t \leq \delta\left(1 + \frac{2}{K}\right)^{K-k}\left(\left(1 + \frac{1}{K}\right)\Delta_{i,k-1}^t + 18\eta^2 K a^2\frac{G^2}{S}\right.$$

$$\left. + 18\eta^2 K\mathbb{E}\|\boldsymbol{e}^{t-1}\|^2 + 6\eta^2 K\|\nabla f(\boldsymbol{y}_{i,k-1}^t)\|^2\right)$$

$$\leq \delta\left(1 + \frac{2}{K}\right)^{K-(k-1)}\Delta_{i,k-1}^t - \frac{\delta}{K}\left(1 + \frac{2}{K}\right)^{K-k}\Delta_{i,k-1}^t$$

$$+ 48\eta^2\delta K\mathbb{E}\|\nabla f(\boldsymbol{y}_{i,k-1}^t)\|^2 + \frac{144\eta^2\delta K a^2 G^2}{S} + 144\eta^2\delta K\mathbb{E}\|\boldsymbol{e}^{t-1}\|^2$$

$$\leq \delta\left(1 + \frac{2}{K}\right)^{K-(k-1)}\Delta_{i,k-1}^t - \frac{\delta}{K}\Delta_{i,k-1}^t + 48\eta^2\delta K\mathbb{E}\|\nabla f(\boldsymbol{y}_{i,k-1}^t)\|^2$$

$$+ \frac{144\eta^2\delta K a^2 G^2}{S} + 144\eta^2\delta K\mathbb{E}\|\boldsymbol{e}^{t-1}\|^2.$$

Adding these two inequalities together yields

$$\mathbb{E}f(\boldsymbol{y}_{i,k}^t) + \delta\left(1 + \frac{2}{K}\right)^{K-k}\Delta_{i,k}^t \leq \mathbb{E}f(\boldsymbol{y}_{i,k-1}^t) + \delta\left(1 + \frac{2}{K}\right)^{K-(k-1)}\Delta_{i,k-1}^t$$

$$- \left(\frac{\eta}{2} - 48\eta^2\delta K\right)\mathbb{E}\|\nabla f(\boldsymbol{y}_{i,k-1}^t)\|^2$$

$$+ \left(\frac{3\eta}{2} + 144\eta^2\delta K\right)\mathbb{E}\|\boldsymbol{e}^{t-1}\|^2$$

$$+ \left(\frac{3\eta}{2} + +144\eta^2\delta K\right)\frac{a^2 G^2}{S}.$$

Using our bound on the step-size that $\eta \leq \frac{1}{192\delta K}$ implies that $\eta\delta K \leq \frac{1}{48*4}$. $\qquad\square$

## H.4 Change in each round

We now see how the quantities we defined change across rounds.

**Distance moved in a round.**

**Lemma 22.** *For MimeMVR updates* (13) *with* $\eta \leq \frac{1}{6K\delta}$ *and given* (A1) *and* (A2), *the following holds*

$$\Delta^t \leq 54K^2\eta^2 \, \mathbb{E}\|e^{t-1}\|^2 + \frac{54K^2\eta^2 a^2 G^2}{S} + \frac{1}{KS}\sum_{i,k} 18K^2\eta^2 \, \mathbb{E}\|\nabla f(\boldsymbol{y}_{i,k-1}^t)\|^2 \,,$$

*where we define* $\Delta^t := \mathbb{E}\|\boldsymbol{x}^t - \boldsymbol{x}^{t-1}\|^2$.

*Proof.* Starting from the MimeMVR update (13) and following the proof of Lemma 20,

$$\mathbb{E}\|\boldsymbol{y}_{i,k}^t - \boldsymbol{x}^{t-1}\|^2 = \mathbb{E}\|\boldsymbol{y}_{i,k-1}^t - \eta\boldsymbol{d}_{i,k}^t - \boldsymbol{x}^{t-1}\|^2$$

$$\leq \left(1 + \frac{1}{2K}\right)\mathbb{E}\|\boldsymbol{y}_{i,k-1}^t - \boldsymbol{x}^{t-1}\|^2 + (2K+1)\eta^2\,\mathbb{E}\|\boldsymbol{d}_{i,k}^t\|^2$$

$$\leq \left(1 + \frac{1}{2K}\right)\mathbb{E}\|\boldsymbol{y}_{i,k-1}^t - \boldsymbol{x}^{t-1}\|^2 + 6K\eta^2\,\mathbb{E}\|\boldsymbol{d}_{i,k}^t - \nabla f(\boldsymbol{y}_{i,k-1}^t)\|^2$$

$$\qquad\qquad + 6K\eta^2\,\mathbb{E}\|\nabla f(\boldsymbol{y}_{i,k-1}^t)\|^2$$

$$\leq \left(1 + \frac{1}{K}\right)\mathbb{E}\|\boldsymbol{y}_{i,k-1}^t - \boldsymbol{x}^{t-1}\|^2$$

$$\qquad\qquad + 18K\eta^2\,\mathbb{E}\|e^{t-1}\|^2 + \frac{18K\eta^2 a^2 G^2}{S} + 6K\eta^2\,\mathbb{E}\|\nabla f(\boldsymbol{y}_{i,k-1}^t)\|^2 \,.$$

Note that $\boldsymbol{x}^t = \frac{1}{S}\sum_{i\in\mathcal{S}}\boldsymbol{y}_{i,K}^t$ and so,

$$\mathbb{E}\|\boldsymbol{x}^t - \boldsymbol{x}^{t-1}\|^2$$

$$\leq \frac{1}{S}\sum_{i\in\mathcal{S}}\mathbb{E}\|\boldsymbol{y}_{i,K}^t - \boldsymbol{x}^{t-1}\|^2$$

$$\leq \frac{1}{S}\sum_{i\in\mathcal{S}}\sum_{k}\left(18K\eta^2\,\mathbb{E}\|e^{t-1}\|^2 + \frac{18K\eta^2 a^2 G^2}{S} + 6K\eta^2\,\mathbb{E}\|\nabla f(\boldsymbol{y}_{i,k-1}^t)\|^2\right)\left(1 + \frac{1}{K}\right)^{K-k}$$

$$\leq 54K^2\eta^2\,\mathbb{E}\|e^{t-1}\|^2 + \frac{54K^2\eta^2 a^2 G^2}{S} + \frac{1}{KS}\sum_{i,k} 18K^2\eta^2\,\mathbb{E}\|\nabla f(\boldsymbol{y}_{i,k-1}^t)\|^2 \,.$$

Here we used the inequality that for all $k$, $\left(1 + \frac{1}{K}\right)^{K-k} \leq 3$. $\qquad\square$

**Server momentum variance.** We compute the error of the server momentum $\boldsymbol{m}^{t-1}$ defined as $e^t = \boldsymbol{m}^t - \nabla f(\boldsymbol{x}^{t-1})$. Its expected norm can be bounded as follows.

**Lemma 23.** *For the momentum update* (16), *given* (A1) *and* (A2), *the following holds for any* $\eta \leq \frac{1}{51\delta K}$ *and* $1 \geq a \geq 2592K^2\delta^2\eta^2$,

$$\mathbb{E}\|e^t\|^2 \leq (1 - \tfrac{23a}{24})\,\mathbb{E}\|e^{t-1}\|^2 + \frac{3a^2 G^2}{S} + \frac{1}{KS}\sum_{i,k} 36K^2\delta^2\eta^2\,\mathbb{E}\|\nabla f(\boldsymbol{y}_{i,k-1}^t)\|^2 \,.$$

*Proof.* Starting from the momentum update (16),

$$e^t = (1 - a)e^{t-1}$$

$$+ (1 - a)\left(\frac{1}{S}\sum_{j\in\mathcal{S}^t}(\nabla f_j(\boldsymbol{x}^{t-1}) - \nabla f_j(\boldsymbol{x}^{t-2})) - \nabla f(\boldsymbol{x}^{t-1}) + \nabla f(\boldsymbol{x}^{t-2})\right)$$

$$+ a\left(\frac{1}{S}\sum_{j\in\mathcal{S}^t}(\nabla f_j(\boldsymbol{x}^{t-1}) - \nabla f(\boldsymbol{x}^{t-1}))\right) \,.$$

Now, the term $e^{t-1}$ does not have any information from round $t$ and hence is statistically independent of the rest of the terms. Further, the rest of the terms have mean 0. Hence, we can separate out the zero mean noise terms from the $e^{t-1}$ following Lemma 2 and then the relaxed triangle inequality Lemma 1 to claim

$$\mathbb{E}\|e^t\|^2 \leq (1-a)^2 \, \mathbb{E}\|e^{t-1}\|^2$$
$$+ 2(1-a)^2 \left\| \frac{1}{S} \sum_{j \in \mathcal{S}^t} (\nabla f_j(x^{t-1}) - \nabla f_j(x^{t-2})) - \nabla f(x^{t-1}) + \nabla f(x^{t-2}) \right\|^2$$
$$+ 2a^2 \left\| \frac{1}{S} \sum_{j \in \mathcal{S}^t} (\nabla f_j(x^{t-1}) - \nabla f(x^{t-1})) \right\|^2$$
$$\leq (1-a)^2 \, \mathbb{E}\|e^{t-1}\|^2 + 2(1-a)^2 \delta^2 \|x^{t-1} - x^{t-2}\|^2 + \frac{2a^2 G^2}{S} \, .$$

The inequality used the Hessian similarity Lemma 3 to bound the second term and the heterogeneity bound (A1) to bound the last term. Finally, note that $(1-a)^2 \leq (1-a) \leq 1$ for $a \in [0,1]$. We can continue by bounding $\Delta^{t-1}$ using Lemma 22.

$$\mathbb{E}\|e^t\|^2 \leq (1-a) \, \mathbb{E}\|e^{t-1}\|^2 + 2\delta^2 \Delta^{t-1} + \frac{2a^2 G^2}{S}$$
$$\leq (1-a) \, \mathbb{E}\|e^{t-1}\|^2 + \frac{2a^2 G^2}{S}$$
$$+ 108 K^2 \delta^2 \eta^2 \, \mathbb{E}\|e^{t-1}\|^2 + \frac{108 K^2 \delta^2 \eta^2 a^2 G^2}{S} + \frac{1}{KS} \sum_{i,k} 36 K^2 \delta^2 \eta^2 \, \mathbb{E}\|\nabla f(y_{i,k-1}^t)\|^2$$
$$\leq (1 - \tfrac{23a}{24}) \, \mathbb{E}\|e^{t-1}\|^2 + \frac{3a^2 G^2}{S} + \frac{1}{KS} \sum_{i,k} 36 K^2 \delta^2 \eta^2 \, \mathbb{E}\|\nabla f(y_{i,k-1}^t)\|^2 \, .$$

The last step used our bound on the momentum parameter that $1 \geq a \geq 2592 \eta^2 \delta^2 K^2$. Note that $\eta \leq \frac{1}{51 \delta K}$ ensures that this set is non-empty. $\qquad \square$

**Progress in one round.** Finally, we can compute the progress made in a round. Note that we need a technical condition that $f$ is $\delta$-weakly convex. However, this is only needed because we insist on running the algorithm on $S$ clients in parallel and then averaging their weights—the averaging requires weak convexity to ensure that the loss doesn't blow up. It has been experimentally observed in [43] that with the right initialization, averaging of the parameters does not increase the loss value and so weak convexity within this region might be vaalid. Finally note that if we instead simply run the local updates on a single chosen client with all the rest only being used to compute $c^{t-1}$, we will retain all convergence rates without needing weak-convexity.

**Lemma 24.** *For any round of MimeMVR with step size* $\eta \leq \min\left(\frac{1}{L}, \frac{1}{864 \delta K}\right)$ *and momentum parameter* $a \geq 912 \eta^2 \delta^2 K^2$. *Then, given that* (A1)–(A2) *hold and* $f$ *is* $\delta$-*weakly convex, we have*

$$\frac{\eta}{24 KS} \sum_{k \in [K], j \in \mathcal{S}^t} \mathbb{E}\|\nabla f(y_{i,k-1}^t)\|^2 \leq \Phi^{t-1} - \Phi^t + \frac{17 \eta a \delta^2 K^2 G^2}{S} \, ,$$

*where we define the sequence*

$$\Phi^t := \tfrac{1}{K} \, \mathbb{E}[f(x^t) - f^\star] + \frac{96 \eta}{23a} \, \mathbb{E}\|e^t\|^2 + \frac{8\delta}{K} \Delta^t \, .$$

*Proof.* We start by summing over the progress in single client updates as in Lemma 21

$$\sum_{k \in [K]} \frac{\eta}{4} \mathbb{E}\|\nabla f(\boldsymbol{y}_{i,0}^t)\|^2 \leq \mathbb{E} f(\boldsymbol{y}_{i,0}^t) + \delta\left(1 + \frac{2}{K}\right)^K \Delta_{i,0}^t$$

$$- \mathbb{E} f(\boldsymbol{y}_{i,K}^t) - \delta\Delta_{i,K}^t$$

$$+ 3\eta K \mathbb{E}\|\boldsymbol{e}^{t-1}\|^2 + \frac{3\eta K a^2 G^2}{S}$$

$$\leq \mathbb{E} f(\boldsymbol{y}_{i,0}^t) + 8\delta\Delta_{i,0}^t - \mathbb{E} f(\boldsymbol{y}_{i,K}^t) - \delta\Delta_{i,K}^t$$

$$+ 3\eta K \mathbb{E}\|\boldsymbol{e}^{t-1}\|^2 + \frac{3\eta K a^2 G^2}{S}$$

$$\leq \mathbb{E} f(\boldsymbol{x}^{t-1}) + 8\delta\Delta^{t-1} - \mathbb{E} f(\boldsymbol{y}_{i,K}^t) - \delta\Delta_{i,K}^t$$

$$+ 3\eta K \mathbb{E}\|\boldsymbol{e}^{t-1}\|^2 + \frac{3\eta K a^2 G^2}{S} \, .$$

Recall that $\Delta_{i,k}^t = \mathbb{E}\|\boldsymbol{y}_{i,k}^t - \boldsymbol{x}^{t-2}\|^2$ and $\boldsymbol{y}_{i,0}^t = \boldsymbol{x}^{t-1}$. This gives the last step above, making $\Delta_{i,0}^t = \Delta^{t-1}$. Then by the averaging Lemma 4, we have

$$\frac{1}{S}\sum_{j \in \mathcal{S}^t} \mathbb{E}[f(\boldsymbol{y}_{j,K}^t)] + \delta\Delta_{j,K}^t = \frac{1}{S}\sum_{j \in \mathcal{S}} \mathbb{E}[f(\boldsymbol{y}_{j,K}^t)] + \delta\,\mathbb{E}\|\boldsymbol{x}^{t-2} - \boldsymbol{y}_{j,K}^t\|^2$$

$$\geq \mathbb{E}[f(\boldsymbol{x}^t)] + \delta\,\mathbb{E}\|\boldsymbol{x}^{t-2} - \boldsymbol{x}^t\|^2 \, .$$

So by averaging our inequality over the sampled clients, and diving our summation over the updates by $K$, we get

$$\frac{\eta}{4KS}\sum_{k \in [K], j \in \mathcal{S}^t} \mathbb{E}\|\nabla f(\boldsymbol{y}_{i,k-1}^t)\|^2$$

$$\leq \tfrac{1}{K}\,\mathbb{E}[f(\boldsymbol{x}^{t-1})] + 3\eta\,\mathbb{E}\|\boldsymbol{e}^{t-1}\|^2 + \frac{8\delta}{K}\Delta^{t-1} - \tfrac{1}{K}\,\mathbb{E}[f(\boldsymbol{x}^t)] + \frac{3\eta a^2 G^2}{S} \, .$$

We can use the bound on $\Delta_t$ from Lemma 22 to proceed as

$$\frac{\eta}{4KS}\sum_{k \in [K], j \in \mathcal{S}^t} \mathbb{E}\|\nabla f(\boldsymbol{y}_{i,k-1}^t)\|^2$$

$$\leq \tfrac{1}{K}\,\mathbb{E}[f(\boldsymbol{x}^{t-1})] - \tfrac{1}{K}\,\mathbb{E}[f(\boldsymbol{x}^t)] + 3\eta\,\mathbb{E}\|\boldsymbol{e}^{t-1}\|^2 + \frac{3\eta a^2 G^2}{S}$$

$$+ \frac{8\delta}{K}\Delta^{t-1} - \frac{8\delta}{K}\Delta^t$$

$$+ 432K\delta\eta^2\,\mathbb{E}\|\boldsymbol{e}^{t-1}\|^2 + \frac{432K\delta\eta^2 a^2 G^2}{S} + \frac{1}{KS}\sum_{i,k} 144K\delta\eta^2\,\mathbb{E}\|\nabla f(\boldsymbol{y}_{i,k-1}^t)\|^2$$

$$\leq \tfrac{1}{K}\,\mathbb{E}[f(\boldsymbol{x}^{t-1})] - \tfrac{1}{K}\,\mathbb{E}[f(\boldsymbol{x}^t)] + 4\eta\,\mathbb{E}\|\boldsymbol{e}^{t-1}\|^2 + \frac{4\eta a^2 G^2}{S}$$

$$+ \frac{8\delta}{K}\Delta^{t-1} - \frac{8\delta}{K}\Delta^t + \frac{\eta}{6KS}\sum_{i,k} \mathbb{E}\|\nabla f(\boldsymbol{y}_{i,k-1}^t)\|^2$$

The last step used the bound on the step size that $\eta \leq \frac{1}{864\delta K}$. Now, multiplying the error bound Lemma 23 by $\frac{96\eta}{23a}$ gives

$$\frac{96\eta}{23a}\,\mathbb{E}\|\boldsymbol{e}^t\|^2 \leq \frac{4 * 24\eta}{23a}\left(1 - \tfrac{23a}{24}\right)\mathbb{E}\|\boldsymbol{e}^{t-1}\|^2 + \frac{13\eta a G^2}{S} + \frac{1}{KS}\sum_{i,k} \frac{38K^2\delta^2\eta^3}{a}\,\mathbb{E}\|\nabla f(\boldsymbol{y}_{i,k-1}^t)\|^2 \, .$$

Adding this to the previously obtained bound yields

$$\frac{\eta}{4KS} \sum_{k\in[K],j\in\mathcal{S}^t} \mathbb{E}\|\nabla f(\boldsymbol{y}_{i,k-1}^t)\|^2 \leq \left(\frac{1}{6} + \frac{38K^2\delta^2\eta^2}{a}\right) \frac{\eta}{KS} \sum_{k\in[K],j\in\mathcal{S}^t} \mathbb{E}\|\nabla f(\boldsymbol{y}_{i,k-1}^t)\|^2$$

$$+ \tfrac{1}{K}\,\mathbb{E}[f(\boldsymbol{x}^{t-1})] - \tfrac{1}{K}\,\mathbb{E}[f(\boldsymbol{x}^t)]$$

$$+ \frac{96\eta}{23a}\,\mathbb{E}\|\boldsymbol{e}^{t-1}\|^2 - \frac{96\eta}{23a}\,\mathbb{E}\|\boldsymbol{e}^t\|^2$$

$$+ \frac{8\delta}{K}\Delta^{t-1} - \frac{8\delta}{K}\Delta^t$$

$$- \tfrac{1}{K}\,\mathbb{E}[f(\boldsymbol{x}^t)] - \frac{4\eta}{a}\,\mathbb{E}\|\boldsymbol{e}^t\|^2$$

$$+ \left(13\eta a + 3\eta a^2\right)\frac{G^2}{S}\,.$$

Since $a \geq 912\eta^2 K^2\delta^2$, we have $\frac{1}{4} - \left(\frac{1}{6} - \frac{38K^2\delta^2\eta^2}{a}\right) \geq \frac{1}{24}$. Using this proves the lemma. $\qquad\square$

## H.5 Final convergence rates

**Theorem V** (Convergence of MimeMVR). *Let us run MimeMVR with step size* $\eta = \min\left(\frac{1}{L}, \frac{1}{864\delta K}, \left(\frac{S(f(\boldsymbol{x}^0)-f^\star)}{6936K^3T\delta^2 G^2}\right)^{1/3}\right)$ *and momentum parameter* $a = \max\left(1536\eta^2\delta^2 K^2, \frac{1}{T}\right)$.
*Then, given that* (A1) *and* (A2) *hold, we have*

$$\frac{1}{KST} \sum_{t\in[T]} \sum_{k\in[K]} \sum_{j\in\mathcal{S}^t} \mathbb{E}\|\nabla f(\boldsymbol{y}_{i,k-1}^t)\|^2 \leq \mathcal{O}\left(\left(\frac{\delta^2 G^2 F}{ST^2}\right)^{1/3} + \frac{G^2}{ST} + \frac{(L+\delta K)F}{KT}\right),$$

*where we define* $F := f(\boldsymbol{x}^0) - f^\star$.

*Proof.* Unroll the one round progress Lemma 24 and average over $T$ rounds to get

$$\frac{1}{KST} \sum_{t\in[T]} \sum_{k\in[K]} \sum_{j\in\mathcal{S}^t} \mathbb{E}\|f(\boldsymbol{y}_{i,k-1}^t)\|^2 \leq \frac{24(\Phi^0 - \Phi^T)}{\eta T} + \frac{408aG^2}{S}\,.$$

Recall that we defined

$$\Phi^t := \tfrac{1}{K}\,\mathbb{E}[f(\boldsymbol{x}^t) - f^\star] + \frac{96\eta}{23a}\,\mathbb{E}\|\boldsymbol{e}^t\|^2 + \frac{8\delta}{K}\Delta^t\,.$$

Hence, $\Phi^T \geq 0$. Further, note that by definition $\Delta^0 = 0$ and $\mathbb{E}\|\boldsymbol{e}_0\|^2 := \mathbb{E}\|\boldsymbol{m}^0 - \nabla f(\boldsymbol{x}^0)\|^2$. [14] show that by using time-varying step sizes, it is possible to directly control the error $\boldsymbol{e}_0$. Alternatively, [62] use a large initial accumulation for the momentum term. For the sake of simplicity, we will follow the latter approach. It is straightforward to extend our techniques to the time-varying step-size case as well but with additional proof complexity. Note that either way, the total complexity only changes by a factor of 2. Suppose that we run the algorithm for $2T$ rounds wherein for the first $T$ rounds, we simply compute $\boldsymbol{m}^0 = \frac{1}{T_0 S} \sum_{t=1}^{T_0} \sum_{j\in\mathcal{S}^t} \nabla f_j(\boldsymbol{x}^0)$. With this, we have $\boldsymbol{e}_0 = \mathbb{E}\|\boldsymbol{m}^0 - \nabla f(\boldsymbol{x}^0)\|^2 \leq \frac{G^2}{ST}$. Thus, we have for the first round $t = 1$

$$\Phi^0 = \tfrac{1}{K}\,\mathbb{E}[f(\boldsymbol{x}^0) - f^\star] + \frac{96\eta}{23a}\,\mathbb{E}\|\boldsymbol{e}^0\|^2 \leq \tfrac{1}{K}\,\mathbb{E}[f(\boldsymbol{x}^0) - f^\star] + \frac{96\eta G^2}{23aTS}\,.$$

Together, this gives

$$\frac{1}{KST} \sum_{t\in[T]} \sum_{k\in[K]} \sum_{i\in\mathcal{S}^t} \mathbb{E}\|f(\boldsymbol{y}_{i,k-1}^t)\|^2 \leq \frac{24(f(\boldsymbol{x}^0) - f^\star)}{\eta KT} + \frac{96G^2}{aT^2 S} + \frac{408aG^2}{S}\,.$$

The above equation holds for any choice of $\eta \leq \min\left(\frac{1}{L}, \frac{1}{864\delta K}\right)$ and momentum parameter $a \geq 912\eta^2\delta^2 K^2$. Set the momentum parameter as

$$a = \max\left(912\eta^2\delta^2 K^2, \frac{1}{T}\right)$$

With this choice, we can simplify the rate of convergence as

$$\frac{24(f(\boldsymbol{x}^0) - f^\star)}{\eta K T} + \frac{96G^2}{TS} + \frac{166464\eta^2\delta^2 K^2 G^2}{S} + \frac{408G^2}{ST} \, .$$

Now let us pick

$$\eta = \min\left(\frac{1}{L}, \frac{1}{864\delta K}, \left(\frac{S(f(\boldsymbol{x}^0) - f^\star)}{6936K^3 T\delta^2 G^2}\right)^{1/3}\right) \, .$$

For this combination of step size $\eta$ and $a$, the rate simplifies to

$$\frac{504G^2}{TS} + 916\left(\frac{(f(\boldsymbol{x}^0) - f^\star)\delta^2 G^2}{ST^2}\right)^{1/3} + \frac{24(L + 864\delta K)(f(\boldsymbol{x}^0) - f^\star)}{KT} \, .$$

This finishes the proof of the theorem. $\qquad\square$

**Theorem VI** (Convergence of MimeLiteMVR). *Let us run MimeLiteMVR with step size $\eta = \min\left(\frac{1}{L}, \frac{1}{864\delta K}, \left(\frac{(f(\boldsymbol{x}^0) - f^\star)}{6936K^3 T\delta^2 (G^2+\sigma^2)}\right)^{1/3}\right)$ and momentum parameter $a = \max\left(1536\eta^2\delta^2 K^2, \frac{1}{T}\right)$. Then, given that (A1) and (A2\*) hold, we have*

$$\frac{1}{KST}\sum_{t\in[T]}\sum_{k\in[K]}\sum_{j\in\mathcal{S}^t}\mathbb{E}\|\nabla f(\boldsymbol{y}_{i,k-1}^t)\|^2 \leq \mathcal{O}\left(\left(\frac{\delta^2(G^2+\sigma^2)F}{T^2}\right)^{1/3} + \frac{G^2+\sigma^2}{T} + \frac{(L+\delta K)F}{KT}\right),$$

*where we define $F := f(\boldsymbol{x}^0) - f^\star$.*

*Proof.* The proof for MimeLiteMVR is identical to that of MimeMVR, except that as noted in Lemma 18, the $\frac{G^2}{S}$ term in Mime gets replaced by $(G^2 + \sigma^2)$ everywhere. Note that MimeLiteMVR (Lemma 18) requires a weaker Hessian variance condition of $\|\nabla^2 f_i(\boldsymbol{x}) - \nabla^2 f(\boldsymbol{x})\| \leq \delta$ as opposed to MimeMVR which needs $\|\nabla^2 f_i(\boldsymbol{x}; \zeta) - \nabla^2 f(\boldsymbol{x})\| \leq \delta$. $\qquad\square$

Note that the final convergence rates of MimeMVR and MimeLiteMVR both include the intermediate client parameters. To implement this algorithm would require additional communication where in each round a random client parameter in $\{\boldsymbol{y}_{i,1}^t, \ldots, \boldsymbol{y}_{i,K}^t\}$ is communicated to the server. However, as is common in non-convex stochastic analysis, we expect the last iterate to converge at a similar rate as well in practice.

## I Algorithm pseudocodes

---
**Algorithm 2** FedAvg framework
---
    **input:** initial $\boldsymbol{x}$ and $\boldsymbol{s}$, server learning rate $\eta_g$, client learning rate $\eta_l$, and base optimizer $\mathcal{B} = (\mathcal{U}, \mathcal{V})$
    **for** each round $t = 1, \cdots, T$ **do**
      sample subset $\mathcal{S}$ of clients
      **communicate** $\boldsymbol{x}$ to all clients $i \in \mathcal{S}$
      **on client** $i \in \mathcal{S}$ **in parallel do**
        initialize local model $\boldsymbol{y}_i \leftarrow \boldsymbol{x}$
        **for** $k = 1, \cdots, K$ **do**
          sample mini-batch $\zeta$ from local data
          update $\boldsymbol{y}_i \leftarrow \boldsymbol{y}_i - \eta_l \nabla f_i(\boldsymbol{y}_i; \zeta)$
        **end for**
        **communicate** $\boldsymbol{y}_i$
      **end on client**
      compute aggregate pseudo-gradient $\boldsymbol{g} \leftarrow \frac{1}{|\mathcal{S}|}\sum_{i\in\mathcal{S}}(\boldsymbol{x} - \boldsymbol{y}_i)$
      $\boldsymbol{x} \leftarrow \eta_g\mathcal{U}(\boldsymbol{g}, \boldsymbol{s})$ (update server parameters)
      $\boldsymbol{s} \leftarrow \mathcal{V}(\boldsymbol{g}, \boldsymbol{s})$ (update optimizer state)
    **end for**
---

---

**Algorithm 3** **MimeMom** and **MimeLiteMom**

---

**input:** initial $\boldsymbol{x}$, and hyperparameters $\eta$, $\beta$. optional $\eta_g$ (default $= 1$)

initialize $\boldsymbol{m} \leftarrow 0$, $\boldsymbol{c} \leftarrow 0$

**for** each round $t = 1, \cdots, T$ **do**

    sample subset $\mathcal{S}$ of clients

    **communicate** $(\boldsymbol{x}, \boldsymbol{s} = \boldsymbol{m})$ and $\boxed{\boldsymbol{c}\ \text{(only Mime)}}$ to all clients $i \in \mathcal{S}$

    **on client** $i \in \mathcal{S}$ **in parallel do**

        initialize local model $\boldsymbol{y}_i \leftarrow \boldsymbol{x}$

        **for** $k = 1, \cdots, K$ **do**

            sample mini-batch $\zeta$ from local data

            $\boxed{\boldsymbol{g}_i \leftarrow \nabla f_i(\boldsymbol{y}_i; \zeta) - \nabla f_i(\boldsymbol{x}; \zeta) + \boldsymbol{c}\ \ \textbf{(Mime)}}$

            $\boxed{\boldsymbol{g}_i \leftarrow \nabla f_i(\boldsymbol{y}_i; \zeta)\ \ \textbf{(MimeLite)}}$

            update using server momentum $\boldsymbol{y}_i \leftarrow \boldsymbol{y}_i - \eta((1 - \beta_1)\boldsymbol{g}_i + \beta_1 \boldsymbol{m})$

        **end for**

        compute full local-batch gradient $\nabla f_i(\boldsymbol{x})$

        **communicate** $(\boldsymbol{y}_i, \nabla f_i(\boldsymbol{x}))$

    **end on client**

    compute $\boldsymbol{c} \leftarrow \frac{1}{|\mathcal{S}|} \sum_{i \in \mathcal{S}} \nabla f_i(\boldsymbol{x})$

    $\boldsymbol{m} \leftarrow ((1 - \beta_1)\boldsymbol{c} + \beta_1 \boldsymbol{m})$ (update server momentum)

    $\boldsymbol{x} \leftarrow \boldsymbol{x} - \eta_g \frac{1}{|\mathcal{S}|} \sum_{i \in \mathcal{S}} (\boldsymbol{x} - \boldsymbol{y}_i)$ (update server parameters)

**end for**

---
---

**Algorithm 4** **MimeAdam** and **MimeLiteAdam**

---

**input:** initial $\boldsymbol{x}$, and hyperparameters $\eta$, $\beta_1, \beta_2, \varepsilon_0$. optional $\eta_g$ (default $= 1$)

initialize $\boldsymbol{m} \leftarrow 0$, $\boldsymbol{v} \leftarrow 0$, $\boldsymbol{c} \leftarrow 0$

**for** each round $t = 1, \cdots, T$ **do**

    sample subset $\mathcal{S}$ of clients

    **communicate** $(\boldsymbol{x}, \boldsymbol{s} = (\boldsymbol{m}, \boldsymbol{v}))$ and $\boxed{\boldsymbol{c}\ \text{(only Mime)}}$ to all clients $i \in \mathcal{S}$

    **on client** $i \in \mathcal{S}$ **in parallel do**

        initialize local model $\boldsymbol{y}_i \leftarrow \boldsymbol{x}$

        **for** $k = 1, \cdots, K$ **do**

            sample mini-batch $\zeta$ from local data

            $\boxed{\boldsymbol{g}_i \leftarrow \nabla f_i(\boldsymbol{y}_i; \zeta) - \nabla f_i(\boldsymbol{x}; \zeta) + \boldsymbol{c}\ \ \textbf{(Mime)}}$

            $\boxed{\boldsymbol{g}_i \leftarrow \nabla f_i(\boldsymbol{y}_i; \zeta)\ \ \textbf{(MimeLite)}}$

            update $\boldsymbol{y}_i \leftarrow \boldsymbol{y}_i - \eta((1 - \beta_1)\boldsymbol{g}_i + \beta_1 \boldsymbol{m})/((\sqrt{\boldsymbol{v}} + \varepsilon_0)(1 - \beta_1^t))$

        **end for**

        compute full local-batch gradient $\nabla f_i(\boldsymbol{x})$

        **communicate** $(\boldsymbol{y}_i, \nabla f_i(\boldsymbol{x}))$

    **end on client**

    compute $\boldsymbol{c} \leftarrow \frac{1}{|\mathcal{S}|} \sum_{i \in \mathcal{S}} \nabla f_i(\boldsymbol{x})$

    $\boldsymbol{m} \leftarrow ((1 - \beta_1)\boldsymbol{c} + \beta_1 \boldsymbol{m})/(1 - \beta_1^t)$

    $\boldsymbol{v} \leftarrow ((1 - \beta_2)\boldsymbol{c}^2 + \beta_2 \boldsymbol{v})/(1 - \beta_2^t)$

    $\boldsymbol{x} \leftarrow \boldsymbol{x} - \eta_g \frac{1}{|\mathcal{S}|} \sum_{i \in \mathcal{S}} (\boldsymbol{x} - \boldsymbol{y}_i)$ (update server parameters)

**end for**

---

**Algorithm 5** MimeMVR pseudocode

---

**input:** initial $\boldsymbol{x}^0$, learning rate $\eta$

initialize $\boldsymbol{c}^0 \leftarrow 0$, $\boldsymbol{m}^0 \leftarrow 0$

**for** each round $t = 1, \cdots, T$ **do**
    sample subset $\mathcal{S}$ of clients
    **communicate** $\boldsymbol{x}^{t-1}, \boldsymbol{x}^{t-2}, m^{t-1}, c^{t-1}$ to all clients $i \in \mathcal{S}$
    **on client** $i \in \mathcal{S}$ **in parallel do**
        initialize local model $\boldsymbol{y}_{i,0}^t \leftarrow \boldsymbol{x}^{t-1}$
        **for** $k = 1, \cdots, K$ **do**
            sample mini-batch $\zeta_{i,k}^t$ from local data
            compute SVRG gradient $\boldsymbol{g}_{i,k}^t \leftarrow \nabla f_i(\boldsymbol{y}_{i,k-1}^t; \zeta_{i,k}^t) - \nabla f_i(\boldsymbol{x}^{t-1}; \zeta_{i,k}^t) + \boldsymbol{c}^{t-1}$
            compute corrected momentum $\boldsymbol{d}_{i,k}^t \leftarrow a\boldsymbol{g}_{i,k}^t + (1-a)\boldsymbol{m}^{t-1} + (1-a)(\nabla f_i(\boldsymbol{y}_{i,k-1}^t; \zeta_{i,k}^t) - \nabla f_i(\boldsymbol{x}^{t-1}; \zeta_{i,k}^t))$
            update $\boldsymbol{y}_{i,k}^t = \boldsymbol{y}_{i,k-1}^t - \eta \boldsymbol{d}_{i,k}^t$
        **end for**
        compute full local-batch gradients $\nabla f_i(\boldsymbol{x}^{t-1}), \nabla f_i(\boldsymbol{x}^{t-2})$
        **communicate** $(\boldsymbol{y}_{i,K}^t, \nabla f_i(\boldsymbol{x}^{t-1}), \nabla f_i(\boldsymbol{x}^{t-2}))$
    **end on client**
    compute new aggregate pseudo-gradient $\boldsymbol{c}^t \leftarrow \frac{1}{|\mathcal{S}|} \sum_{i \in \mathcal{S}} \nabla f_i(\boldsymbol{x}^{t-1})$
    compute old aggregate pseudo-gradient $\tilde{\boldsymbol{c}}^t \leftarrow \frac{1}{|\mathcal{S}|} \sum_{i \in \mathcal{S}} \nabla f_i(\boldsymbol{x}^{t-2})$
    update server momentum $\boldsymbol{m}^t \leftarrow a\boldsymbol{c}^t + (1-a)\boldsymbol{m}^{t-1} + (1-a)(\boldsymbol{c}^t - \tilde{\boldsymbol{c}}^t)$
    update server parameters $\boldsymbol{x}^t \leftarrow \frac{1}{|\mathcal{S}|} \sum_{i \in \mathcal{S}} \boldsymbol{y}_{i,K}^t$
**end for**

---