# OpenReview forum: "Breaking the centralized barrier for cross-device federated learning"
_NeurIPS.cc/2021/Conference — NeurIPS 2021 Poster_

### Official Review · Reviewer_aDPX · 2021-07-16

**Rating:** 6
**Confidence:** 4

**Summary:**

This paper studies cross-device federated learning setting where the number of clients is very large and most of the clients participate in only one round of communication. Because of this low client participation per round, it is hard to have any client state/memory. State is usually helpful in traditional federated learning setting (eg: in SCAFFOLD) in estimating client heterogeneity, which can then be used to correct for the bias of the client. To mitigate this, the authors propose two meta-algorithms: MIME and MIMELITE, to adaptive a general class of centralized algorithms to this cross-device setting. MIME uses variance reduction at the client (like SCAFFOLD) and MIMELITE does not.

The paper analyzes the effect of these meta-algorithms for three standard centralized (server-only) deep learning optimization methods: SGD, Adam, MomentumVarianceReduction (MVR). However, reviewer could not find the pseduo-code for the MVR in the paper. For SGD, and Adam authors show that MIME modification performs at least as good as their centralized base algorithms for nonconvex problems. For MVR, authors prove that the server-level momentum of MIME helps beat the performance of centralized baselines under low-heterogeneity.

**Limitations And Societal Impact:**

Authors could improve the discussion on limitations of their methods. Authors could mention some open questions arising from their study. For example, authors could discuss why MIME predominantly can only match the centralized rates, but they may outperform centralized methods in practice (mismatch of theory and practice). Authors could also mention why or why not MIME can help in convex setting.

**Main Review:**

Contributions are original and their quality seems solid, but it is hard to verify it due to lack of clarity (see below). Improving clarity could better convince the readers of the claims. Although it is not clear if the contributions are consistently useful in practice (see below), these analysis and the experiments could be impact for future research and practice. Reviewer is willing to reconsider the score if the clarity and empirical quality can be improved.

-Strengths
1. If the base centralized algorithms use a momentum, MIME and MIMELITE uses an unbiased estimator of the server-level momentum. These algorithms communicate this momentum to the participating clients at the beginning of each round, and the clients use it to correct their bias when taking gradient descent steps. Note that clients do not update the momentum for their whole round to maintain the lack of bias of the momentum.

2. It it shown that for SGD and Adam, MIME matches the performance of the corresponding centralized (without local gradient steps) base algorithm. Whereas, MIMELITE is slightly worse and it could match centralized performance under certain conditions.

3. For MVR algorithm, MIME is shown to perform much better than centralized algorithm when the client Hessian heterogeneity is smaller than the smoothness of the global loss. This seems to be first such result for smooth non-quadratic setting proving the benefit of federated algorithm over centralized methods. However it was hard to verify this claim as MVR algorithm is not given.

4. Authors report (small scale single-run) ablation study on 2 layer MLP with base SGD+momentum algorithm, to show that MIME & MIMELITE perform better than FedAvg in cross-device setting. It is shown that momentum helps MIMEs however it doesn’t affect FedAvg.

5. Reviewer appreciates that authors also report both positive and negative results they get on large-scale datasets. However, from the given single runs it is is not clear if their meta-algorithms are consistently better than baselines. One way or the other, it would be useful to know.



-Comments
1. Although there are only some subtle differences between original FedAvg/SCAFFOLD and MIME-SGD/MIMELITE-SGD, latter gets a much better rate nonconvex in cross-device setting. Can the authors provide more discussion on this difference and intuition behind this improvement?

2. App A: Over-all App A is confusing.
a. Fig 1: It is not clear which algorithm is FedAvg. I think authors use FedAvg both for some meta-algorithm (not given) and standard FedAvg.
b. What is “server-momentum” in App A and mentioned everywhere else? Can the authors please define it? Why is it called “momentum” and not some kind of pseudo “gradient” for FedAvg?

3. App B is confusing. It could be written more clearly. For example,
line 649: where is the convex case in Theorem IV? It is hard to understand the section due to the sudden analogy with convex case.

4. Table 1 and Appendix: Can the authors please add the reference for optimal statistical rate?

5. Where are the full pseudo codes of MVR & MIME-MVR algorithm given? Cannot verify the result without the algorithm. It is not given in Table 5. There is not MVR algorithm in [14].

6. Why are there no simulations of MVR and its MIME variants?

7. Table1:
a. Why is FedAvg rate not given?
b. What is FedSGD? Same for FedSGD, FedSGDm, and FedAdam in the Appendix.

8. What is the FedAvg meta algorithm used in the experiments? It is confusing to use FedAvg to refer to the original FedAvg and the some other meta algorithm from [45].

9. Table 2: It is not clear why authors are confident that numbers will hold the same over multiple runs. As this is an paper on optimization methods, it is appropriate give the numbers over multiple runs and give their standard errors. Most of the numbers are very close to each other, so it is not conclusive that Mime or Mimelite are useful for large datasets, except for Stackoverflow with Adam.

10. Are authors using “SGDm” and “Momentum” interchangeably?

11. Why does the theory (SGD,MVR) and expts (Adagrad,SGD+momentum) use different base algorithms? I was expecting to the see the experiment with MVR for different levels of heterogeneity.

12. Ablation study: it is not clear why MIME{Adam,Adagrad} is not compared with Fed{Adam,Adagrad} which are much more practical base algorithm. [https://openreview.net/forum?id=LkFG3lB13U5]

13. Although MIME predominantly can only match the centralized rates, but authors claim they may outperform centralized methods in practice (mismatch of theory and practice). Why is this the case?

14. Why is there no discussion of MIME on convex problems?


-Other Comments
1. line 890: $\pm$ is not a standard notation for adding and subtracting. Missing brackets for argument inside $\mathcal{U}$
2. line 677: uses i both outside and inside sum
3. line. 323: “”App ??”
4. When saying that “MimeMVR is the first algorithm which beat centralized rates”, are the authors claiming this for general nonconvex setting or is this applicable to convex problems too? Authors may want to be explicit about this for non-expert readers.
5. Algo 1: From lines 140-141 and 146-147 it is not clear how c & s are calculated. Authors could mention the direction of communication when calculating and sending c.
6. line 325: Why is sparsity of gradients useful?

—After author response—
\
I keep my score the same as I am still confused about some parts of the paper. I highly encourage authors to improve the clarity in the next revision.

—__After after__ author response—
\
With their latest response, authors have clarified most of my initial concerns. Unfortunately, I am still keeping my scores the same because now I am not sure if empirical results are directly validating the theoretical results.

**Time Spent Reviewing:**

7

---

> ### Author Response · Authors · 2021-08-06
> **Clarifications and improving writing**
>
>
> Thank you for the time invested in reviewing our work, and for your detailed feedback! We address the comments raised below.
>
> 1. There are two main sources of improvements to the rates: i) using MVR instead of momentum (this improvement is seen in the centralized setting as well as shown in [14, 56]), and ii) using an improved analysis via Hessian dissimilarity. The latter is completely novel to our work and required substantial new technical work (see Appendix B lines 668-673).
>
> 2. Server momentum refers to the FedAvgMom algorithm. Both “server momentum” and the FedAvg meta algorithm are defined in [45]. We will add descriptions of these to make our paper more consistent and self-contained. In Figure 2, the red arrows represent the momentum direction, the left side is FedAvgMom and the right side is MimeMom.
>
> 3. The discussion 648--652 (related to the convex case) was leftover from an earlier draft and should be removed. Thank you for the pointer.
>
> 4. The optimal lower bound for the stochastic non-convex case can be found in [5]. The general $\frac{G^2}{S}$ statistical lower bound is more “folklore” and holds for simple *mean estimation* (see Section F of [70] with $\alpha =2$ for an explicit construction).
>
> 5. The full MVR algorithm and its MimeMVR counterpart are described in App. G.1. It is a simplified version of [14] (see also [56]). We will add the full pseudocodes for MimeAdam and MimeMVR in the appendix.
>
> 6. MVR is known to be slower than standard momentum for empirical deep learning experiments, even though it has better theory. Hence, we wouldn’t expect MimeMVR to perform better than MimeMom. In general, our goal was to “mimic” centralized methods -- methods which have better empirical performance (momentum and Adam) we showed also perform well in the federated setting when combined with Mime, and similarly methods which have better theoretical rates (MVR) have good rates with Mime as well.
>
> 7. FedSGD -> FedAvgSGD (which is classically known as FedAvg), FedSGDm -> FedAvgMom, FedAdam -> FedAvgAdam. Sorry for the confusion, we will make sure to ensure consistency in our usage. FedAvgAdam has no known convergence results in the cross-device setting as far as we are aware.
>
> 8. The FedAvg meta algorithm is defined in [45]. We will make a pass to make sure we use FedAvg to only mean the meta algorithm, and FedAvgSGD to mean the algorithm.
>
> 9. We are in the process of repeating the experiments multiple times and will be sure to report them in the final version.
>
> 10. Yes. Thank you for the pointer, we will replace SGDm with momentum.
>
> 11. See point 6.
>
> 12. We also indeed ran FedAvg, Mime, and MimeLite with Adam. Exactly the same conclusions hold. We will include this plot in the Appendix.
>
> 13. MimeMVR is the first method (convex or non-convex) where an improvement is seen over centralized methods for general smooth functions. Previously, such an improvement was known only for the very special case of strongly convex quadratic functions without any stochasticity [29, 47].
>
> 14. The MVR base algorithm itself has only been analyzed in the non-convex case. Hence, we too analyze MimeMVR in this setting. Also, our goal was to develop new algorithms for federated deep learning, which are inherently non-convex. However, we believe our techniques can be adapted to the convex case as well.
>
> Further comments
>
> **Novelty**: See 13.
>
> **Gradient sparsity**: The utility of gradient sparsity for NLP has been previously used to heuristically justify the bad performance of Adam (see [69]). Also sparse gradients were shown to improve the “regret” of AdaGrad in online learning, though its applicability in deep learning is admittedly questionable.
>
> **Limitations and Societal impact**
> One of the main contributions is that Mime has an improved rate over centralized methods. E.g. when $\delta$ is small, the rate of MimeMVR is $\frac{1}{\epsilon}$ whereas centralized methods even when $\delta$ is small only have a rate of $\frac{1}{\epsilon^{3/2}$. What is true however is that FedAvg performs significantly better in practice than what the current theory indicates. Understanding this is an important open question.
>
> Further, testing the Mime framework in more benchmarks and also with other popular deep learning optimizers such as LARS, Yogi etc. are also excellent follow up questions.
>
> Theoretically, finding weaker versions of assumption (A2) under which similar improvements can be shown and designing algorithms which don't need full batch local gradients are great open questions.
>
> Finally, figuring out communication complexity lower bounds for the cross-device federated learning setting is a very challenging but important open question.

---

> > ### Comment · Reviewer_aDPX · 2021-09-02
> > **Details and Clarity missing**
> >
> > Thank you for your response. It addresses some of my comments.
> > \
> > \
> > However, I really wish the authors gave a more detailed and clarifying response to my comments. Currently, the reviewers need to refer to multiple very recent works to understand some of the background details ([45], [14], [5], [70] etc). This is challenging in a short review cycle. It would have been nicer if the authors could paraphrase the relevant information in the manuscript and the response. Additionally
> >
> > 1. Some of the reference doesn't even make sense: _"see Section F of [70] with $\alpha=2$ for an explicit construction"_. I couldn't find any such information in that reference.
> >
> > 2. Authors also seems to have misunderstood some of the comments. For example:
> > \
> > _"13. Although MIME predominantly can only match the centralized rates, but authors claim they may outperform centralized methods in practice (mismatch of theory and practice). Why is this the case?"_
> > \
> > This question was about the practical performance. But authors responded with something about the MVR algorithm.
> > \
> > \
> > _"1. Although there are only some subtle differences between original FedAvg/SCAFFOLD and MIME-SGD/MIMELITE-SGD, latter gets a much better rate nonconvex in cross-device setting. Can the authors provide more discussion on this difference and intuition behind this improvement?"_
> > \
> > This question was not about (MIME)MVR at all as the authors seems to have understood. The question was about why SCAFFOLD achieves worse rates than MIME-SGD.
> >
> > 3. Although the reviewer mentioned that App. A and B are confusing, the authors did not make any attempt to clarify these sections other than clarifying some typos.
> >
> > Overall, it looks like the response was not well thought out and I am still confused about some parts of the paper. Therefore, I keep my score the same. I highly encourage authors improve the clarity in the next revision.

---

> > > ### Author Response · Authors · 2021-09-02
> > > **Apologies for the shortcomings in our previous response**
> > >
> > >
> > > We thank the reviewer for the clarifications and apologize for our misunderstandings and shortcomings of the previous response. We hope the below response is more satisfactory.
> > >
> > > **FedAvg meta algorithm** utilizes a base optimizer, a client learning rate and a server learning rate. Each client performs K local update steps of SGD using the client learning rate and communicates the net update (difference between final and initial parameters) to the server. This difference is then treated as a ‘pseudo-gradient’ and is input into the optimizer (say momentum or Adam) to update the server parameters using the server learning rate. When the base optimizer uses momentum, this momentum is computed at the server level using the pseudo-gradients and is referred to as server momentum.
> > >
> > > **MVR algorithm** Standard momentum is of the form
> > > $$m^t = (1-\beta)\nabla f(x^{t-1}; \zeta^t) + \beta m^{t-1}\,.$$
> > >  This is an exponential moving average over many independent stochastic gradients. Hence, the variance of $m^t$ reduces by a factor of $(1-\beta)$ i.e if the variance of $\nabla f(x^{t-1}; \zeta^t)$ is $G^2$, then the variance of $m^t$ is $(1-\beta)G^2$. However, $E[m^t] \neq E[\nabla f(x^{t-1})]$ and so the update is biased. Thus, standard momentum reduces variance at the cost of increasing bias.
> > > Momentum based variance reduction (such as in [56]) adds an additional correction term to reduce this bias
> > > $$d^t = (1-\beta)\nabla f(x^{t-1}; \zeta^t) + \beta d^{t-1}  + \color{red}{\beta\big(\nabla f(x^{t-1}; \zeta^t) - \nabla f(x^{t-2}; \zeta^t) \big) }\,.$$
> > > The above can be rearranged to $d^t = \nabla f(x^{t-1}; \zeta^t) + \color{blue}{\beta (d^{t-1} - \nabla f(x^{t-2}; \zeta^t))}$. Now, by induction we can see that if $E[d^{t-1}] = \nabla f(x^{t-2})$ i.e. the blue term has mean 0 implies $E[d^{t}] = \nabla f(x^{t-1})$. Thus, by adding the red correction term above, we managed to fix the bias. Further, this correction is very small in magnitude since by smoothness,
> > > $$
> > > \|| \nabla f(x^{t-1}; \zeta^t) - \nabla f(x^{t-2}; \zeta^t) \||^2 \leq L^2\||x^{t-1} - x^{t-2} \||^2\,.
> > > $$
> > > Thus, if $x^{t-1}$ and $x^{t-2}$ are close to each other (i.e. we move slowly), the variance of $d^t$ will still only be $\approx (1-\beta)G^2$. This means $d^t$ is both unbiased and also has low variance. This is the key idea behind momentum based variance reduction in [14,56].
> > >
> > > Addressing additional points raised:
> > >
> > > **1.** Apologies for the typo, we meant [71]. We summarize the lower bound here. Consider the following 1D, 1-strongly convex, and 1-smooth optimization problem
> > > $$
> > > \min_{x \in R}E_{\zeta} (x - \zeta)^2 ,$$
> > > where $\zeta$ is a binomial random variable with mean $\mu$. The above problem is minimized at $x^\star = \mu$.
> > >
> > > Now, standard information theoretic results yield that we need $\Omega(\frac{1}{\epsilon})$ samples of $\zeta$ to get an estimate $\hat x$ with error $(\hat x - \mu)^2 \leq \epsilon$. This yields the required statistical lower bound.
> > >
> > > **2a.** Mime does not “only match the centralized rates” as stated by the reviewer. E.g. MimeMVR has rates faster than its centralized counterpart. While we were only able to prove such an improvement in the case of MVR, we believe that similar improvements should be possible with the other base algorithms as well. So we don’t see a contradiction between the theory and practice here.
> > >
> > > **2b.** The correction term $(c - c_i)$ utilized by SCAFFOLD is computed using the update of client $i$ in the round it was last sampled. E.g. if client 1 was sampled in round 1 and then again in round 1000, then round 1000 of SCAFFOLD uses $c_1$ computed in round 1. Thus, if clients repeat very infrequently the $c_i$ computed is very outdated and leads to a slow down. In contrast, the SVRG correction term in MimeSGD is computed using clients sampled in the current or previous rounds, and so is much more accurate. This difference becomes especially important when the total number of clients $N$ becomes very large since the frequency of sampling a client reduces as $\frac{S}{N}$. This is also reflected in the rate of convergence of SCAFFOLD which has this dependence on $\frac{S}{N}$ as seen in Table 1. When $N$ is small (e.g. in the cross-silo setting), we indeed expect SCAFFOLD to outperform MimeSGD.
> > >
> > > **Appendix A**: We described the FedAvg meta algorithm above and hope that this clarifies some of the terminology. Specifically, in Fig. 2 we compare FedAvgMomentum (left) with MimeMomentum (right).
> > >
> > > The local updates of FedAvgMom (shown in blue and green) overfit to the individual client, moving towards their individual optimum ($x_1^\star$ and $x_2^\star$ respectively). Thus, when averaged, FedAvgMomentum ends up converging to $\frac{x_1^\star + x_2^\star}{2}$ instead of the actual global optimum. Using server momentum (in red) only exacerbates the issue.
> > >
> > > In contrast, in MimeMomentum we 1) compute the momentum using only the initial update thereby ensuring its direction is unbiased, and 2) apply this momentum at every client step. Together, these significantly reduces the drift of the updates and is closer to the global optimizer $x^\star$.
> > >
> > > **Appendix B**: Here we attempt to provide an intuition behind our proof of convergence of MimeMVR. The main concern raised here related to the convex case, but as we said in our previous response, this was left over from a previous version and should have been removed. We attempt to summarize rest of the argument from the section here. There are three key components utilized in our proof: 1. Momentum - this we explained in the MVR algorithm paragraph above. 2. Local steps - we use a new proof technique to bound the bias caused by the local updates utilizing Hessian differences. 3. SVRG correction in Mime - this is computed directly on the clients sampled in the current or previous round. This replaces the correction in SCAFFOLD which required sampling the same client very frequently (see 2b. above).
> > >
> > > We hope that our current response is more complete and addresses the concerns of the reviewer and thank them for taking the time to point out the shortcomings.

---

> > > > ### Comment · Reviewer_aDPX · 2021-09-03
> > > > **Follow-up remarks**
> > > >
> > > > Thanks for your prompt response. This clears most of my issues. Please revise the paper keeping in mind these clarifications and include the points in your responses.
> > > >
> > > > Just to highlight again, only major concern not addressed in this thread is the lack of statistics of metrics over multiple runs. I believe you are running additional experiments. Do you have those numbers yet?

---

> > > > > ### Author Response · Authors · 2021-09-03
> > > > > **Multiple run experiments**
> > > > >
> > > > > We appreciate the constructive feedback and will definitely incorporate all the comments and discussion in the next version. Unfortunately, we do not yet have the results of the experiments with repeated seeds. These will be included in the next version as well.

---

> ### Comment · Reviewer_aDPX · 2021-09-02
> **Where are the train metrics?**
>
> I happened to notice that all the reported empirical results are either validation or test accuracy/loss. Why is training loss and accuracy not reported? Can you please report them for all the same plots?
>
> It is not clear how we can corroborate the optimization error convergence rate theory from the validation or testing metrics. Can you please comment on this discrepancy? For example, all these gains could have been in the generalization error instead in the optimization error.
>
> I am not saying these test accuracy results are not important. But the problem is the theory is for train accuracy/loss.

---

> > ### Author Response · Authors · 2021-09-03
> > **Results for train metrics are identical**
> >
> > We focused on test accuracy following the experimental setup of several prominent federated learning papers (e.g. https://arxiv.org/pdf/1602.05629.pdf, https://arxiv.org/pdf/1910.06378.pdf, https://arxiv.org/pdf/1610.05492.pdf,  https://arxiv.org/pdf/2003.00295.pdf). However, the train metrics have an identical trend and we will add them to our paper.
> >
> > Further, note that the cross-device setting makes very few passes over the full training data ($N \gg S$). Hence, generalization (over-fitting to the training data) is typically less of a concern. E.g. in EMNIST62 in 1k rounds we make less than 6 passes over all the clients, and in StackOverflow we only see 15\% of the clients (less than 1 pass).

---

### Official Review · Reviewer_zGM1 · 2021-07-16

**Rating:** 7
**Confidence:** 4

**Summary:**

The paper introduces a new generalized algorithmic optimization framework named MIME for adapting and analyzing the convergence of any optimization algorithm applied in centralized settings into cross-device federated learning environments. The algorithmic reduction is accomplished by decoupling the centralized optimization algorithm into optimizer (server-side) and parameter state (client-side) updates in federated settings. The convergence of the proposed adaptation is shown both theoretically and empirically over a range of challenging real-world datasets for different optimization algorithms. Interestingly, when applying the proposed federated adaptation for a specific momentum-based optimization algorithm, the adaptation is proven to be asymptotically faster than its centralized counterpart.

**Ethical Concerns:**

No ethical concerns.

**Limitations And Societal Impact:**

The work does not seem to have any limitations or potentially negative societal impact.

**Main Review:**

**Pros**
-	Interesting adaptation of any centralized optimization algorithm in federated learning settings, by decoupling the optimization problem into optimizer state (w/ complex) updates and parameter state (w/ linear) updates.
-	Proven upper bound convergence guarantees of the MIME (MIMELite) framework for different optimizers and comparison against their centralized counterpart. The upper bound convergence rate of the Momentum-based variance federated adaptation is proven to beat the lower bound of the centralized version when clients' heterogeneity is small.
-	Experiments on real-world datasets and large-scale experiment setup evaluation for EMNIST62, Shakespeare, and StackOverflow datasets.

**Cons**
-	The proposed method introduces additional communication cost in federated cross-device settings.
-	Even though the empirical evaluation is sufficient, further clarification is needed.
-	Paper structure is not necessarily coherent; some methods are introduced differently in some parts of the paper and are hard to follow.


**Discussion**

- Communication. With your proposed framework, there seems to be a lot more extra communication cost from server to clients (global model, server state, and SVRG style correction lines 140, 141) and clients to the server (updated local model weights, plus full-batch gradients). Can you propose any approaches on how this extra cost (at least on the client-end) can be mitigated? You address some part of the latter case in line 274, and in Appendix section C.3, but I think this should be introduced earlier in the text to alleviate some concerns.

- From my understanding, your analysis focus on the perspective that local batch gradient and parameter update are performed at the same client. Does the creation of the two subgroups (local batch gradient and parameter update) affect your convergence? It is clear that out of S clients, S/2 send full-batch gradients and S/2 parameter updates. Instead of creating two subgroups would a single subgroup of size S/2 lead to the same performance? My concern here is that since clients have heterogeneous distributions, considering only local batch gradients from half of the clients and parameter updates from the remaining might introduce additional noise in the state updates. Is such perspective captured in your theoretical evaluation?

- Is there a case that some of the participating clients cannot compute the full local batch gradient because of resource constraints (Line 146, Algorithm 1)? Should the subsampling method take this constraint into account?

- Experiments. Does Figure 1 show test accuracy or validation accuracy? Why do you provide the validation accuracy in Table 2, the testing accuracy would have been more informative. If this is not possible then, how were the validation datasets constructed and how was the validation accuracy computed across all clients? Moreover, showing the learning curves of the training methods presented in Table 2, based on federation round, would greatly simplify the understanding of the communication cost for each method. Finally, it would be nice to provide/plot the performance of the centralized model (e.g., as horizontal line) in Figure 1.

- Paper structure. In Table 1 you introduce the μ-PL inequality, but in the text there is no discussion on why this convergence analysis is important, what its intricacies are (except of course in Appendix E.1) and why its analysis is not provided for ADAM and MVR – it would be better to discuss this in the text. When presenting the empirical evaluation with Momentum, it is not very clear whether you are discussing server-side or client-side momentum (additional reading is needed). Moreover, the presented momentum results are without variance (not MVR), correct? If that’s the case then what is the convergence of MVR? In addition, in Table 1 you provide convergence guarantees for SGD, Adam, and MVR, however, from the Algorithm it is not clear how the MVR or the Adam is applied since the correction term (c) seems to be always present in the MIME, which is also why MIME is different from MIMELite. Can you please elaborate more in the text to address these discrepancies?

**Additional Related Work**

- In [1], Momentum GD is used as part of the client local step (without server-side optimization) and it is shown to provide accelerated convergence compared to vanilla SGD (FedAvg). Their approach seems to be very similar to your FedAvgMomentum training method.
- In the related work section, "Comparison to FedAvg and variants", in [2], the authors also assign more local steps to computationally fast learners in cross-silo settings and (empirically) show faster convergence.

[1] https://arxiv.org/pdf/1910.03197.pdf, [2] https://arxiv.org/pdf/2102.02849.pdf

**Text corrections / suggestions**

- Lines 100, 107: Notation D is a bit misleading when referring to clients (most of the time denotes dataset), would another notation (e.g., C) be more appropriate?
- Line 119: please add eq. or equation 1
- Line 130: change table 4 reference to table 5
- Line 133: while … while fragmentation
- Table 1: Why is MimeLiteSGD equivalent to FedSGD? Did you refer to FedAvg? FedSGD is different from FedAvg [3], similarly in Appendix section C.5 Line 759.
- Line 204: epsilon(ε) refers to accuracy or error? In your analysis, it refers to error
- Line 221: which increases to increase
- Line 259: Appendix D does not refer to proof sketch, change to B
- Line 285: Appendix B does not refer to the experimental setup, change to C
- Please be consistent with the Mime or MIME naming (see sections 4 and 5).
- Please align subfigures in Figure 1.
- Line 331: Statement slightly contradicts your empirical evaluation, which is also shown in Table 2 – MIME does not always outperform MIMELite.


[3] https://arxiv.org/pdf/1602.05629.pdf



**POST AUTHORS RESPONSE COMMENT:**
I read carefully the response of the authors to my points and the points raised by the other reviewers. My primary concern is the additional communication cost introduced by the framework in cross-device federated settings. In their follow-up answer, the authors propose an approach to reduce this cost through sampling and indicated (in another response) that the lower bound analysis of the communication complexity remains an open question. I would like to see this question and others raised during the reviewing process as part of the paper's discussion/future work section since it can help readers to further investigate these open problems. My score remains unchanged.



**Time Spent Reviewing:**

16

---

> ### Author Response · Authors · 2021-08-06
> **Clarifications regarding the discussions**
>
> Thank you for your detailed comments, careful reading, and the positive evaluation of our work. We appreciate the time you invested in reviewing! We address the comments below:
>
> - **Communication**: We agree that this is a key concern. We will add forward pointers to the discussion in C.3 and elsewhere earlier in the paper. The increased client -> server communication is easily reduced using subsampling as we discuss in line 274. Subsampling also can reduce the total communication sent by the server (Appendix C.2 ). Reducing the server -> client communication received by an individual client is harder, and one that might not be  possible without sacrificing the convergence rate. Intuitively, every client will need some additional information about the rest of the clients in order to prevent client drift.
>
> - **Subsampling clients**: Using different clients does not affect the proof at all (see lines 858--861). In fact, for the proof of MimeMVR, we can use ($S-1$) clients for computing full batch gradients and only $1$ client for computing the parameter updates and the theoretical guarantees still hold (see lines 1020--1026).
>
> - **Full-batch gradients**: This is a great question. For the practical side, we can simply replace the full batch gradients with a “large enough” batch. We also believe our theory should be easy to adapt to this setting, though it may no longer yield optimal rates.
>
> - **Experiments**: We report what is popularly called the “test accuracy”. We call it validation accuracy because the hyper-parameters of all algorithms (learning rate etc.) are tuned to give the best accuracy. This should (in real applications) not be done on the test set, but rather on the validation set. We will add the centralized accuracy to Fig.1. We see that, as expected, Mime/MimeLite > centralized > FedAvg.
>
> - **Paper structure**: We presented the $\mu$-PL convergence results to be able to better compare with existing works. MVR and Adam don’t have convergence results with PL-inequality as far as we are aware, and so we don’t attempt such an analysis for their Mime counterparts either. In contrast, FedAvg and SGD do have such analyses, and so we showed how Mime / MimeLite behaves in this setting.
> Apologies for the confusion regarding the algorithm descriptions. We will add full descriptions of MimeAdam and MimeMVR in the appendix. In summary, MimeAdam works by keeping the first and second moments fixed (the state s) and using the (c) correction for the local gradient update. Table 5 in the Appendix may provide some additional information--$m$ and $v$ are fixed and $g$ is the *corrected local gradient*.
> We did not experiment with MimeMVR since we expect it to be empirically slower than using simple momentum (MimeMom). This is because empirically, MVR is slower than standard momentum in the centralized setting (even though it has better theory). Our goal was to “mimic” centralized algorithms - ones with good empirical properties perform good empirically, and ones with good theoretical properties retain this.
>
> - **Related work and Typos**: Thank you for the pointers. The papers are definitely very relevant and will be included in our discussion.  We will also incorporate the writing suggestions made.

---

### Official Review · Reviewer_WoRC · 2021-07-17

**Rating:** 6
**Confidence:** 4

**Summary:**

The paper proposes MIME, a general algorithmic framework for federated learning (FL) that can adapt to (and match the performance of) arbitrary centralized algorithms like Adam, Momentum SGD, and SGD. The authors show that via the MIME framework the convergence of any centralized algorithm can be translated into the convergence of the algorithm in the FL setting. The authors also propose a momentum-based variance reduction (MVR) algorithm that the authors claim to be faster than any centralized algorithm. Finally, the authors conduct extensive numerical experiments to show that the algorithms under the MIME framework perform well on real datasets.

**Ethical Concerns:**

None noted.

**Limitations And Societal Impact:**

None noted.

**Main Review:**

Overall the paper is well written with a clear presentation. The ideas presented in the paper are significantly original and of importance to the research community. However, there are a few issues with the paper, predominantly with the technical content of the paper which the authors should address. The detailed comments are listed below:

1.	Theorem I (Corollary II and III): It seems that there is a technical issue with the results presented in Theorem I which then affects the results presented in Corollaries II and III too. Please clarify the following issue.

In the proof of Theorem I, the authors have used the fact that $\sum_k \nabla f_i(x; \zeta_{i,k}^t) = K \nabla f_i(x)$ (in Lines 873 and 877 of the supplementary material) which follows from assuming $K$ to be the number of epochs.  This implies that either $\zeta_{i,k}^t$ are sampled without replacement or $\zeta_{i,k}^t$ represents the full data, i.e., $\nabla f_i(x; \zeta_{i,k}^t) = \nabla f_i(x)$. In my understanding, this means that the stochastic gradient is not an unbiased estimate of the local gradient (or it is the full gradient). However, in Assumption (A3) this stochastic gradient is assumed to be an unbiased estimate of the local client’s gradient. The proof Theorem I (via Lemmas 7, 8) and Corollaries II and III rely on Assumption (A3) which might not hold if $\sum_k \nabla f_i(x; \zeta_{i,k}^t) = K \nabla f_i(x)$, as in my understanding Assumption (A3) and $\sum_k \nabla f_i(x; \zeta_{i,k}^t) = K \nabla f_i(x)$ both cannot be true at the same time. If my observation is correct the proofs of Theorem I and Corollaries II and III do not hold with the stated set of assumptions. Please clarify.

2.	Assumptions: Note that the linearity of the update $\mathcal{U}(g_1 + g_2) = \mathcal{U}(g_1)  + \mathcal{U}(g_2)$ in Algorithm 1 (and Theorem I) might not hold in general unless $g_1$ and $g_2$ come from the same round of Algorithm 1. This can be seen from the fact that for adaptive algorithms like Adam updates are not linear in $g$, however, since the adaptive step is fixed at the start of the round, linearity will hold, but $g_1$ and $g_2$ must be generated within the same round. Please update the assumption accordingly.

3.	Speed-up over centralized methods: The authors claim that MIME with MVR beats the lower bound for centralized methods, thus breaking a fundamental barrier. These claims are a bit misleading as the lower bounds corresponding to the centralized algorithms are on the computations, whereas the authors measure the total communication rounds required to reach an approximate stationary point, wherein within each round the clients compute multiple gradients. MimeMVR required total communication of $O(1/\epsilon^{3/2})$, however, it is well known that for FL problems better  $O(1/\epsilon)$ communication requirement can be achieved.

4.	The authors should include Assumption (A3) in the main text of the paper as it is important and $\sigma^2$ appears in the statements of Theorems and Corollaries. Moreover, the assumption $\mathbb{E}_i [\nabla f_i(x)]  = \nabla f(x)$ is not stated in the paper, however it is being used in the proofs.

5.	The bounded Hessian variance and L-smooth assumptions (Assumptions (A2) and (A2*)) are strong assumptions as they are made on the stochastic samples of the local client’s data. In fact, (A2) loosely forces the data at each client to be homogeneous. Generally, for FL problems (A2) is not required and a weaker version of L-smoothness (A2*), namely, mean square smoothness is required if some sort of variance reduction is used. It seems the analysis relies on these stronger assumptions. The authors mention that Hessian similarity is crucial for federated optimization and refer to [6], however [6] considers a convex problem that is different from the non-convex problems authors tackle. Please clarify.  Moreover, the MIME framework relies on computing the full gradients of the sampled clients at each round which seems to be a stronger requirement compared to existing FL algorithms.

6.	In the discussion of Theorem IV the authors discuss that if $\delta \approx 0$ then MimeMVR requires $O(1/\epsilon)$ communication rounds. However, the statement of Theorem IV requires $K \geq L/\delta$, which implies as $\delta \to 0$ we will have $K \to \infty$. What implication this will have on the performance of MimeMVR?

7.	In the Appendix, the authors mention that Mime MVR does not exactly fit the MIME framework as it relies on gradient computation at two consecutive points. I think instead of just including the convergence result of the MVR in the main paper, the authors should also include the MVR algorithm in the main paper since it seems to be a non-trivial construction.   Also, in Table 5 it would be helpful if the authors can add the tracking step used by the Mime framework rather than the tracking step of the centralized algorithm.

8.	Experiments:
The authors have conducted sufficient experiments to validate the proposed algorithms. A few concerns are discussed next. In the Experiments section, from Figure 1 it seems that Mime and MimeLite clearly outperform the rest of the algorithms, however, in Table 1 the reported performance of Mime and MimeLite is very close to other algorithms. Why is this happening?
Moreover, it seems that Mime and MimeLite compute more stochastic gradients per client compared to FedAvg between two communication rounds. If this is true isn’t it true that the improved performance of Mime framework is expected?

9.	Typos and minor issues:

$x^{\text{out}}$ in the statement of Corollary II is not defined.

Line 832: Replace $(\mathbb{E}[X - \mathbb{E}[X]])^2$ by $\mathbb{E}[X - \mathbb{E}[X]]^2$

Line 853: The inequality is in the wrong direction.

Expression in the statement of Lemma 4: Please use braces with the l.h.s. term to show that the sum over $i \in \mathcal{S}$ over the complete term.

With the proofs of Theorems, Lemmas, and Corollaries the authors should explain the reasoning behind inequalities (and equalities).

Line 974-975: Please state the tracking step for MimeAdam and MimeLiteAdam.

The authors first use $e^t$ to denote the perturbation in the update direction of the centralized algorithm and then later in MVR for denoting the error in server momentum. Please use a different symbol for each if possible.


**Time Spent Reviewing:**

8 hours

---

> ### Author Response · Authors · 2021-08-06
> **Clarifications regarding the theory**
>
> Thank you for such a thorough and careful reading of our work! We address the points raised below.
>
> 1. This is a great catch, and one which we had overlooked in our proof. We can easily remove the sampling without replacement requirement in Lemma 5. This can be done by re-defining the error term as
> $$
> e_t = \sum_{k} [\nabla f_i(y_{i,k-1}^t; \zeta_{i,k}^t) - \nabla f_i(x)]
> $$
> Then, the statement of Lemma 6 can be subsequently modified by first removing the variance to get
> $$
> E \|| e_t \||^2 \leq E\||\sum_{k} [\nabla f_i(y_{i,k-1}^t) - \nabla f_i(x)]\||^2 + \frac{\sigma^2}{KS} \leq L^2 E_k^t + \frac{\sigma^2}{KS} \leq \frac{G^2}{S}\,.
> $$
> The last inequality assumed $K \geq \frac{\sigma^2}{G^2}$. This additional term is quite small and does not affect the rest of the proof beyond constants. In particular, line 180 of Theorem I will become $E\||g_t - \nabla f(x_{t-1}) \||^2 \leq \frac{2G^2}{S}$ for $K \geq \frac{\sigma^2}{G^2}$, leaving the rest unchanged.
>
> 2. We only assume linearity of $U$ over $g$ only for a fixed optimizer state $s$, i.e. $U(g_1 + g_2, s) = U(g_1, s) + U(g_2, s)$ for any given optimizer state $s$. Across multiple rounds, the state $s$ may change and hence the assumption is no longer applicable, whereas it is kept fixed within a single round. We will make this dependence on $s$ more explicit in our theorem statement and assumption.
>
> 3. Firstly, under the non-iid *cross-device* setting studied here, we don’t know any work which shows $\frac{1}{\epsilon}$ communication under standard assumptions. This is only possible either i) by additionally assuming PL-inequality, or ii) in the cross-silo setting with a dependence on $N$ (e.g. by SCAFFOLD as in Table 1).
> Secondly, as we state in lines 217--223, indeed the lower bounds are on computation and not on communication. However, centralized server-only methods which do not take local steps have equal computation and communication. Hence, to beat the lower bound requires methods which can take additional local steps. While it may be very intuitive that taking additional local steps gives an advantage (because we are using more computation),  it has been very challenging to show any theoretically faster rates---all rates for local step methods thus far either matched or were slower than their centralized server-only counterparts.
>
> 4. We agree regarding the assumption statements and will modify them as suggested.
>
> 4. It is true that [6] only studies the convex version--we only use their results to motivate the definition of $\delta$ Hessian dissimilarity. Obtaining communication complexity lower bounds for the distributed stochastic non-convex setting in the style of [6] remains an open problem. This is the reason why we don’t claim that our dependence on $\delta$ is optimal (though we believe it is), since no lower bounds are known.
> Similarly, we agree that (A2) and (A2*) can probably be further weakened (see lines 794--799). In fact, the proof of MimeLiteMVR only needs (A2) to hold between the clients and not on the stochastic samples. However, it obtains slower rates than MimeMVR. We believe that instead of using Mime with the SVRG correction, if we use MimeLiteMVR with SPIDER correction, it may be possible to simultaneously weaken (A2) and remove the need for computing full local gradients. However, this is non-trivial and we leave it for future work.
>
> 5. The first term in Theorem IV is of the order $\frac{\delta}{\epsilon^{1.5}}$. Thus, $\delta = \sqrt{\epsilon}$ already gives an $\frac{1}{\epsilon}$ rate. With this, we will only need $K = \frac{1}{\sqrt{\epsilon}}$. This is to be expected since the total computation cannot be smaller than $\frac{1}{\epsilon^{3/2}}$ i.e we can only reduce communication by increasing local computation.
>
> 6. We were forced to move this to the Appendix for a lack of space. We will strongly consider the suggestions and do our best to incorporate them.
>
> 7. The difference in performance between Figure 1 (10 local epochs) and Table 1 (1 local epoch) we conjecture is because of the number of local rounds. FedAvg seems to fail quite badly when a lot of local steps are used (see Section 5.5 in the recent work [Charles et al. 2021](https://arxiv.org/abs/2106.07820). In Figure 1, only half the clients perform local steps in Mime and MimeLite and the other half compute the full client gradients (see lines 271--277). In Table 1, this is even smaller. Thus, in all our experiments, the number of stochastic gradients computed by Mime/MimeLite is smaller than FedAvg.
>
> 8. Thank you for the pointers! We will fix these immediately.

---

> > ### Author Response · Authors · 2021-09-02
> > **Additional clarifications?**
> >
> > Dear reviewer, we hope that our response above has satisfied the concerns previously raised. If not, we would be happy to provide any additional clarifications required.
> >
> > Thank you for your time!

---

> > ### Comment · Reviewer_WoRC · 2021-09-11
> > **Response to Rebuttal**
> >
> > Thanks for the response. The authors sufficiently addressed my concerns. One issue I have is the claim that their algorithm beats the centralized lower bound, which is expected because they allow multiple local updates. But nonetheless, I can raise my score to 6 after the rebuttal.

---

> > > ### Author Response · Authors · 2021-09-14
> > > **Thank you for the update!**
> > >
> > > We appreciate the updated  and the increased score :)
> > >
> > > > One issue I have is the claim that their algorithm beats the centralized lower bound, which is expected because they allow multiple local updates.
> > >
> > > Indeed, we circumvent centralized lower bounds using local steps. While it may be intuitive that local steps should help, designing an algorithm for which this is provably true has been a long standing open question. MimeMVR obtains the first such speed up and represents a major breakthrough in cross device federated learning.

---

### Official Review · Reviewer_ebxx · 2021-07-19

**Rating:** 6
**Confidence:** 3

**Summary:**

The paper proposes a framework MIME to convert centralized optimization algorithms to the federated learning setting. The key components are some control variates to reduce the effect of data distribution heterogeneity.
Contribution:
1. A framework to convert centralized algorithms to the federated learning setting.
2. Theoretical analysis to characterize the convergence of converted algorithms
3. Experiments show MIME framework can have better performance than FedAvg.

**Limitations And Societal Impact:**

Yes.

**Main Review:**

Strengths:
1. The convergence rate of converted algorithms matches their centralized versions.
2. MimeMVR provides an improved convergence rate assuming small Hessian variance. This is an interesting theoretical result.
3. The algorithms converted by Mime show strong empirical performance.

Weaknesses:
1. Since the 'breaking the lower bound' contribution is highlighted in the title, this point may deserve more discussion. More specifically, it will be helpful to discuss some technicality on how to use the Hessian variance to improve convergence rate and why centralized algorithms cannot achieve such a rate.


--------------after rebuttal-------------
My concerns are addressed.

**Time Spent Reviewing:**

2

---

> ### Author Response · Authors · 2021-08-06
> **Why small Hessian variance helps local methods**
>
> We explain the role of Hessian variance in Appendix B (ll. 662--673). A sketch of the argument follows.
> Suppose we sampled client $i$ and there is no local variance ($\sigma = 0$). At the first step, we have $E[\nabla f_i(x)] = \nabla f(x)$ i.e. the gradient is unbiased. However, after taking multiple steps and computing client parameters $y_i$, $E[\nabla f_i(y_i)] \neq \nabla f(y_i)$. This is because $y_i$ now also depends on client $i$. This bias then causes client drift, leading to slow convergence.
>
> We show that the bias introduced in such local steps is controlled by the Hessian variance. To see this, define $\Psi(z) = f_i(z) - f(z)$ and if the Hessian variance is smaller than $\delta$, then $\Psi$ is $\delta$-smooth. Then, we have
> $$
> E[\nabla f_i(y_i) - \nabla f(y_i)] = E[\nabla f_i(y_i) - \nabla f_i(x) + \nabla f(x) - \nabla f(y_i)] = E[\nabla \Psi(y_i) - \nabla \Psi(x)] \approx \nabla^2 \Psi(x) (y_i - x) \approx \delta(y_i - x)\,.
> $$
> Thus, when $\delta$ is small, local update methods have low bias and converge faster. In contrast, centralized server only methods always take only a single step in each round. So they cannot take advantage of small Hessian variance.

---

### Author Response · Authors · 2021-08-06
**Summary of author response**

We are truly grateful for the detailed, thoughtful feedback and the amount of time invested in reviewing our work! Many excellent questions were raised, and small gaps in our current work were pointed out. We believe that, along with some minor edits to our work, our responses address all of them.

Main edits we will perform to improve our work include:
1. We will modify Lemma 6 to account for sampling without replacement. This does not affect the rest of the proofs or final results beyond numerical constants.
2. We will re-run the experiments in Table 2 with different seeds to get variance measurements.
3. We will improve the readability of the paper utilizing the extra space provided by the final version and make it more self contained. In particular, we will add full pseudocodes for the FedAvg meta-algorithm, MimeAdam, and MimeMVR. Other numerous suggestions will also be incorporated.

We address individual concerns raised below.

---

### Decision · Program_Chairs · 2021-09-27

**Decision:**

Accept (Poster)

**Comment:**

Most of the concerns of the reviewers were addressed during the rebuttal period and the reviewers unanimously believe that the paper is above the acceptance threshold as the theoretical results are novel. Please revise the paper according to the reviewer's comments to clarify the confusions. In addition, please address the following concern which was raised by the reviewers in the discussion period:

-- All plots are for test accuracy/error. Please include the optimization error convergence rate plots to have a better connection with the rest of the paper.